# BILATERAL DISTRIBUTION COMPRESSION: REDUCING BOTH DATA SIZE AND DIMENSIONALITY

## ABSTRACT

Existing distribution compression methods reduce dataset size by minimising the *Maximum Mean Discrepancy* (MMD) between original and compressed sets, but modern datasets are often large in both sample size and dimensionality. We propose *Bilateral Distribution Compression* (BDC), a two-stage framework that compresses along both axes while preserving the underlying distribution, with overall linear time and memory complexity in dataset size and dimension. Central to BDC is the *Decoded MMD* (DMMD), which quantifies the discrepancy between the original data and a compressed set decoded from a low-dimensional latent space. BDC proceeds by (i) learning a low-dimensional projection using the *Reconstruction MMD* (RMMD), and (ii) optimising a latent compressed set with the *Encoded MMD* (EMMD). We show that this procedure minimises the DMMD, guaranteeing that the compressed set faithfully represents the original distribution. Experiments show that across a variety of scenarios BDC can achieve comparable or superior performance to ambient-space compression at substantially lower cost.

## 1 INTRODUCTION

Given a dataset $\mathcal{D} := \{x_i\}_{i=1}^n \subset \mathbb{R}^d$ with $n, d \gg 1$, sampled i.i.d. from a distribution $\mathbb{P}_X$, a key challenge is to construct a compressed set $\mathcal{C} := \{z_j\}_{j=1}^m \subset \mathbb{R}^p$ with $m \ll n$ and $p \ll d$ that preserves the essential statistical properties of the original data. We refer to this problem as *Bilateral Distribution Compression* (BDC), highlighting that compression acts simultaneously on both the number of samples and their dimensionality. This challenge is especially pressing in the era of frontier models, where training requires enormous amounts of data, computation, and energy, with associated costs in money, time, and environmental impact (Thompson et al., 2020). Data reduction methods aim to mitigate these burdens by constructing smaller, representative datasets that retain distributional fidelity while enabling more efficient large-scale learning.

Existing distribution compression methods include Kernel Herding (Chen et al., 2010; Bach et al., 2012), Kernel Thinning (Dwivedi & Mackey, 2021b; Shetty et al., 2022; Dwivedi & Mackey, 2021a; Gong et al., 2025), and Gradient Flow approaches (Arbel et al., 2019; Chen et al., 2025). These methods minimise the *Maximum Mean Discrepancy* (MMD) (Gretton et al., 2012) between the empirical distributions of the compressed set $\hat{\mathbb{P}}_Z$, and the target dataset $\hat{\mathbb{P}}_X$. However, these approaches focus exclusively on reducing the number of observations ($m \ll n$), overlooking the fact that modern datasets are often large both in sample size *and* in dimensionality. Dimensionality reduction techniques typically focus on preserving pairwise similarities or distances, popular examples include PCA (Maćkiewicz & Ratajczak, 1993; Schölkopf et al., 1998), t-SNE (van der Maaten & Hinton, 2008), UMAP (McInnes et al., 2020), and autoencoders (Goodfellow et al., 2016). These methods map data into lower-dimensional spaces ($p \ll d$) but do not reduce the number of observations.

To address this gap, we propose BDC, a two-stage method that minimises the *Decoded Maximum Mean Discrepancy* (DMMD), which measures how well a compressed set, once decoded, preserves the original distribution in ambient space. Directly optimising the DMMD is challenging, as it requires simultaneous optimisation of both the decoder and the compressed set. Moreover, a highly flexible decoder risks overfitting by simply memorising the target distribution rather than learning a meaningful representation. To avoid this, we adopt a two-stage procedure:

1. **Autoencoder training**: Train an encoder $\psi : \mathbb{R}^d \to \mathbb{R}^p$ and decoder $\phi : \mathbb{R}^p \to \mathbb{R}^d$ by minimising the *Reconstruction MMD* (RMMD) between $\mathcal{D}$ and its reconstruction $\phi(\psi(\mathcal{D}))$, ensuring sufficient regularisation to avoid memorisation.
2. **Latent compression**: With $\phi, \psi$ fixed, project $\mathcal{D}$ into latent space and optimise a compressed set $\mathcal{C} \subset \mathbb{R}^p$ by minimising the *Encoded MMD* (EMMD) between $\mathcal{C}$ and $\psi(\mathcal{D})$.

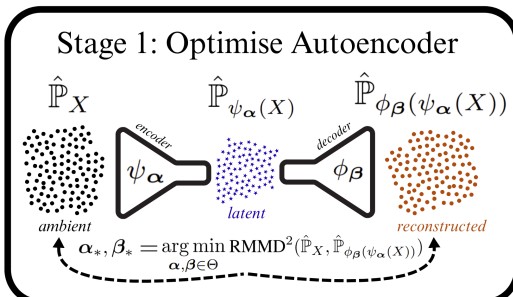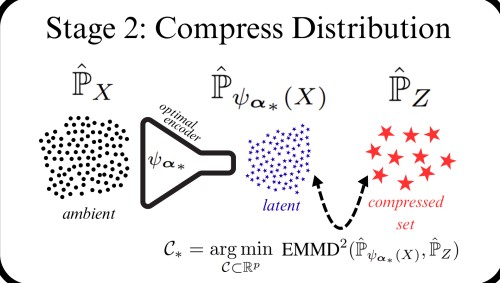

Figure 1: The Bilateral Distribution Compression (BDC) framework.

We show that vanishing RMMD and EMMD together imply vanishing DMMD, and moreover, for a certain choice of latent space kernel, the DMMD can further be bounded by the sum of RMMD and EMMD. We consider both linear autoencoders, idempotent rank-$p$ projections of the form $\phi(\psi(\boldsymbol{x})) = \boldsymbol{x} V V^\top$ with $V^\top V = I_p$, and neural autoencoders. Section B contains proofs of all our theoretical results.

**Our key contributions are:**

1. In Section 3, we formulate the problem of compressing datasets in both observations ($m \ll n$) and dimensionality ($p \ll d$), and introduce the Decoded MMD (DMMD) as a natural objective for measuring distributional similarity.
2. In Section 3.1, we propose a two-stage procedure: (i) train an autoencoder $\phi \circ \psi$ via RMMD, and (ii) optimise a latent compressed set via EMMD. We prove that zero RMMD and EMMD imply zero DMMD, and show that for certain kernels DMMD is bounded by their sum.
3. In Section 3.1.2, we motivate the use of RMMD in dimensionality reduction, showing that minimising RMMD with a quadratic kernel recovers Principal Component Analysis.
4. In Section 3.3, we extend BDC to labelled data through supervised DMMD, RMMD, and EMMD variants, incorporating responses into both autoencoder training and latent space compression.
5. In Section 4, we present empirical results across a range of tasks, datasets, and metrics, showing that BDC is able to match or surpasses the performance of ambient-space distribution compression methods while incurring lower cost.

## 2 PRELIMINARIES AND RELATED WORK

In this section, we set up notation, review the necessary background on RKHSs, and discuss related work. For surveys of RKHS theory, see Berlinet & Thomas-Agnan (2004); Muandet et al. (2017).

**Notation**: Let $(\Omega, \mathcal{F}, \mathbb{P})$ be the underlying probability space. Define random variables $X : \Omega \to \mathbb{R}^d$ and $Z : \Omega \to \mathbb{R}^p$, with distributions $\mathbb{P}_X$ and $\mathbb{P}_Z$. Here, $X$ represents ambient-space features and $Z$ intrinsic-space features. Given a dataset $\mathcal{D} := \{\boldsymbol{x}_i\}_{i=1}^n$, with $\boldsymbol{x}_i \in \mathbb{R}^d$, we denote its empirical distribution by $\hat{\mathbb{P}}_X$, and for a compressed set $\mathcal{C} := \{\boldsymbol{z}_i\}_{i=1}^m$, with $m \ll n$ and $\boldsymbol{z}_i \in \mathbb{R}^p$, $p \ll d$, we denote its empirical distribution by $\hat{\mathbb{P}}_Z$. For a measurable map $\psi : \mathbb{R}^d \to \mathbb{R}^p$ and $\mathbb{P}_X$, the *pushforward measure* (Çinlar, 2011) of $\mathbb{P}_X$ under $\psi$ is $\psi_\# \mathbb{P}_X(A) = \mathbb{P}_X(\psi^{-1}(A))$, where $\psi^{-1}(A)$ is the inverse image of $A \subset \mathbb{R}^p$. To reduce notational burden, we will refer to $\psi_\# \mathbb{P}_X$ as $\mathbb{P}_{\psi(X)}$.

**Reproducing Kernel Hilbert Spaces**: Each positive definite kernel function $k : \mathcal{X} \times \mathcal{X} \to \mathbb{R}$ induces a vector space of functions from $\mathcal{X}$ to $\mathbb{R}$, known as a RKHS (Aronszajn, 1950). Given an additional random variable $Y : \Omega \to \mathbb{R}^d$, one can consider the *Maximum Mean Discrepancy* (MMD) (Gretton et al., 2012), a kernel-based distance between probability measures:

$$\text{MMD}^2(\mathbb{P}_X, \mathbb{P}_{X'}) = \mathbb{E})_{X,X'}[k(X, X')] - 2\,\mathbb{E}_{X,Y}[k(X, Y)] + \mathbb{E}_{Y,Y'}[k(Y, Y')],$$

where $X, X' \sim \mathbb{P}_X$ and $Y, Y' \sim \mathbb{P}_Y$. For a broad class of *characteristic* kernels—including standard choices such as Gaussian, Laplacian, and Matérn kernels (Sriperumbudur et al., 2011)—we have $\mathrm{MMD}(\mathbb{P}_X, \mathbb{P}_Y) = 0$ if and only if $\mathbb{P}_X = \mathbb{P}_Y$ (Gretton et al., 2012).

**Related Work**: Dataset condensation produces a compressed set which attempts to replicate the performance of the original data on a downstream task, typically image classification (Sachdeva & McAuley, 2023). The closest method to ours is *M3D* (Zhang et al., 2024), which uses a distribution-matching strategy, optimising images in *ambient space*, via gradient descent on the MMD computed in the latent space of randomly initialised encoders (see Section A.1.2). However, this method gives no guarantee of preserving the original distribution, and is restricted to classification tasks. Unlike M3D, BDC reduces both sample size *and* dimensionality, targeting distributional preservation, with accompanying theoretical guarantees. BDC is thus *task-agnostic*: the compressed set could be reused across diverse downstream applications. Moreover, BDC is domain-agnostic (applicable wherever a positive definite kernels may be defined), leverages both features and responses during learning, and directly supports a wide range of downstream tasks in latent space beyond just image classification. Section A.1 also reviews (non-bilateral) distribution compression methods.

## 3 BILATERAL DISTRIBUTION COMPRESSION

Given a dataset $\mathcal{D} := \{\boldsymbol{x}_i\}_{i=1}^n \subset \mathbb{R}^d$ with $n, d \gg 1$, sampled i.i.d. from $\mathbb{P}_X$, we wish to construct a compressed representation $\mathcal{C} := \{\boldsymbol{z}_j\}_{j=1}^m \subset \mathbb{R}^p$ with $m \ll n$ and $p \ll d$, such that the essential properties of the original dataset are preserved.

We first need to define a notion of discrepancy between the empirical distribution $\hat{\mathbb{P}}_X$ in ambient space, and the empirical distribution of the compressed set $\hat{\mathbb{P}}_Z$ in latent space. To this end, we propose the *Decoded Maximum Mean Discrepancy* (DMMD):

$$\mathrm{DMMD}^2(\hat{\mathbb{P}}_X, \hat{\mathbb{P}}_{\phi(Z)}) := \sum_{i,j=1}^n k(\boldsymbol{x}_i, \boldsymbol{x}_j) - 2 \sum_{i,j=1}^{n,m} k(\boldsymbol{x}_i, \phi(\boldsymbol{z}_j)) + \sum_{i,j=1}^m k(\phi(\boldsymbol{z}_i), \phi(\boldsymbol{z}_j)) \quad (1)$$

where $\phi : \mathbb{R}^p \to \mathbb{R}^d$ is a *decoder* that maps latent points back into ambient space, and $k : \mathbb{R}^d \times \mathbb{R}^d \to \mathbb{R}$ is a positive definite kernel.

The DMMD is attractive as it measures how much distributional information is lost between the compressed set in latent space and the original ambient distribution. Despite this appeal, the DMMD has two practical limitations:

**PL1:** Joint optimisation over both the decoder and the compressed set is challenging: their roles are tightly coupled within the DMMD objective, leading to a highly non-convex and entangled optimisation landscape.

**PL2:** If the decoder is overly flexible, it may memorise the target distribution, giving strong DMMD even if the latent space is uninformative. As the aim is to perform compression in latent space and then use this set for downstream tasks, it is crucial that the latent representation itself remains meaningful. DMMD alone would not detect such degeneration.

To address these issues, we adopt a two-stage optimisation procedure and show that, with appropriate regularisation, our approach effectively mitigates these practical limitations.

### 3.1 ADDRESSING **PL1** VIA TWO-STAGE OPTIMISATION

To address **PL1**, the first step is to decouple the optimisation of the decoder $\phi$ from that of the compressed set $\mathcal{C}$.

### 3.1.1 LATENT SPACE OPTIMISATION OBJECTIVE

To optimise an informative latent space, we first require a suitable loss function. For this purpose, we define the *Reconstruction Maximum Mean Discrepancy* (RMMD) as

$$\mathrm{RMMD}^2(\hat{\mathbb{P}}_X, \hat{\mathbb{P}}_{\phi(\psi(X))}) := \sum_{i,j=1}^n k(\boldsymbol{x}_i, \boldsymbol{x}_j) - 2 \sum_{i,j=1}^n k(\boldsymbol{x}_i, \tilde{\boldsymbol{x}}_j) + \sum_{i,j=1}^n k(\tilde{\boldsymbol{x}}_i, \tilde{\boldsymbol{x}}_j), \quad (2)$$

where $\tilde{\boldsymbol{x}}_i := \phi(\psi(\boldsymbol{x}_i))$. Here, $\psi : \mathbb{R}^d \to \mathbb{R}^p$ is an *encoder* mapping data from ambient to latent space, and $\phi \circ \psi : \mathbb{R}^d \to \mathbb{R}^d$ is the corresponding *autoencoder*, which projects data into latent space and reconstructs it back into ambient space. A low RMMD indicates that the autoencoder is capable of reconstructing the original distribution accurately.

### 3.1.2 MOTIVATING THE RMMD

Autoencoders are commonly trained by minimising the *Mean Squared Reconstruction Error* (MSRE). To motivate the use of RMMD instead, we compare against *Principal Component Analysis* (PCA). PCA finds a $p$-dimensional subspace which solves

$$V_*^{\mathrm{MSRE}} = \underset{V \in \mathbb{R}^{d \times p}, \, V^\top V = I_p}{\arg\min} \ \mathrm{MSRE}(\mathcal{D}), \quad \mathrm{MSRE}(\mathcal{D}) := \frac{1}{nd} \sum_{i=1}^{n} \left\| \boldsymbol{x}_i - \boldsymbol{x}_i V V^\top \right\|_2^2. \quad (3)$$

The optimal $V_*^{\mathrm{MSRE}}$ consists of the top $p$ eigenvectors of the sample covariance matrix (Maćkiewicz & Ratajczak, 1993). While PCA minimises MSRE, it does not necessarily minimise RMMD. The key difference is that PCA aligns reconstructed points with the originals in Euclidean distance, whereas RMMD compares the entire distribution. As a result, PCA captures high-variance (second-order) directions without preserving higher-order moments. This motivates replacing the MSRE in (3) with the RMMD. The next result clarifies how the choice of kernel $k(\cdot, \cdot)$ shapes the outcome:

**Theorem 3.1** *Assume the distribution $\mathbb{P}_X$ has zero mean, and let the kernel $k : \mathbb{R}^d \times \mathbb{R}^d \to \mathbb{R}$ be the quadratic kernel defined by $k(\boldsymbol{x}, \boldsymbol{y}) = (1 + \boldsymbol{x}^\top \boldsymbol{y})^2$. Then $V_*^{RMMD}$ is given by (a permutation of) the top $p$ eigenvectors of the sample covariance matrix.*

**Remark 3.2** *Theorem 3.1 says that minimising RMMD with a quadratic kernel identifies the same subspace as PCA. This is intuitive: PCA assumes zero-mean data and identifies directions of maximal variance, while the MMD between two zero-mean distributions under the polynomial kernel is sensitive only to differences in second-order moments (Sriperumbudur et al., 2010). Hence, choosing a characteristic kernel makes RMMD sensitive to all distributional moments, yielding an autoencoder that preserves more distributional information than PCA's second-order moment matching.*

### 3.1.3 LATENT SPACE COMPRESSION OBJECTIVE

Given an encoder $\psi$ mapping into the latent space, we require an objective function to optimise the compressed set. For this purpose, we define the *Encoded Maximum Mean Discrepancy* (EMMD) as

$$\mathrm{EMMD}(\hat{\mathbb{P}}_{\psi(X)}, \hat{\mathbb{P}}_Z) := \sum_{i,j=1}^{n} h(\psi(\boldsymbol{x}_i), \psi(\boldsymbol{x}_j)) - 2 \sum_{i,j=1}^{n,m} h(\psi(\boldsymbol{x}_i), \boldsymbol{z}_j) + \sum_{i,j=1}^{m} h(\boldsymbol{z}_i, \boldsymbol{z}_j), \quad (4)$$

where $h : \mathbb{R}^p \times \mathbb{R}^p \to \mathbb{R}$ is a positive definite kernel on the latent space. This objective quantifies how well the distribution of the compressed set $\hat{\mathbb{P}}_Z$ matches the encoded dataset distribution $\hat{\mathbb{P}}_{\psi(X)}$.

### 3.1.4 TWO-STAGE OPTIMISATION PROCEDURE

Having identified natural objectives for optimising the autoencoder $\phi \circ \psi$ and the compressed set $\mathcal{C}$, we propose the following two-stage procedure, and show it minimises the DMMD:

**Step 1**: Optimise the autoencoder $\phi \circ \psi$ to minimise the RMMD. Letting the encoder and decoder be parameterised by $\boldsymbol{\alpha}$ and $\boldsymbol{\beta}$ respectively, we solve

$$\boldsymbol{\alpha}_*, \boldsymbol{\beta}_* = \underset{\boldsymbol{\alpha}, \boldsymbol{\beta} \in \Theta}{\arg\min} \ \mathrm{RMMD}^2(\hat{\mathbb{P}}_X, \hat{\mathbb{P}}_{\phi_{\boldsymbol{\beta}}(\psi_{\boldsymbol{\alpha}}(X))})$$

via minibatch gradient descent on $\boldsymbol{\alpha}$ and $\boldsymbol{\beta}$.

**Step 2**: Given the optimal encoder $\psi_{\boldsymbol{\alpha}_*}$, optimise the compressed set $\mathcal{C} \subset \mathbb{R}^p$ to minimise the EMMD. That is, we solve

$$\mathcal{C}_* = \underset{\mathcal{C} \subset \mathbb{R}^p}{\arg\min} \ \mathrm{EMMD}^2(\hat{\mathbb{P}}_{\psi_{\boldsymbol{\alpha}_*}(X)}, \hat{\mathbb{P}}_Z)$$

via gradient descent on $\mathcal{C}$.[1]

---

[1]This step can be analysed as a discretised Wasserstein gradient flow and admits global convergence guarantees under certain convexity assumptions on the objective (Arbel et al., 2019).

The overall time and memory cost of the two-stage procedure scales *linearly* with both the size and the dimensionality of the input dataset (see Section C.4). As our goal is to minimise the DMMD, the following provides a sufficient condition under which this is achieved by the above method:

**Theorem 3.3** *Let $k : \mathbb{R}^d \times \mathbb{R}^d \to \mathbb{R}$ and $h : \mathbb{R}^p \times \mathbb{R}^p \to \mathbb{R}$ be characteristic kernels, and let $\psi : \mathbb{R}^d \to \mathbb{R}^p$, $\phi : \mathbb{R}^p \to \mathbb{R}^d$ be measurable. Suppose $\mathbb{P}_X$ and $\mathbb{P}_Z$ are distributions such that RMMD $\left(\mathbb{P}_X, \mathbb{P}_{\phi(\psi(X))}\right) = 0$ and $\left(\mathbb{P}_{\psi(X)}, \mathbb{P}_Z\right) = 0$, then DMMD $\left(\mathbb{P}_X, \mathbb{P}_{\phi(Z)}\right) = 0$.*

**Remark 3.4** *As MMD estimates converge to their population values as $n \to \infty$ (Muandet et al., 2017), the following empirical analogue of Theorem 3.3 holds. If it were the case that for a fixed autoencoder $\phi \circ \psi$ and compressed set $\mathcal{C}$, $\text{RMMD}(\hat{\mathbb{P}}_X, \hat{\mathbb{P}}_{\phi(\psi(X))}) \to 0$ and $\text{EMMD}(\hat{\mathbb{P}}_{\psi(X)}, \hat{\mathbb{P}}_Z) \to 0$ as $n \to \infty$, then Theorem 3.3 implies that $\text{DMMD}(\hat{\mathbb{P}}_X, \hat{\mathbb{P}}_{\phi(Z)}) \to 0$ as $n \to \infty$.*

We can also give a bound that holds without requiring exact matching:

**Theorem 3.5** *Let $k : \mathbb{R}^d \times \mathbb{R}^d \to \mathbb{R}$ be a positive definite kernel, and let $\psi : \mathbb{R}^d \to \mathbb{R}^p$, $\phi : \mathbb{R}^p \to \mathbb{R}^d$ be measurable. Given the pull-back kernel $h : \mathbb{R}^p \times \mathbb{R}^p \to \mathbb{R}$, $h(\boldsymbol{z}, \boldsymbol{z}') = k\left(\phi(\boldsymbol{z}), \phi(\boldsymbol{z}')\right)$, we have that, for any probability measures $\mathbb{P}_X$ and $\mathbb{P}_Z$,*

$$\text{DMMD}\left(\mathbb{P}_X, \mathbb{P}_{\phi(Z)}\right) \leq \text{RMMD}\left(\mathbb{P}_X, \mathbb{P}_{\phi(\psi(X))}\right) + \text{EMMD}\left(\mathbb{P}_{\psi(X)}, \mathbb{P}_Z\right).$$

**Remark 3.6** *The kernel $h$ is commonly referred to as the pull-back of $k$ through $\phi$ (Paulsen & Raghupathi, 2016). Note that if $k$ is characteristic, then $h$ is clearly characteristic modulo the equivalence relation $\mathbb{P}_Z \sim \mathbb{Q}_Z \iff \mathbb{P}_{\phi(Z)} = \mathbb{Q}_{\phi(Z)}$, that is, it distinguishes latent distributions up to those that decode to the same ambient distribution.*

Theorem 3.3 provides an ideal-case guarantee: if both RMMD and EMMD are reduced to zero, then DMMD is also exactly zero. Theorem 3.5 instead bounds DMMD by the sum of RMMD and EMMD. In practice, Theorem 3.3 is sufficient to justify the two-stage procedure, while Theorem 3.5 offers an explicit bound on DMMD if required, at the cost of prescribing the latent-space kernel.

### 3.2 ADDRESSING **PL2** VIA REGULARISATION

We now turn to concrete autoencoder instantiations and discuss how regularisation addresses **PL2**.

#### 3.2.1 LINEAR AUTOENCODER

As discussed in Section 3.1.2, restricting the autoencoder to reconstruct data within a $p$-dimensional linear subspace of $\mathbb{R}^d$ inherently limits its expressive capacity. The orthonormality constraint on the encoder further enforces that $p$ orthogonal directions are preserved, ensuring that the latent representation retains meaningful structure. Note that we enforce the orthonormality constraint during optimisation via minibatch gradient descent on the *Stiefel manifold* (Absil et al., 2007) (see Section C.1 for details). Collectively, these constraints act as an effective form of regularisation, preventing the encoder from arbitrarily discarding information.

#### 3.2.2 NONLINEAR AUTOENCODERS

Neural autoencoders were originally introduced for nonlinear dimensionality reduction (Hinton & Salakhutdinov, 2006). Compared to linear methods, they often achieve superior performance on downstream tasks by exploiting greater flexibility to learn complex nonlinear patterns in data (Fournier & Aloise, 2019). However, excessive flexibility can produce poor latent representations. In the extreme, the network may approximate the identity map, yielding latents that are uninformative for downstream use. Regularisation is therefore essential. Common strategies include enforcing a bottleneck by restricting the size of hidden layers and the final latent dimension, tying encoder and decoder weights, or limiting decoder capacity, each of which reduces the risk of memorisation and encourages the latent space to capture meaningful structure rather than simply copying the input (Goodfellow et al., 2016).

The RMMD is attractive as, with characteristic kernels, it is sensitive to distributional differences, whereas MSRE only measures pointwise error. However, RMMD requires only that the reconstructed distribution match the data in aggregate, so a flexible neural autoencoder may reconstruct

individual samples at different locations in the distribution than their inputs. The MSRE term mitigates this by encouraging the autoencoder to approximate a homeomorphism, preserving topology and preventing mixing; see Figure 2. Thus, for neural autoencoders we adopt a hybrid objective:

$$\text{RMMD}_k^2 \left( \hat{\mathbb{P}}_X, \hat{\mathbb{P}}_{\phi(\psi(X))} \right) + \frac{1}{nd} \sum_{i=1}^{n} \| \boldsymbol{x}_i - \phi(\psi(\boldsymbol{x}_i)) \|_2^2. \quad (5)$$

See Section A.2.1 for additional discussion, including a toy example where RMMD alone can fail to detect loss of cluster structure.

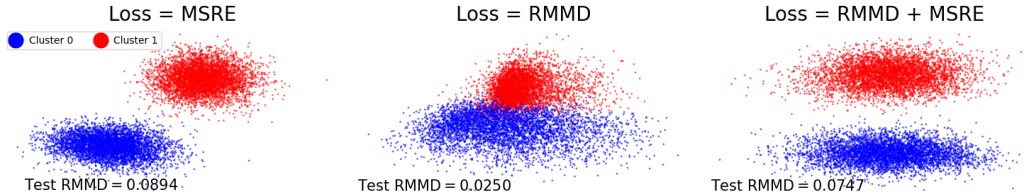

Figure 2: Latent representations of a dataset constructed by sampling two well-separated clusters from a 50-dimensional Gaussian with identity covariance. Training with MSRE preserves cluster separation but misaligns reconstructed distributions (higher test RMMD), RMMD aligns distributions but loses cluster separation, while RMMD+MSRE balances both.

## 3.3 EXTENSION TO LABELLED DATA

So far we have assumed that the target dataset is unlabelled. In many applications, however, responses $y \in \mathcal{Y}$ are also available, so that $\mathcal{D} = \{(\boldsymbol{x}_i, \boldsymbol{y}_i)\}_{i=1}^{n} \subset \mathbb{R}^d \times \mathcal{Y}$. BDC extends naturally to this supervised setting. Let $l : \mathcal{Y} \times \mathcal{Y} \to \mathbb{R}$ be a kernel on the response space and let $W : \Omega \to \mathcal{Y}$ denote the random variable corresponding to the compressed responses. It is then straightforward to define supervised analogues of our MMD objectives,

$$\text{DMMD} \left( \hat{\mathbb{P}}_{X,Y}, \hat{\mathbb{P}}_{\phi(Z),W} \right), \quad \text{RMMD} \left( \hat{\mathbb{P}}_{X,Y}, \hat{\mathbb{P}}_{\phi(\psi(X)),Y} \right), \quad \text{EMMD} \left( \hat{\mathbb{P}}_{\psi(X),Y}, \hat{\mathbb{P}}_{Z,W} \right),$$

which use *tensor product kernels* $r((\boldsymbol{x}, \boldsymbol{y}), (\boldsymbol{x}', \boldsymbol{y}')) := k(\boldsymbol{x}, \boldsymbol{x}') \, l(\boldsymbol{y}, \boldsymbol{y}')$ and $q((\boldsymbol{z}, \boldsymbol{y}), (\boldsymbol{z}', \boldsymbol{y}')) := h(\boldsymbol{z}, \boldsymbol{z}') \, l(\boldsymbol{y}, \boldsymbol{y}')$. For full technical details see Section C.3.

In this way, responses are incorporated directly into autoencoder training, ensuring that feature–response dependencies are preserved. Moreover, in the compressed set $\mathcal{C} = \{(\boldsymbol{z}_j, \boldsymbol{w}_j)\}_{j=1}^{m}$ we jointly optimise both the latent codes $\boldsymbol{z}_j$ and their paired responses $\boldsymbol{w}_j$, so that compression is now performed on input–output pairs rather than on features alone.

## 4 EXPERIMENTS

We evaluate BDC on real and synthetic datasets across both labelled and unlabelled tasks. As ambient-space baselines, we consider: (i) uniform random subsampling (URS); (ii) ambient-space distribution compression via gradient descent (ADC) (Arbel et al., 2019; Broadbent et al., 2025); and (iii) the full dataset (FULL), where computationally feasible. Within BDC, we compare two variants: BDC-L, which employs a linear autoencoder trained on the Stiefel manifold, and BDC-NL, which employs a nonlinear neural autoencoder. For image classification, we additionally compare against M3D, as discussed in the related work. Unless otherwise stated, we define kernels directly on the learned latent spaces, without employing the pull-back kernel. For further details and additional experiments, including ablation studies on latent dimensionality and kernel choice, see Section D.

## 4.1 EXACT BILATERAL DISTRIBUTION COMPRESSION

In general, the EMMD, RMMD, and DMMD must be estimated, but for certain kernel–distribution pairs analytical formulae are available (Briol et al., 2025). To demonstrate that BDC effectively minimises the DMMD, we generate $n = 20{,}000$ samples from a two-dimensional mixture of Gaussians

| Dataset | Method | | | | |
|---------|--------|------|------|------|------|
| | FULL | M3D | ADC | BDC-NL (**Ours**) | BDC-L (**Ours**) |
| *Swiss-Roll* | 944.1 | NA | $197.65 \pm 1.88$ | $9.54 \pm 0.86$ | $8.59 \pm 0.23$ |
| *CT-Slice* | NA | NA | $776.86 \pm 30.47$ | $117.81 \pm 4.31$ | $57.10 \pm 1.90$ |
| *MNIST* | NA | $827.34 \pm 27.99$ | $1003.62 \pm 1.80$ | $206.65 \pm 3.01$ | $72.26 \pm 2.29$ |
| *Clusters* | 1800.62 | NA | $691.00 \pm 26.50$ | $84.91 \pm 2.84$ | $46.74 \pm 1.82$ |

Table 1: Overall time comparison (construction of the compressed set plus downstream task fitting) in seconds, across datasets, reported as mean and standard deviation.

and project them to $d = 250$ dimensions using a matrix $V \in \mathbb{R}^{250 \times 2}$ of standard Gaussian draws. With a Gaussian kernel, the EMMD, RMMD, and DMMD can then be evaluated exactly (see Section D.2). We apply BDC to this 250-dimensional dataset using a linear autoencoder with intrinsic dimension $p = 2$; results are shown in Figure 4. The figure shows that the BDC-compressed set, when decoded into ambient space, achieves MMD which is equal to that achieved by the full dataset, while requiring only $0.018\%$ of the storage, at the largest size. We also empirically verify the bound from Theorem 3.5 as the pull-back kernel is used here.

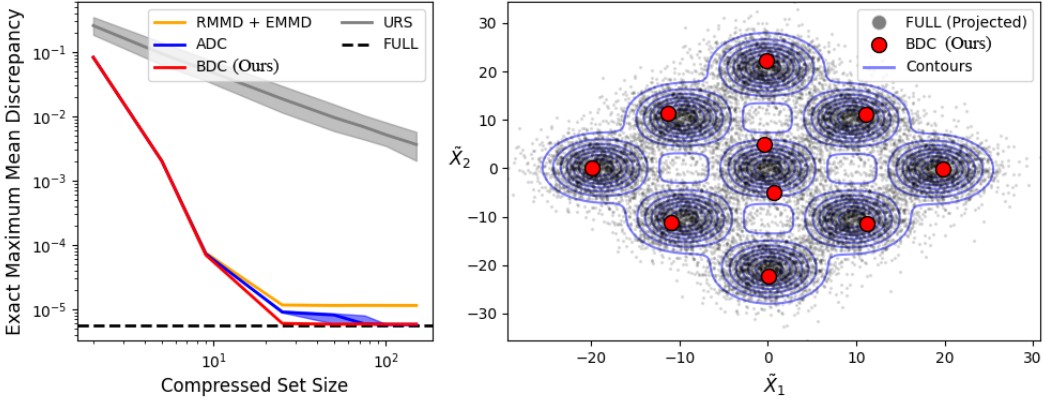

Figure 4: BDC on a Gaussian mixture dataset. Left: ambient space MMD achieved by BDC (red) and ADC (blue), with the RMMD+EMMD bound (orange), each reported over 5 runs. Compared with URS over 250 runs (shaded bands: 25th–75th percentiles), with FULL as baseline (black). Right: Full dataset projected into latent space (black), with BDC compressed set of size 10 (red).

### 4.2 MANIFOLD UNCERTAINTY QUANTIFICATION

Many datasets lie in high-dimensional spaces, yet the relationship between features and responses often varies along a low-dimensional manifold $\mathcal{M} \subset \mathbb{R}^d$ (Whiteley et al., 2025). To assess how well BDC preserves this structure, we evaluate using Gaussian Processes, as they capture not only predictive accuracy but also calibrated uncertainty. We construct compressed sets, train a GP on each set—in latent space for BDC and in ambient space for URS, ADC, and FULL—and report both the total runtime of the procedure and the predictive metrics on held-out data.

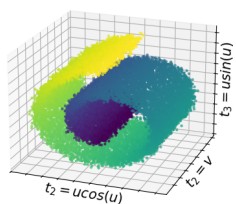

Figure 3: Swiss-roll manifold, coloured with value of $u$.

#### 4.2.1 REGRESSION

**Swiss-Roll**: We first consider a synthetic dataset lying on the Swiss-roll manifold embedded in $\mathbb{R}^{200}$; see Figure 3 (Yang & Dunson, 2014). We generate $n = 20{,}000$ samples $\boldsymbol{x} = (t_1, t_2, t_3)^\top$ by

$$t_1 = u\cos(u), \quad t_2 = v, \quad t_3 = u\sin(u), \quad u \sim \mathcal{U}(\tfrac{3\pi}{2}, \tfrac{9\pi}{2}), \quad v \sim \mathcal{U}(0, 20)$$

Gaussian noise $\mathcal{N}(\mathbf{0}, I_3)$ is then added to $\boldsymbol{x}$. The data are nonlinearly projected to ambient space via a randomly initialised 2-layer feed-forward neural net. Responses are defined as

$$y = f(u, v) + \epsilon, \quad f(u, v) = 4 \left( \frac{u}{3\pi} - \frac{1+3\pi}{2} \right)^2 + \frac{\pi}{20} v$$

We compress to $m = 200$ pairs with latent dimension $p = 3$ (99.99% compression). Figure 5 and Table 1 show that BDC-NL consistently outperforms ADC across all metrics on $n_t = 1,000$ test pairs, while being 20× faster. As expected, BDC-L performs poorly due to the misspecification of a linear autoencoder in this nonlinear setting.

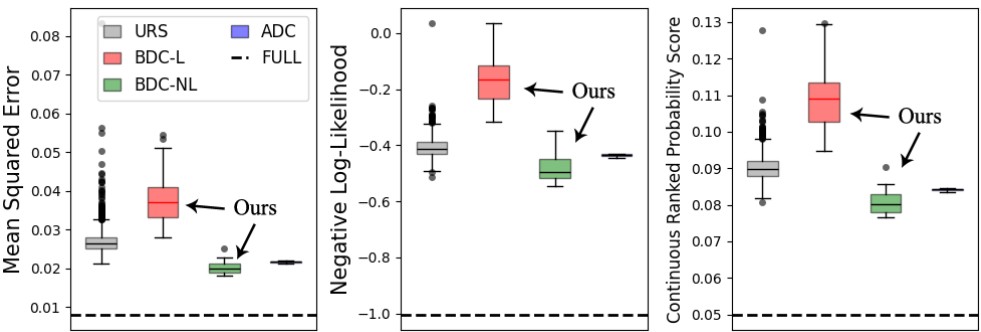

Figure 5: Test performance on *Swiss-Roll*. BDC-L (red), BDC-NL (green), and ADC (blue) each reported over 25 runs, URS (grey) over 1000 runs, and FULL is shown as a black dashed line.

**CT Slice**: The *CT-Slice* dataset (Graf et al., 2011) contains $n = 53,500$ instances with $d = 384$ features derived from CT scans, with the task of predicting slice position along the body. We hold out $n_t = 3,500$ pairs for computing test performance. We compress to $m = 200$ pairs with latent dimension $p = 27$ (99.97% compression). Figure 6 and Table 1 show that BDC outperforms ADC in both predictive performance *and* speed.

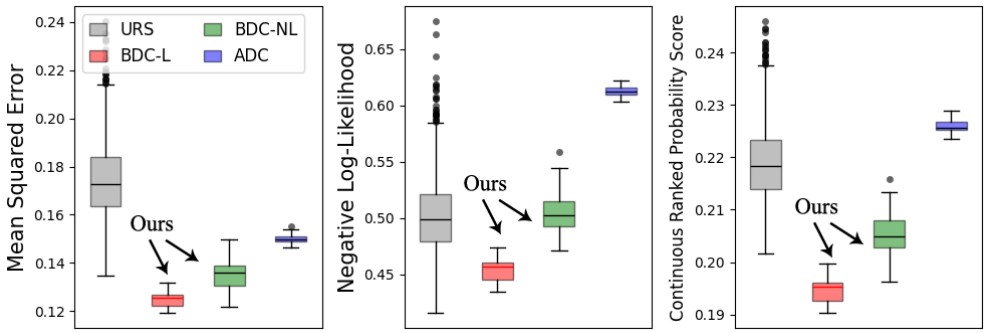

Figure 6: Test performance on *CT-Slice*. BDC-L (red), BDC-NL (green), and ADC (blue) each reported over 25 runs, URS (grey) over 1000 runs.

### 4.2.2 CLASSIFICATION

**MNIST**: The *MNIST* dataset (LeCun & Cortes, 2010) contains $n = 70,000$ handwritten digits from 0-9 in $d = 784$ dimensions, with the task of predicting the digit. We hold out the last $n_t = 10,000$ pairs for computing test performance. We compress to $m = 200$ pairs with latent dimension $p = 28$ (99.99% compression). As this is an image classification task, we can compare against M3D (Zhang et al., 2024), as discussed in the related work; to enable a fair comparison to ADC, we do not employ the up-sampling technique of Zhao et al. (2023); Zhang et al. (2024). In principle, this strategy could be applied to ADC, and its impact is a potentially interesting avenue for future investigation. Figure 7 and Table 1 shows that BDC outperforms ADC and M3D in both speed *and* predictive performance.

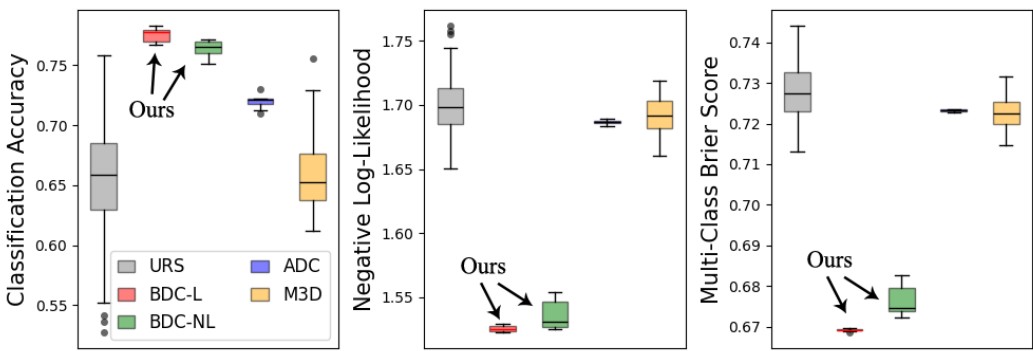

Figure 7: Test performance on *MNIST*. BDC-L (red), BDC-NL (green), and ADC (blue) each reported over 10 runs, URS (grey) over 200 runs.

### 4.3 CLUSTERING

**Clusters**: We construct a synthetic dataset of $n = 100,000$ two-dimensional points arranged in 10 clusters of varying shapes, with an additional 2,000 points of uniformly distributed noise (see Figure 8b). The data are then projected nonlinearly to $d = 500$ dimensions using a four-layer decoder with Tanh activations. To recover the clusters, we apply HDBSCAN (Campello et al., 2013), which can detect non-elliptical structures while leaving uncertain points—such as noise—unclustered. Ground-truth labels are used only for evaluation, where we use an additional $n_t = 20,400$ held-out points. We compress to $m = 300$ in latent spaces of $p = 2$ (BDC-NL, $99.999\%$ compression) and $p = 5$ (BDC-L, $99.997\%$). Both BDC-L and BDC-NL are misspecified, with BDC-NL restricted to a one-hidden-layer ReLU autoencoder. Figure 8a and Table 1 show that BDC achieves cluster recovery performance comparable to ADC, while requiring substantially less runtime. Figure 8b also illustrates the original intrinsic data alongside the compressed set constructed by BDC-NL.

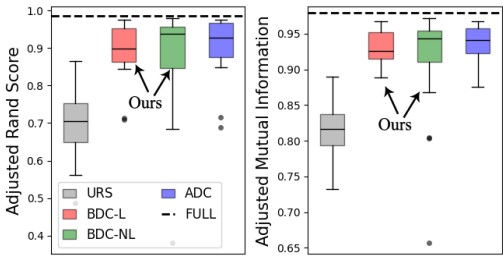
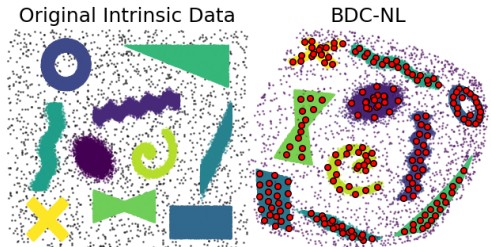

(a) Test performance on *Clusters*. BDC-L (red), BDC-NL (green), and ADC (blue) each reported over 25 runs, URS (grey) over 100 runs, and FULL is shown as a black dashed line.

(b) Original intrinsic data before projection, next to a compressed set constructed by BDC-NL. The encoder recovers the intrinsic space and the compressed set clearly delineates the clusters, ignoring noise.

Figure 8: Results on the *Clusters* dataset.

## 5 CONCLUSIONS

We introduced the problem of *bilateral distribution compression*, which seeks to reduce both the sample size and feature dimensionality of a dataset while targeting distributional preservation. We proposed a tractable linear-time two-stage framework targeting the DMMD. The first stage learns a low-dimensional representation by minimising the RMMD, and the second stage selects a compressed set in latent space by minimising the EMMD. We established theoretical guarantees linking these quantities to the DMMD, and demonstrated empirically that the resulting compressed sets achieve comparable or superior performance across a range of tasks versus ambient-space distribution compression methods, with much reduced runtime. A detailed discussion of future work, limitations, and related work is given in Section A.

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

# Part I

# Appendix

## Table of Contents

## A  FURTHER DISCUSSION

In this section we include further discussions and conclusions that could not fit in the main body, including related work, applications, limitations and future work.

### A.1  RELATED WORK

In this section, we provide a comprehensive overview of related work, highlighting how the Bilateral Distribution Compression technique introduced in this paper is novel.

### A.1.1  STANDARD DISTRIBUTION COMPRESSION

As noted in the introduction, numerous distribution compression methods in the literature focus on reducing the *number of observations* in a dataset. Most of these are kernel-based methods that target the MMD, with one notable exception, which can be shown to be equivalent to a kernel-based method for a specific choice of kernel. To the best of our knowledge, no existing method performs *Bilateral Distribution Compression* as defined in this work. Nevertheless, we now briefly summarise the existing approaches.

**Kernel Herding** (Chen et al., 2010; Bach et al., 2012) constructs a compressed set by greedily minimising the MMD, optimising one point at a time. Let $\boldsymbol{X} := [\boldsymbol{x}_1, \boldsymbol{x}_2, \ldots, \boldsymbol{x}_n]^\top \subset \mathbb{R}^d$ be an i.i.d. sample from the target distribution $\mathbb{P}_X$, and let $\boldsymbol{Z} := [\boldsymbol{z}_1, \boldsymbol{z}_2, \ldots, \boldsymbol{z}_m]^\top \subset \mathbb{R}^d$ denote the current compressed set. At each iteration, the next point is chosen by solving

$$\boldsymbol{z}_{m+1} = \arg\min_{\boldsymbol{z} \in \mathbb{R}^d} \ \frac{1}{m+1} \sum_{j=1}^{m} k(\boldsymbol{z}, \boldsymbol{z}_j) - \mathbb{E}_{\boldsymbol{x}' \sim \mathbb{P}_X} \left[ k(\boldsymbol{z}, \boldsymbol{x}') \right],$$

after which the compressed set is updated to $\boldsymbol{Z} := [\boldsymbol{z}_1, \boldsymbol{z}_2, \ldots, \boldsymbol{z}_m, \boldsymbol{z}_{m+1}]^\top$, and the process is repeated. This greedy strategy has the advantage of focusing computational effort on optimising one new point at a time while leaving previously selected points fixed. However, it may yield suboptimal solutions, as earlier selections are never revisited or refined.

**Gradient Flow** (Arbel et al., 2019; Chen et al., 2025) methods address a similar objective by solving a discretised Wasserstein gradient flow of the MMD. In practice, this corresponds to performing

gradient descent on all points in the compressed set simultaneously, i.e., solving

$$\arg\min_{\boldsymbol{Z} \subset \mathbb{R}^d} \frac{1}{m^2} \sum_{i,j=1}^{m} k(\boldsymbol{z}_i, \boldsymbol{z}_j) - \frac{2}{m} \sum_{i=1}^{m} \mathbb{E}_{\boldsymbol{x}' \sim \mathbb{P}_X} \left[ k(\boldsymbol{z}_i, \boldsymbol{x}') \right].$$

Under certain technical convexity assumptions on the objective (Arbel et al., 2019), this method can be shown to converge to the global optimum; in practice, however, one should only expect to obtain a locally optimal solution. We adopt this approach for optimising the compressed set in latent space, as it has been shown to perform strongly relative to other methods (Chen et al., 2025). This method was recently extended to target the joint distribution in Broadbent et al. (2025).

**Kernel Thinning** (Dwivedi & Mackey, 2021a;b; Shetty et al., 2022) constructs a compressed set that is a proper subset of the original dataset, i.e., all points in the compressed set are drawn directly from the input data. This restriction stems from the method's original motivation—thinning the output of Markov Chain Monte Carlo (MCMC) methods—where subsampling is commonly referred to as standard thinning. Kernel thinning proceeds via a two-stage procedure targeting the MMD. First, the input dataset is probabilistically split into $2^m$ MMD-balanced candidate sets, each of size $\lfloor n/2^m \rfloor$, discarding the remaining $n - 2^m \lfloor n/2^m \rfloor$ points if $n$ is not evenly divisible. Second, the best candidate set from this partitioning is selected, and then greedily refined by iteratively replacing points with others from the original dataset whenever doing so improves the MMD. While this construction enables the derivation of convergence rates that are state of the art in the literature, its practical performance has been found to be inferior to alternative approaches; see, for example, Chen et al. (2025).

**Support Points** (Mak & Joseph, 2018) is a method for constructing compressed sets that takes a similar optimisation-based approach to Gradient Flow methods, but targets the *Energy Distance* (ED) rather than the MMD. The ED between distributions $\mathbb{P}_X$ and $\mathbb{Q}_X$ is defined as

$$\text{ED}(\mathbb{P}_X, \mathbb{Q}_X) := 2\,\mathbb{E}[\|\boldsymbol{a} - \boldsymbol{b}\|_2] - \mathbb{E}[\|\boldsymbol{a} - \boldsymbol{a}'\|_2] - \mathbb{E}[\|\boldsymbol{b} - \boldsymbol{b}'\|_2],$$

where $\boldsymbol{a}, \boldsymbol{a}' \sim \mathbb{P}_X$ and $\boldsymbol{b}, \boldsymbol{b}' \sim \mathbb{Q}_X$. Much like the MMD, the energy distance is zero if and only if $\mathbb{P}_X = \mathbb{Q}_X$ (Mak & Joseph, 2018, Theorem 1). Although Support Points is not initially expressed in kernel form, the energy distance is in fact equivalent to the MMD for a particular choice of negative-definite kernel (Sejdinovic et al., 2013). In practice, Support Points is generally outperformed by kernel-based methods, and in particular, the Gradient Flow approach that we adopt here (Chen et al., 2025).

### A.1.2 DATASET CONDENSATION

In the dataset condensation literature, considerable attention has been given to optimising a small set of synthetic images that act as a substitute for the full training dataset. The goal is to reduce training cost while preserving predictive performance for a specific model or task, assessed through classification accuracy.

We now briefly review the approaches most relevant to our work and highlight the important distinctions with our framework.

**Dataset Condensation via Distribution Matching**: Recently, Zhao & Bilen (2022) proposed a *distribution-matching* approach based on randomly initialised neural encoders. Given a task and dataset, a neural architecture $\psi_{\boldsymbol{\theta}}$ is first selected, and a compressed set of synthetic images $\mathcal{C}$ is randomly initialised. The encoder weights are sampled from a distribution $\mathbb{P}_{\boldsymbol{\theta}}$. For each image class, batches from the original dataset $\mathcal{D} = \{\boldsymbol{x}_i\}_{i=1}^{n}$ and the compressed set $\mathcal{C} = \{\boldsymbol{z}_i\}_{i=1}^{m}$ are then embedded into the latent space of this randomly initialised network. The compressed set is subsequently optimised for a few iterations by gradient descent, targeting the alignment between the latent embeddings:

$$\left\| \frac{1}{n} \sum_{i=1}^{n} \psi_{\boldsymbol{\theta}}(\boldsymbol{x}_i) - \frac{1}{m} \sum_{j=1}^{m} \psi_{\boldsymbol{\theta}}(\boldsymbol{z}_j) \right\|_2, \tag{6}$$

and the procedure is repeated with fresh random initialisations.

In words, this method exploits the latent space of randomly initialised CNNs to optimise a compressed set of synthetic images in the original input space. Empirically, training a CNN on such a

compressed set achieves strong classification accuracy (Zhao & Bilen, 2022). It is important to note, however, that the objective in (6) is *not* an MMD: it compares only the first moment of the embedded distributions, and therefore does not measure distributional equality. A subsequent refinement of this approach by Zhao et al. (2023) incorporated augmentation, sampling, and regularisation techniques, though the underlying methodology remained unchanged.

Zhang et al. (2024) advanced this line of work by introducing *M3D*, which replaces the first-order objective in (6) with a genuine MMD, defined using a kernel $h$ on the latent space. Specifically, they minimise

$$\frac{1}{m} \sum_{i,j=1}^{m} h\left(\psi_{\boldsymbol{\theta}}(\boldsymbol{z}_i), \psi_{\boldsymbol{\theta}}(\boldsymbol{z}_j)\right) - \frac{2}{m} \sum_{i=1}^{n} \sum_{j=1}^{m} h\left(\psi_{\boldsymbol{\theta}}(\boldsymbol{x}_i), \psi_{\boldsymbol{\theta}}(\boldsymbol{z}_j)\right).$$

They report improved performance, attributing it to the kernel's ability to capture discrepancies across all moments of the distribution, rather than only the first.

Despite these variations, the above methods share the same result: they optimise a compressed set directly in ambient image space and provide no guarantee that the original distribution has been faithfully preserved. By contrast, our approach simultaneously reduces both the size and dimensionality of the dataset, and Theorems 3.3 and 3.5 establish that it achieves compression at the level of the ambient distribution.

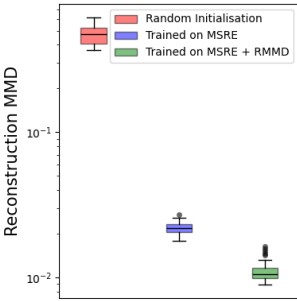

Figure 9: Test reconstruction MMD on 01-MNIST encoded into two dimensions. A randomly initialised autoencoder (red) exhibits high RMMD, suggesting poor preservation of the distributional structure. Training on MSRE alone (blue) reduces RMMD, but incorporating RMMD into the training objective (green) yields consistently lower values.

Even on a simple dataset such as 01-MNIST (LeCun & Cortes, 2010; LeCun et al., 1998), Figure 9 highlights that random latent representations, as employed in Zhao & Bilen (2022); Zhang et al. (2024), may fail to preserve distributional structure. In contrast, training with a combination of RMMD and MSRE (Equation 5) consistently achieves lower RMMD than using MSRE alone.

**Latent Space Generative Dataset Condensation**: Recent work has sought to enhance dataset condensation by exploiting the latent spaces of pre-trained generative models. Moser et al. (2024) introduce Latent Dataset Distillation with Diffusion Models (LD3M), which optimises the latent representations of synthetic images in the latent space of a diffusion model, while Cazenavette et al. (2023) apply the same idea to GANs.

Although these methods optimise latent codes, the codes themselves are never treated as objects of interest. Instead, they serve only as intermediates, relying solely on decoding back to images and measuring task-specific classification accuracy. Since the latent space is inherited from a pre-trained generator, applicability is limited to domains where such models exist—primarily natural images. The latent dimensionality is fixed by the generator architecture, preventing adaptation to the dataset or compression objective. Moreover, as noted by Cazenavette et al. (2023), large-scale generative models admit many possible latent representations, and test accuracy can vary substantially depending on which one is chosen. Consequently, both the size and nature of the latent representation are dictated by arbitrary design choices in the pre-trained model, rather than being chosen for distributional compression. Additionally, even when discounting the substantial cost of pre-training the generative model, these methods appear to be very slow. Moser et al. (2024) report training times

of roughly 10 hours on an A100-40GB GPU—hardware considerably more powerful than the GTX 4070 Ti 12GB GPU used in our experiments. Moreover, these methods provide no formal guarantees of distributional preservation.

By contrast, our method does not depend on a pre-trained generative model. Instead, it defines compression in terms of kernels, allowing application to arbitrary feature domains where a positive definite kernel can be specified. The informative latent representation is learned directly from the data, with dimensionality treated as a tunable parameter that can be tailored to the application, rather than inherited from an external architecture. This flexibility makes the approach inherently task-agnostic and domain-independent. Furthermore, Theorems 3.3 and 3.5 establish that our procedure achieves compression at the level of the distribution, not merely task-specific performance. In Sections 4 and D, we evaluate distributional fidelity under diverse experimental conditions, where like-for-like comparison to these generative approaches is not possible due to their reliance on pre-trained models, which either do not exist for the datasets considered, or have larger latent spaces.

### A.1.3 Domain Adaption Autoencoder

In Section 3.2.2 we augmented the standard MSRE with the RMMD to avoid the failure mode described in **PL2**. It is worth noting that in the autoencoder literature the MSRE is frequently modified, or even replaced altogether, to address specific objectives. Examples include VAEs, which endow autoencoders with a probabilistic generative model (Kingma & Welling, 2022); SAEs, which promote sparsity in the latent codes (Ng, 2011); and CAEs, which enhance robustness of the latent representation (Rifai et al., 2011). Similarly, the MMD has also been employed within autoencoders, for instance in domain adaptation (Lin et al., 2018), which we briefly review here.

Given a dataset of in-domain data $\mathcal{D}^{\text{in}} := \{\boldsymbol{x}_j^{\text{in}}\}_{j=1}^{n_{\text{in}}}$ and out-domain data $\mathcal{D}^{\text{out}} := \{\boldsymbol{y}_j\}_{j=1}^{n_{\text{out}}}$, the goal of Lin et al. (2018) is to learn a mapping $\boldsymbol{z} = \psi(\boldsymbol{x})$ such that the transformed data $\{\boldsymbol{z}_j^{\text{in}}\}_{j=1}^{n_{\text{in}}}$ and $\{\boldsymbol{z}_j^{\text{out}}\}_{j=1}^{n_{\text{out}}}$ are as similar as possible. To achieve this, they optimise the encoder targeting the MMD between the two:

$$\mathcal{D}_{\text{MMD}} = \frac{1}{n_{\text{in}}^2} \sum_{i,j=1}^{n_{\text{in}}} h(\boldsymbol{z}_i^{\text{in}}, \boldsymbol{z}_j^{\text{in}}) - \frac{2}{n_{\text{in}} n_{\text{out}}} \sum_{i,j=1}^{n_{\text{in}}, n_{\text{out}}} h(\boldsymbol{z}_i^{\text{in}}, \boldsymbol{z}_j^{\text{out}}) + \frac{1}{n_{\text{out}}^2} \sum_{i,j=1}^{n_{\text{out}}} h(\boldsymbol{z}_i^{\text{out}}, \boldsymbol{z}_j^{\text{out}})$$

for some kernel $h$ defined on the latent space. If one has access to more than two sources of data, this objective can be further generalised (Lin et al., 2018). The decoder $\phi$ is optimised targeting the mean squared decoding error between true and decoded observations, i.e.

$$\frac{1}{2} \left( \frac{1}{n_{\text{in}}} \sum_{i=1}^{n_{\text{in}}} \|\boldsymbol{x}_i^{\text{in}} - \phi(\psi(\boldsymbol{x}_i^{\text{in}})) + \frac{1}{n_{\text{out}}} \sum_{i=1}^{n_{\text{out}}} \|\boldsymbol{x}_i^{\text{out}} - \phi(\psi(\boldsymbol{x}_i^{\text{out}}))\|_2^2 \right).$$

They observe significantly improved empirical performance in *speaker verification* tasks, where it is common for datasets to come from very different distributions. We note that they use MMD only for training the encoder, however, because our objective is to minimise the overall difference in distribution between the original and reconstructed data, we also train the decoder according to the MMD, including a MSRE term to avoid mixing (**PL2**).

We also note here that the MMD has been successfully applied in Variational (Kingma & Welling, 2022) and Wasserstein (Tolstikhin et al., 2019) autoencoders as a regulariser, where the primary objective is not dimensionality reduction, but rather to learn a generative model of the data distribution.

### A.2 Limitations

In this section, we provide a detailed discussion of the limitations of our approach.

### A.2.1 Informative Latent Spaces

As stated in Section 3.2.2, neural autoencoders may outperform linear ones on downstream tasks by exploiting their flexibility to capture complex nonlinear patterns in data. However, they are inherently black-box models. It is often difficult to anticipate how architectural choices—such as the number and width of hidden layers, the choice of activation function, or the adoption of specialised

architectures like convolutional networks—will affect the resulting latent representation. The interplay between architecture, loss function, and latent structure is highly nontrivial, meaning that without significant expertise and extensive experimentation it is challenging to design autoencoders that reliably yield informative representations.

In Section 3.2.2, for flexible neural autoencoders we introduced a hybrid objective consisting of a sum of the RMMD and MSRE. The MSRE term can be interpreted as enforcing that the decoder approximately inverts the encoder. A continuous, invertible function with a continuous inverse is a homeomorphism; if the encoder is close to a homeomorphism, then the latent space preserves topological properties of the input distribution, such as connected components and holes. This perspective clarifies why RMMD alone may be insufficient for flexible neural autoencoders. Consider three input points $x_1, x_2, x_3$, where $x_1$ and $x_2$ are close and form one cluster, while $x_3$ lies far away as a separate cluster. Suppose the latent dimension equals the ambient dimension. In the first scenario, both encoder $\psi$ and decoder $\phi$ are identity maps, so that $z_i = \psi(x_i)$ and $x_i = \phi(z_i)$; the latent representation reproduces the clustering of the data. In the second scenario, the encoder swaps $x_2$ and $x_3$ in latent space while the decoder remains the identity. Here, $\phi$ is no longer the inverse of $\psi$, so the mapping is not a homeomorphism. Yet in both cases, the RMMD is zero, since the reconstructed distribution matches the input in aggregate, despite the clustering structure being distorted.

While our hybrid objective combining RMMD and MSRE has been shown empirically to perform well across diverse scenarios, it is only one way of encouraging informative latent representations. Other strategies offer complementary regularisation. Sparse autoencoders (Olshausen & Field, 1996) encourage parsimonious representations through sparsity-inducing penalties, while contractive autoencoders (Rifai et al., 2011) enforce robustness by penalising the sensitivity of the encoder to perturbations. Integrating such techniques into the BDC framework is a promising direction for future work, as different forms of regularisation may align more closely with specific data properties or downstream tasks.

It would also be valuable to investigate how other nonlinear dimensionality-reduction techniques interact with our framework. Methods such as Kernel PCA (Schölkopf et al., 1998), t-SNE (van der Maaten & Hinton, 2008), and UMAP (McInnes et al., 2020) each learn nonlinear embeddings via distinct objectives. One intriguing avenue would be to replace or augment their loss functions with a distributional term such as RMMD, potentially yielding representations that are easier to reason about than those obtained from neural autoencoders.

### A.2.2 MANIFOLD STRUCTURE

BDC is most effective when the data distribution lies near a low-dimensional manifold embedded in ambient space. This assumption, known as the *Manifold Hypothesis*, is well-known in the literature, see for example, Whiteley et al. (2025). In such cases, a suitably structured and trained autoencoder may be able to recover the manifold, allowing BDC to achieve strong compression without sacrificing downstream performance. However, if the data lack any meaningful low-dimensional structure, this advantage disappears. Because BDC reduces dimensionality as well as sample size, it risks discarding too much information and may underperform compared to methods that operate entirely in ambient space. An example of this may be behind the results given in Section D.1.3, where ADC outperforms BDC in terms of predictive metrics.

### A.2.3 CONVERGENCE GUARANTEES

Step 2 of our optimisation procedure can be analysed as a discretised Wasserstein gradient flow, for which global convergence can be guaranteed under certain convexity assumptions on the objective (Arbel et al., 2019). By contrast, Step 1 of our two-stage procedure—autoencoder training—does not have a general theoretical convergence guarantee. Establishing such a guarantee is difficult in general. Convergence results are known in more restricted settings, such as for training *linear* neural networks under the MSRE loss, and these may be possible to extend to linear autoencoders targeting the RMMD. More generally, recent work has shown convergence for shallow autoencoders under the MSRE loss when data are generated from specific models (Nguyen et al., 2019), but extending these results to broader settings, and in particular to our RMMD objective, is a very challenging task.

While our experiments demonstrate stable convergence across a range of synthetic and real-world distributions, a general proof remains an open problem.

An associated limitation is the choice of latent dimension $p$. Numerous methods exist for estimating the intrinsic dimension of data (Kégl, 2002; Levina & Bickel, 2004; Hein & Audibert, 2005; Carter et al., 2010; Little, 2011), but these approaches often yield conflicting estimates and, depending on the method, may be computationally expensive as a preprocessing step. In this work, we instead selected $p$ pragmatically, aiming for values that yield sufficiently low RMMD. This reflects a broader challenge: the problem of choosing $p$ is analogous to that of selecting the compressed set size $m$, in that both are difficult to determine a priori. A practical strategy may therefore be to base these choices on the computational resources available—balancing the cost of constructing the compressed set against the demands of the downstream task. Nevertheless, an interesting direction for future work would be to explore integrating intrinsic-dimension estimation methods with BDC.

### A.2.4 GENERAL FEATURE DOMAINS

In this work we have restricted our study of bilateral distribution compression to the setting where data lie in a real ambient space $\mathbb{R}^d$. While the framework naturally extends to more general feature domains $\mathcal{X}$, provided a positive definite kernel can be defined, we have not explored such cases empirically. In principle, the use of an autoencoder may allow us to extend gradient-based distribution compression to domains where direct gradient computation on the raw data is difficult or infeasible, such as graphs, text, and other structured or discrete inputs, where non-gradient methods like Kernel Thinning (Dwivedi & Mackey, 2021b) would be the standard. In these settings, the autoencoder maps the data into a continuous latent space where gradient-based optimisation is tractable, potentially enabling efficient, high-quality compression even in challenging modalities. We leave a detailed investigation of this potential capability to future work.

### A.3 FUTURE WORK

This section provides a detailed discussion of several promising directions for future work.

### A.3.1 CONDITIONAL BILATERAL DISTRIBUTION COMPRESSION

In Section 3.3, we discussed how the Bilateral Distribution Compression framework developed in this work can be readily extended to target joint distributions by modifying the DMMD, RMMD, and EMMD to operate on the corresponding joint distributions. Recently, Broadbent et al. (2025) introduced *conditional distribution compression* via the *Average Maximum Conditional Mean Discrepancy* (AMCMD). This naturally raises the question of whether the framework proposed here can be extended to bilaterally compress conditional distributions.

We address this by first briefly introducing the *Kernel Conditional Mean Embedding* (KCME).

Under the integrability condition $\mathbb{E}_{\mathbb{P}_Y}[\sqrt{l(Y,Y)}] < \infty$, the KCME of $\mathbb{P}_{Y|X}$ is defined as $\mu_{Y|X} := \mathbb{E}_{\mathbb{P}_{Y|X}}[l(Y,\cdot) \mid X]$ where $\mu_{Y|X} : \Omega \to \mathcal{H}_l$ is an $X$-measurable random variable outputting *functions* in $\mathcal{H}_l$. Similar to the unconditional case, for any $f \in \mathcal{H}_l$, it can be shown that $\mathbb{E}_{\mathbb{P}_{Y|X}}[f(Y) \mid X] \overset{\text{a.s.}}{=} \langle f, \mu_{Y|X}\rangle_{\mathcal{H}_l}$ (Park & Muandet, 2020). The KCME can be written as the composition of a function $F_{Y|X} : \mathcal{X} \to \mathcal{H}_l$ and the random variable $X : \Omega \to \mathcal{X}$, i.e. $\mu_{Y|X} = F_{Y|X} \circ X$ (Theorem 4.1, Park & Muandet (2020)). Whenever we refer to the KCME $\mu_{Y|X}$, we mean $F_{Y|X}$. The KCME can be estimated directly (Park & Muandet, 2020; Grünewälder et al., 2012) using i.i.d. samples from the joint distribution $\mathcal{D}$ as

$$\hat{\mu}_{Y|X}^{\mathcal{D}} := \sum_{i,j=1}^{n} k(\boldsymbol{x}_i, \cdot) W_{ij} l(\boldsymbol{y}_j, \cdot)$$

where the superscript $\mathcal{D}$ refers to the data used to estimate $\mu_{Y|X}$, we define $W := (K + \lambda I)^{-1}$, $[K]_{ij} = k(\boldsymbol{x}_i, \boldsymbol{x}_j)$, $i, j = 1, \ldots, n$, and $\lambda > 0$ is a regularisation parameter.

Given an additional random variable $X^* : \Omega \to \mathcal{X}$ with distribution $\mathbb{P}_{X^*}$, Broadbent et al. (2025) introduce the AMCMD as the target of a distribution compression procedure. The AMCMD is

defined as

$$\text{AMCMD}\left[\mathbb{P}_{X^*}, \mathbb{P}_{Y|X}, \mathbb{P}_{Y'|X'}\right] := \sqrt{\mathbb{E}_{\boldsymbol{x} \sim \mathbb{P}_{X^*}}\left[\|\mu_{Y|X=\boldsymbol{x}} - \mu_{Y'|X'=\boldsymbol{x}}\|_{\mathcal{H}_l}^2\right]}.$$

Much like the MMD, the AMCMD can be shown to be a proper metric:

**Theorem A.1** *(Theorem 4.1, Broadbent et al. (2025)) Suppose that $l : \mathcal{Y} \times \mathcal{Y} \to \mathbb{R}$ is a characteristic kernel, that $\mathbb{P}_X$, $\mathbb{P}_{X'}$, and $\mathbb{P}_{X^*}$ are absolutely continuous with respect to each other, and that $\mathbb{P}(\cdot \mid X)$ and $\mathbb{P}(\cdot \mid X')$ admit regular versions. Then, $AMCMD\left[\mathbb{P}_{X^*}, \mathbb{P}_{Y|X}, \mathbb{P}_{Y'|X'}\right] = 0$ if and only if, for almost all $\boldsymbol{x} \in \mathcal{X}$ wrt $\mathbb{P}_{X^*}$, $\mathbb{P}_{Y|X=\boldsymbol{x}}(B) = \mathbb{P}_{Y'|X'=\boldsymbol{x}}(B)$ for all $B \in \mathscr{Y}$.*

*Moreover, assuming the Radon-Nikodym derivatives $\frac{d\mathbb{P}_{X^*}}{d\mathbb{P}_X}$, $\frac{d\mathbb{P}_{X^*}}{d\mathbb{P}_{X'}}$, $\frac{d\mathbb{P}_{X^*}}{d\mathbb{P}_{X''}}$ are bounded, then the triangle inequality is satisfied, i.e. $AMCMD\left[\mathbb{P}_{X^*}, \mathbb{P}_{Y|X}, \mathbb{P}_{Y''|X''}\right] \le AMCMD\left[\mathbb{P}_{X^*}, \mathbb{P}_{Y|X}, \mathbb{P}_{Y'|X'}\right] + AMCMD\left[\mathbb{P}_{X^*}, \mathbb{P}_{Y'|X'}, \mathbb{P}_{Y''|X''}\right].$*

Setting $X^* = X$ and omitting the first argument of the AMCMD, we obtain conditional analogues of the DMMD, RMMD, and EMMD:

$$\text{DAMCMD}(\mathbb{P}_{Y|X}, \mathbb{P}_{Y|\phi(Z)}), \quad \text{ARMCMD}(\mathbb{P}_{Y|X}, \mathbb{P}_{Y|\phi(\psi(X))}), \quad \text{AEMCMD}(\mathbb{P}_{Y|\psi(X)}, \mathbb{P}_{Y|Z})$$

corresponding to the *Average Decoded Maximum Conditional Mean Discrepancy* (ADMCMD), *Average Reconstructed Maximum Conditional Mean Discrepancy* (ARMCMD), and *Average Encoded Maximum Conditional Mean Discrepancy* (AEMCMD), respectively. Investigating the relationships between these metrics, and the impact of training autoencoders and bilaterally compressed sets under these conditional objectives, represents a promising direction for future research.

### A.3.2 SEMI-SUPERVISED LEARNING

In Sections 3.3 and A.3.1 we discussed how one may extend the framework developed in this work to target joint and conditional distributions of labelled datasets. However, in many practical settings, fully labelled datasets are difficult or expensive to obtain, while unlabelled data is abundant. Semi-supervised learning seeks to exploit this imbalance by combining the limited information provided by labelled samples with the broader structure revealed by unlabelled data (Chapelle et al., 2009). Our BDC framework can be extended to the semi-supervised setting. Here, we present the joint distribution extension, leaving the conditional case as a natural next step.

We may adapt the two-stage objectives to encourage agreement between distributions at both the marginal and joint levels. Concretely, when training the autoencoder, rather than targeting only the RJMMD on the subset of samples with paired responses, one could instead minimise

$$\text{RJMMD}(\hat{\mathbb{P}}_{X,Y}, \hat{\mathbb{P}}_{\phi(\psi(X)),Y}) + \text{RMMD}(\hat{\mathbb{P}}_X, \hat{\mathbb{P}}_{\phi(\psi(X))}),$$

thereby incorporating both joint information from labelled samples and marginal information from the full dataset. Similarly, when training the compressed set in latent space, we target not only the EJMMD on labelled data but also the EMMD on the unlabelled marginal:

$$\text{EJMMD}(\hat{\mathbb{P}}_{\psi(X),Y}, \hat{\mathbb{P}}_{Z,W}) + \text{EMMD}(\hat{\mathbb{P}}_{\psi(X)}, \hat{\mathbb{P}}_Z).$$

In this way, the valuable information from labels may be leveraged without discarding the comparatively more abundant unlabelled samples.

A full investigation of the effectiveness of this semi-supervised extension is beyond the scope of the present work, and we leave a detailed empirical evaluation to future work.

### A.3.3 DISTRIBUTIONAL PRINCIPAL AUTOENCODERS

Very recently, a new class of autoencoders—the Distributional Principal Autoencoder (DPA) (Shen & Meinshausen, 2024a; Leban, 2025)—has been introduced. This approach guarantees that, in population, the reconstructed data follow the *same distribution* as the original data, thereby achieving *distributionally lossless* compression, regardless of the degree of dimensionality reduction. This claim is supported empirically by measuring the energy distance between reconstructed and original data, which remains very small even under substantial compression. In this work, we instead enforce distributional equality by directly optimising the Reconstruction MMD; however, in Section

D.1.5 we have observed that aggressive dimensionality reduction can still degrade reconstruction quality in terms of RMMD. It therefore may be interesting to explore how the latent spaces given by DPAs, when used in our BDC framework, perform on downstream tasks such as those consider in this work.

We now provide a brief summary of the DPA method, with additional comments once appropriate definitions have been presented.

Given a sample of noise $\epsilon \in \mathbb{R}^q$ following a pre-specified distribution $\mathbb{P}_E$, such as a standard Gaussian, the DPA decoder $\phi : \mathbb{R}^p \times \mathbb{R}^q \to \mathbb{R}^d$ aims to achieve *distributional reconstruction*. That is, for a given encoder $\psi : \mathbb{R}^d \to \mathbb{R}^p$, the decoder is trained so that

$$\phi(\boldsymbol{z}, \boldsymbol{\epsilon}) \overset{\mathrm{d}}{=} \left( X \mid \psi(X) = \boldsymbol{z} \right), \quad \forall \boldsymbol{z} \in \mathbb{R}^p,$$

where $\overset{\mathrm{d}}{=}$ denotes equality in distribution. In other words, given an embedding $\boldsymbol{z}$, the distribution of the random variable $\phi(\boldsymbol{z}, \boldsymbol{\epsilon})$ should match the conditional distribution of the original data whose embedding under $\psi$ equals $\boldsymbol{z}$. If this is achieved, then

$$\phi(\psi(X), \boldsymbol{\epsilon}) \overset{\mathrm{d}}{=} X,$$

which guarantees that the reconstructed data from the decoder have the same distribution $\mathbb{P}_X$ as the original data.

To achieve this, DPAs define the *Oracle Reconstructed Distribution* (ORD), denoted $\mathbb{P}^*_{\boldsymbol{x}, \psi}$, as the conditional distribution of $X$ given that its embedding under $\psi$ matches the embedding of a particular sample $\boldsymbol{x} \sim \mathbb{P}_X$, i.e.,

$$\mathbb{P}^*_{\boldsymbol{x}, \psi} = \mathbb{P}\left( X \mid \psi(X) = \psi(\boldsymbol{x}) \right).$$

In other words, the ORD is the distribution of $X$ restricted to the *level set* of the encoder $\psi$ that contains $\boldsymbol{x}$. The encoder should then optimised to minimise the expected variability of the induced ORD:

$$\psi^* = \arg\min_{\psi} \quad \mathbb{E}_{\boldsymbol{x} \sim \mathbb{P}_X} \left[ \mathbb{E}_{\boldsymbol{y}, \boldsymbol{y}' \sim \mathbb{P}^*_{\boldsymbol{x}, \psi}} \left[ \|\boldsymbol{y} - \boldsymbol{y}'\|_2^{\beta} \right] \right],$$

where $0 < \beta \leq 2$ is a hyperparameter controlling the penalty on variability. They show (Proposition 1, Shen & Meinshausen (2024a)) that this objective is equivalent to minimising the expected reconstruction error

$$\mathbb{E}_{\boldsymbol{x} \sim \mathbb{P}_X} \left[ \mathbb{E}_{\boldsymbol{y} \sim \mathbb{P}^*_{\boldsymbol{x}, \psi}} \left[ \|\boldsymbol{x} - \boldsymbol{y}\|_2^{\beta} \right] \right].$$

Moreover, they prove (Proposition 2, Shen & Meinshausen (2024a)) that, for multivariate Gaussian data and a linear autoencoder with $\beta = 2$, this procedure recovers the same subspace as PCA.

In practice, the ORD is not directly observable, so they propose an empirical surrogate objective, such that the autoencoder can be optimised from data. We present the more intuitive re-parameterised version:

$$\mathbb{E}_{\boldsymbol{x} \sim \mathbb{P}_X} \left[ \mathbb{E}_{\boldsymbol{\epsilon} \sim \mathbb{P}_E} \left[ \|\boldsymbol{x} - \phi(\psi(\boldsymbol{x}, \boldsymbol{\epsilon}))\|_2^{\beta} \right] \right] - \frac{1}{2} \mathbb{E}_{\boldsymbol{x} \sim \mathbb{P}_X} \left[ \mathbb{E}_{\boldsymbol{\epsilon}, \boldsymbol{\epsilon}' \sim \mathbb{P}_E} \left[ \|\phi(\psi(\boldsymbol{x}, \boldsymbol{\epsilon})) - \phi(\psi(\boldsymbol{x}, \boldsymbol{\epsilon}'))\|_2^{\beta} \right] \right],$$

where $\mathbb{P}_{\boldsymbol{z}, \phi}$ denotes the distribution of $\phi(\boldsymbol{z}, \boldsymbol{\epsilon})$ for any $\boldsymbol{z} \in \mathbb{R}^p$ and $\boldsymbol{\epsilon} \in \mathbb{R}^q$. They prove (Theorem 1, Shen & Meinshausen (2024a)), assuming the existence of the ORD within the chosen class of autoencoder, that the optimal encoder according to this surrogate objective achieves the minima of the expected variability of the induced ORD, and that the optimal decoder induces the oracle reconstructed distribution, as desired.

**Kernelised DPA:** Note that the final objective function is the expected *negative energy score* between $\mathbb{P}_X$ and the fitted distribution $\mathbb{P}_{\phi, \psi(X)}$. The energy score is closely related to the *energy distance*, and, analogously, the *kernelised energy score* is closely related to the *MMD* (Shen & Meinshausen, 2024b). In fact, the energy score and energy distance arise as special cases of these quantities for a particular choice of negative definite kernel. This connection suggests that an explicitly kernelised version of DPA would also be worth exploring: the flexibility to choose a kernel has been shown to improve performance in standard distribution compression—e.g., in gradient flow methods (Arbel et al., 2019) compared to Support Points (Mak & Joseph, 2018), which target the energy distance. One might therefore expect a kernelised DPA to outperform standard DPA in domains where the energy distance kernel is suboptimal.

### A.4 Ethics and Reproducibility Statement

**Ethics**: Our study does not involve human subjects, personally identifiable information, or sensitive data. All datasets used are publicly available benchmark datasets, and we follow their respective licences and usage guidelines. The proposed methods are intended for general machine learning research and carry no foreseeable risks of misuse or harmful application beyond standard considerations in the field.

**Reproducibility**: We have taken several measures to ensure the reproducibility of our results. Complete algorithmic details for BDC are provided in Section 3.1, with pseudocode in Algorithms 1 and 2. Theoretical guarantees, assumptions, and proofs are given in the appendix. All datasets, preprocessing steps, and experimental setups are described in detail in Section 4 and D.1. A code repository containing an implementation of BDC is provided at: `https://github.com/bilateraldummy/bilateral_dummy`.

## B Proofs

In this section, we provide technical proofs for the results in the main paper, and rigorously describe and discuss the assumptions that we adopt throughout the paper.

### B.1 Technical Assumptions and Additional Notation

The assumptions that we work under are laid out in this section, alongside commentary on their restrictiveness, as well as some additional notation used in our proofs.

**Notation**: Each positive definite kernel function $k : \mathcal{X} \times \mathcal{X} \to \mathbb{R}$ induces a vector space of functions from $\mathcal{X}$ to $\mathbb{R}$, known as a *Reproducing Kernel Hilbert Space* (RKHS) (Aronszajn, 1950), denoted by $\mathcal{H}_k$. An RKHS $\mathcal{H}_k$ is defined by two key properties: (i) For all $\boldsymbol{x} \in \mathcal{X}$, the function $k(\boldsymbol{x}, \cdot) : \mathcal{X} \to \mathbb{R}$ belongs to $\mathcal{H}_k$; (ii) The kernel function $k$ satisfies the *reproducing property*, meaning that for all $f \in \mathcal{H}_k$ and $\boldsymbol{x} \in \mathcal{X}$, we have $f(\boldsymbol{x}) = \langle f(\cdot), k(\boldsymbol{x}, \cdot) \rangle_{\mathcal{H}_k}$, where $\langle \cdot, \cdot \rangle_{\mathcal{H}_k}$ denotes the inner product in $\mathcal{H}_k$. Under the integrability condition $\mathbb{E}_{\mathbb{P}_X}[\sqrt{k(X, X)}] < \infty$, the *kernel mean embedding* $\mu_X(\cdot) := \mathbb{E}[k(X, \cdot)]$ of $\mathbb{P}_X$ is an element of $\mathcal{H}_k$, and moreover, $\mathbb{E}_{\mathbb{P}_X}[f(X)] = \langle f, \mu_X \rangle_{\mathcal{H}_k}$ for all $f \in \mathcal{H}_k$ (Smola et al., 2007).

**Technical Assumptions** Throughout the paper we assume all kernels are measurable and satisfy the integrability condition $\mathbb{E}_{\boldsymbol{x} \sim \mathbb{P}_X}[\sqrt{k(\boldsymbol{x}, \boldsymbol{x})}] < \infty$ to ensure that the kernel mean embedding $\mu_X \in \mathcal{H}_k$. This condition is trivially satisfied for any bounded kernel, such as the commonly used Gaussian, Laplace, Matérn, and inverse multi-quadratic kernels (Sriperumbudur et al., 2012). We further assume that the autoencoder $\phi \circ \psi$ is measurable; this is trivially satisfied for a linear autoencoder, and for neural autoencoders it holds because they are compositions of measurable maps. When working with labelled data, as in Section 3.3, we instead require $\mathbb{E}_{\boldsymbol{x} \sim \mathbb{P}_X}[k(\boldsymbol{x}, \boldsymbol{x})] < \infty$ and $\mathbb{E}_{\boldsymbol{y} \sim \mathbb{P}_Y}[l(\boldsymbol{y}, \boldsymbol{y})] < \infty$ to guarantee that the joint kernel mean embedding $\mu_{X,Y} \in \mathcal{H}_k \otimes \mathcal{H}_l$. Once again this is trivially satisfied for any choice of bounded kernel.

Note that we prove Theorems 3.3 and 3.5 in their full generality, i.e. for arbitrary ambient feature space $\mathcal{X}$.

### B.2 Proof of Theorem 3.3

**Theorem B.1** *Let* $k : \mathcal{X} \times \mathcal{X} \to \mathbb{R}$ *and* $h : \mathbb{R}^p \times \mathbb{R}^p \to \mathbb{R}$ *be characteristic kernels, and let* $\phi \circ \psi : \mathcal{X} \to \mathcal{X}$ *be an autoencoder defined by the composition of measurable maps* $\psi : \mathcal{X} \to \mathbb{R}^p$ *and* $\phi : \mathbb{R}^p \to \mathcal{X}$. *Suppose* $\mathbb{P}_X$ *and* $\mathbb{P}_Z$ *are distributions such that*

$$RMMD_k\big(\mathbb{P}_X, \mathbb{P}_{\phi(\psi(X))}\big) = 0 \quad and \quad EMMD_h\big(\mathbb{P}_{\psi(X)}, \mathbb{P}_Z\big) = 0.$$

*Then it follows that* $DMMD_k\big(\mathbb{P}_X, \mathbb{P}_{\phi(Z)}\big) = 0.$

**Proof**: Given that $\text{EMMD}_h\big(\mathbb{P}_{\psi(X)}, \mathbb{P}_Z\big) = 0$, we have that $\mathbb{P}_{\psi(X)} = \mathbb{P}_Z$, that is, $\mathbb{P}_Z$ is the pushforward measure of $\mathbb{P}_X$ under the encoder $\psi : \mathcal{X} \to \mathbb{R}^p$ :

$$\mathbb{P}_Z(\mathcal{A}) = \mathbb{P}_X(\psi^{-1}\mathcal{A}), \quad \mathcal{A} \subset \mathbb{R}^p,$$

where we note that

$$\psi^{-1}\mathcal{A} = \{\boldsymbol{x} \in \mathcal{X} \mid \psi(\boldsymbol{x}) \in \mathcal{A}\}$$

is the inverse image of $\mathcal{A}$ under $\psi$ (unrelated to the functional inverse of $\psi$).

Now, consider $\mathbb{P}_{\phi(Z)}$, i.e. the pushforward of $\mathbb{P}_Z$ under the decoder $\phi : \mathbb{R}^p \to \mathcal{X}$, then we have

$$\mathbb{P}_{\phi(Z)}(\mathcal{B}) = \mathbb{P}_Z(\phi^{-1}\mathcal{B}) = \mathbb{P}_X(\psi^{-1}[\phi^{-1}\mathcal{B}]), \quad \mathcal{B} \subset \mathcal{X}$$

where we have

$$\psi^{-1}[\phi^{-1}\mathcal{B}] = \{\boldsymbol{x} \in \mathcal{X} \mid \phi(\psi(\boldsymbol{x})) \in \mathcal{B}\} = [\phi \circ \psi]^{-1}\mathcal{B}.$$

But this pushforward measure is exactly $\mathbb{P}_{\phi(\psi(X))}$, i.e. the pushforward measure of $\mathbb{P}_X$ under the full autoencoder $\phi \circ \psi : \mathcal{X} \to \mathcal{X}$.

All there is left to do is note that we have assumed $\mathrm{RMMD}\left(\mathbb{P}_X, \mathbb{P}_{\phi(\psi(X))}\right) = 0$, and hence we have $\mathbb{P}_{\phi(Z)} = \mathbb{P}_{\phi(\psi(X))} = \mathbb{P}_X$ and thus $\mathrm{DMMD}\left(\mathbb{P}_X, \mathbb{P}_{\phi(Z)}\right) = 0$. ∎

### B.3 PROOF OF THEOREM 3.5

**Theorem B.2** *Let $k : \mathcal{X} \times \mathcal{X} \to \mathbb{R}$ be a positive definite kernel, and let $\psi : \mathcal{X} \to \mathbb{R}^p$, $\phi : \mathbb{R}^p \to \mathcal{X}$ be measurable. Given the pull-back kernel $h : \mathbb{R}^p \times \mathbb{R}^p \to \mathbb{R}$*

$$h(\boldsymbol{z}, \boldsymbol{z}') = k\left(\phi(\boldsymbol{z}), \phi(\boldsymbol{z}')\right),$$

*we have that, for any probability measures $\mathbb{P}_X$ and $\mathbb{P}_Z$,*

$$\mathrm{DMMD}\left(\mathbb{P}_X, \mathbb{P}_{\phi(Z)}\right) \leq \mathrm{RMMD}\left(\mathbb{P}_X, \mathbb{P}_{\phi(\psi(X))}\right) + \mathrm{EMMD}\left(\mathbb{P}_{\psi(X)}, \mathbb{P}_Z\right).$$

**Proof**: By the triangle inequality we have

$$\begin{aligned}
\mathrm{DMMD}_k\left(\mathbb{P}_X, \mathbb{P}_{\phi(Z)}\right) &= \|\mu_X - \mu_{\phi(Z)}\|_{\mathcal{H}_k} \\
&= \|\mu_X - \mu_{\phi(\psi(X))} + \mu_{\phi(\psi(X))} - \mu_{\phi(Z)}\|_{\mathcal{H}_k} \\
&\leq \|\mu_X - \mu_{\phi(\psi(X))}\|_{\mathcal{H}_k} + \|\mu_{\phi(\psi(X))} - \mu_{\phi(Z)}\|_{\mathcal{H}_k}.
\end{aligned}$$

The first term can be immediately recognised as the Reconstruction MMD, i.e.

$$\mathrm{RMMD}_k\left(\mathbb{P}_X, \mathbb{P}_{\phi(\psi(X))}\right) := \|\mu_X - \mu_{\phi(\psi(X))}\|_{\mathcal{H}_k},$$

whereas the second term $\|\mu_{\phi(\psi(X))} - \mu_{\phi(Z)}\|_{\mathcal{H}_k}$ does not immediately look quite like the Encoded MMD:

$$\mathrm{EMMD}_h\left(\mathbb{P}_{\psi(X)}, \mathbb{P}_Z\right) := \|\mu_{\psi(X)} - \mu_Z\|_{\mathcal{H}_h}.$$

However, by defining the pull-back kernel

$$h(\boldsymbol{z}, \boldsymbol{z}') := k\left(\phi(\boldsymbol{z}), \phi(\boldsymbol{z}')\right),$$

and expanding $\mathrm{EMMD}_h^2\left(\mathbb{P}_{\psi(X)}, \mathbb{P}_Z\right)$, we have

$$\begin{aligned}
\mathrm{EMMD}_h^2\left(\mathbb{P}_{\psi(X)}, \mathbb{P}_Z\right) &:= \|\mu_{\psi(X)} - \mu_Z\|_{\mathcal{H}_h}^2 \\
&= \mathbb{E}_{\boldsymbol{x},\boldsymbol{x}'\sim\mathbb{P}_X}[h(\psi(\boldsymbol{x}),\psi(\boldsymbol{x}'))] - 2\mathbb{E}_{\boldsymbol{x}\sim\mathbb{P}_X, \boldsymbol{z}\sim\mathbb{P}_Z}[h(\psi(\boldsymbol{x}),\boldsymbol{z})] \\
&\quad + \mathbb{E}_{\boldsymbol{z},\boldsymbol{z}'\sim\mathbb{P}_Z}[h(\boldsymbol{z},\boldsymbol{z}')] \\
&= \mathbb{E}_{\boldsymbol{x},\boldsymbol{x}'\sim\mathbb{P}_X}[k(\phi(\psi(\boldsymbol{x})),\phi(\psi(\boldsymbol{x}')))] - 2\mathbb{E}_{\boldsymbol{x}\sim\mathbb{P}_X, \boldsymbol{z}\sim\mathbb{P}_Z}[k(\phi(\psi(\boldsymbol{x})),\phi(\boldsymbol{z}))] \\
&\quad + \mathbb{E}_{\boldsymbol{z},\boldsymbol{z}'\sim\mathbb{P}_Z}[k(\phi(\boldsymbol{z}),\phi(\boldsymbol{z}'))] \\
&= \|\mu_{\phi(\psi(X))} - \mu_{\phi(Z)}\|_{\mathcal{H}_k}^2
\end{aligned}$$

where the first line is the definition of the EMMD, the second line follows from the standard expansion of the squared RKHS norm, the third line applies the definition of the pull-back kernel, and the fourth line again uses the standard RKHS norm expansion.

Therefore, with the pull-back kernel,

$$\mathrm{EMMD}\left(\mathbb{P}_{\psi(X)}, \mathbb{P}_Z\right) = \|\mu_{\phi(\psi(X))} - \mu_{\phi(Z)}\|_{\mathcal{H}_k}$$

and we are finished. ∎

### B.4 PROOF OF THEOREM 3.1

**Theorem B.3** *Assume the distribution $\mathbb{P}_X$ has zero mean, and let the kernel $k : \mathbb{R}^d \times \mathbb{R}^d \to \mathbb{R}$ be the second-order polynomial (quadratic) kernel defined by $k(\boldsymbol{x}, \boldsymbol{y}) = (1 + \boldsymbol{x}^\top \boldsymbol{y})^2$. Then the optimal solution to the optimisation problem*

$$V^* = \underset{V \in \mathbb{R}^{d \times p}, \ V^\top V = I_p}{\arg\min} \ RMMD^2 \left( \hat{\mathbb{P}}_X, \hat{\mathbb{P}}_{XVV^\top} \right), \tag{7}$$

*is given by (a permutation of) the top $p$ eigenvectors of the sample covariance matrix.*

**Proof**:

Given an additional random variable $X' : \Omega \to \mathbb{R}^d$ with distribution $\mathbb{P}_{X'}$, and the quadratic kernel $k(\boldsymbol{x}, \boldsymbol{y}) = (1 + \boldsymbol{x}^\top \boldsymbol{y})^2$, Sriperumbudur et al. (2010, Example 3) show that

$$\mathrm{MMD}^2(\mathbb{P}_X, \mathbb{P}_{X'}) = 2\|\boldsymbol{\tau}_X - \boldsymbol{\tau}_{X'}\|_2^2 + \|\Sigma_X - \Sigma_{X'} + \boldsymbol{\tau}_X \boldsymbol{\tau}_X^\top - \boldsymbol{\tau}_{X'} \boldsymbol{\tau}_{X'}^\top\|_F^2,$$

where $\boldsymbol{\tau}_X := \int_{\mathbb{R}^d} \boldsymbol{x} \, d\mathbb{P}_X(\boldsymbol{x})$ is the mean of $\mathbb{P}_X$, $\Sigma_X := \int_{\mathbb{R}^d} \boldsymbol{x}\boldsymbol{x}^\top \, d\mathbb{P}_X(\boldsymbol{x}) - \boldsymbol{\tau}_X \boldsymbol{\tau}_X^\top$ is the covariance matrix, and $\|\cdot\|_F$ denotes the Frobenius norm.

If $\mathbb{P}_X$ is zero-mean, and $P = VV^\top$ is an orthogonal projection, then $\mathbb{P}_{XVV^\top}$ is also zero-mean. In this case, the mean term vanishes, and we obtain

$$\mathrm{RMMD}^2 \left( \mathbb{P}_X, \mathbb{P}_{XVV^\top} \right) = \|\Sigma_X - \Sigma_{XVV^\top}\|_F^2.$$

Given (zero-mean) samples $\mathcal{D} = \{\boldsymbol{x}_i\}_{i=1}^n \overset{\text{i.i.d.}}{\sim} \mathbb{P}_X$, let $\boldsymbol{X} \in \mathbb{R}^{n \times d}$ be the data matrix. The sample covariance matrices are

$$\hat{\Sigma}_X := \frac{1}{n-1} \boldsymbol{X}^\top \boldsymbol{X}, \quad \hat{\Sigma}_{XVV^\top} := \frac{1}{n-1} (\boldsymbol{X}VV^\top)^\top (\boldsymbol{X}VV^\top)$$

$$= \frac{1}{n-1} VV^\top \boldsymbol{X}^\top \boldsymbol{X} VV^\top.$$

Hence,

$$\mathrm{RMMD}^2 \left( \hat{\mathbb{P}}_X, \hat{\mathbb{P}}_{XVV^\top} \right) = \left\| \frac{1}{n-1} \boldsymbol{X}^\top \boldsymbol{X} - \frac{1}{n-1} VV^\top \boldsymbol{X}^\top \boldsymbol{X} VV^\top \right\|_F^2.$$

Dropping the constant factor $(n-1)^{-1}$, the resulting optimisation problem is

$$V^* = \underset{V \in \mathbb{R}^{d \times p}, \ V^\top V = I_p}{\arg\min} \left\| \boldsymbol{X}^\top \boldsymbol{X} - VV^\top \boldsymbol{X}^\top \boldsymbol{X} VV^\top \right\|_F^2,$$

which coincides with (7). The next step is to simplify the objective function, firstly we have

$$\left\| \boldsymbol{X}^\top \boldsymbol{X} - VV^\top \boldsymbol{X}^\top \boldsymbol{X} VV^\top \right\|_F^2 = \|\boldsymbol{X}\boldsymbol{X}^\top\|_2^2 - 2\langle \boldsymbol{X}^\top \boldsymbol{X}, VV^\top \boldsymbol{X}^\top \boldsymbol{X} VV^\top \rangle_F$$

$$+ \langle VV^\top \boldsymbol{X}^\top \boldsymbol{X} VV^\top, VV^\top \boldsymbol{X}^\top \boldsymbol{X} VV^\top \rangle_F$$

but $\langle A, B \rangle_F = \mathrm{Tr}(A^\top B)$, hence

$$\langle \boldsymbol{X}^\top \boldsymbol{X}, VV^\top \boldsymbol{X}^\top \boldsymbol{X} VV^\top \rangle_F = \mathrm{Tr}(\boldsymbol{X}^\top \boldsymbol{X} VV^\top \boldsymbol{X}^\top \boldsymbol{X} VV^\top)$$

and

$$\langle VV^\top \boldsymbol{X}^\top \boldsymbol{X} VV^\top, VV^\top \boldsymbol{X}^\top \boldsymbol{X} VV^\top \rangle_F = \mathrm{Tr}(VV^\top \boldsymbol{X}^\top \boldsymbol{X} VV^\top VV^\top \boldsymbol{X}^\top \boldsymbol{X} VV^\top)$$

$$= \mathrm{Tr}(VV^\top \boldsymbol{X}^\top \boldsymbol{X} VV^\top \boldsymbol{X}^\top \boldsymbol{X} VV^\top)$$

$$= \mathrm{Tr}(V^\top \boldsymbol{X}^\top \boldsymbol{X} VV^\top \boldsymbol{X}^\top \boldsymbol{X} VV^\top V)$$

$$= \mathrm{Tr}(V^\top \boldsymbol{X}^\top \boldsymbol{X} VV^\top \boldsymbol{X}^\top \boldsymbol{X} V)$$

$$= \mathrm{Tr}(\boldsymbol{X}^\top \boldsymbol{X} VV^\top \boldsymbol{X}^\top \boldsymbol{X} VV^\top)$$

where the first line follows by definition of the Frobenius inner product, the second line by the the fact $V^\top V = I_p$, the third line by the cyclic property of the trace, the fourth line again by $V^\top V = I_p$, and the last line by the cyclic property again.

So, ignoring invariant terms, summing identical terms, and applying the cyclic property once more, we have

$$V^* = \underset{V \in \mathbb{R}^{d \times p}, \, V^\top V = I_p}{\arg\max} \operatorname{Tr}(VV^\top \boldsymbol{X}^\top \boldsymbol{X} VV^\top \boldsymbol{X}^\top \boldsymbol{X}).$$

Or, by defining $\hat{\Sigma} := \boldsymbol{X}^\top \boldsymbol{X}$ for ease of notation, and shifting to the equivalent formulation wrt rank-$p$ idempotent orthogonal projection matrices $P = VV^\top$:

$$P^* = \underset{P^2 = P = P^\top, \, \operatorname{rank}(P) = p}{\arg\max} \operatorname{Tr}(P\hat{\Sigma}P\hat{\Sigma}).$$

Since $\hat{\Sigma}$ is symmetric and $P$ is an orthogonal projector ($P^\top = P$, $P^2 = P$), we have

$$\operatorname{Tr}(P\hat{\Sigma}P\hat{\Sigma}) = \operatorname{Tr}\left((\hat{\Sigma}P)^\top (P\hat{\Sigma})\right) = \langle \hat{\Sigma}P, \, P\hat{\Sigma} \rangle_F.$$

Then, by the Cauchy–Schwarz inequality,

$$\langle \hat{\Sigma}P, \, P\hat{\Sigma} \rangle_F \leq \|\hat{\Sigma}P\|_F \, \|P\hat{\Sigma}\|_F.$$

But both norms are equal, since

$$\|\hat{\Sigma}P\|_F^2 = \operatorname{Tr}(P\hat{\Sigma}^2 P) = \|P\hat{\Sigma}\|_F^2,$$

where the equality follows from $P^\top = P$ and the symmetry of $\hat{\Sigma}$. Thus,

$$\operatorname{Tr}(P\hat{\Sigma}P\hat{\Sigma}) \leq \operatorname{Tr}(P\hat{\Sigma}^2 P) = \operatorname{Tr}(P^2 \hat{\Sigma}^2) = \operatorname{Tr}(P\hat{\Sigma}^2),$$

using cyclicity of the trace and the idempotence of $P$. Now, using the John Von Neumann trace inequality (Mirsky, 1975), we have

$$\operatorname{Tr}(P\hat{\Sigma}^2) \leq \sum_{i=1}^{d} 1 \cdot \lambda_k^2$$

where we have additionally used the fact that the eigenvalues of idempotent matrices are either $0$ or $1$, with exactly $p$ ones since $\operatorname{rank}(P) = p$, and $\lambda_1 \geq \cdots \geq \lambda_d \geq 0$ are the eigenvalues of the symmetric positive-definite matrix $\hat{\Sigma}$.

Equality holds if and only if $P$ and $\hat{\Sigma}$ share the same eigenvectors, with the $p$ ones in $P$'s spectrum assigned to the $p$ largest eigenvalues $\lambda_1^2, \ldots, \lambda_p^2$ of $\hat{\Sigma}^2$. Since $\hat{\Sigma}$ and $\hat{\Sigma}^2$ share the same eigenvectors, that is

$$\hat{\Sigma} = U\Lambda U^\top, \quad \hat{\Sigma}^2 = U\Lambda^2 U^\top,$$

it follows that $P$ must be the orthogonal projector onto the span of the top $p$ eigenvectors of $\hat{\Sigma}$. This yields the desired result: $V_*$ is the matrix whose columns are the top $p$ eigenvectors corresponding to the $p$ largest eigenvalues of $\hat{\Sigma}$. ∎

**Remark B.4** *Similar to PCA, $V_*$ need not be unique if $\lambda_p \not> \lambda_{p+1}$. However, unlike PCA, there is no guarantee that the the columns of $V_*$ will be ordered according to the size of the corresponding eigenvalue.*

### B.5 Bounding Integration Error in Latent and Ambient Space

To illustrate the practical consequences of our two-stage optimisation procedure, we also provide two simple results that characterise how well expectations of functions can be approximated after compression. These results show that both in the ambient space and the latent space, integration errors against functions in the corresponding RKHSs are bounded in terms of the discrepancies we control during optimisation.

**Lemma B.5** *With the choice of pull-back kernel given in Theorem 3.5, let $f \in \mathcal{H}_k$, then*

$$|\mathbb{E}_{\mathbb{P}_X}[f(X)] - \mathbb{E}_{\mathbb{P}_Z}[f(\phi(Z))]| \leq \|f\|_{\mathcal{H}_k} \left[ \operatorname{RMMD}\left(\mathbb{P}_X, \mathbb{P}_{\phi(\psi(X))}\right) + \operatorname{EMMD}\left(\mathbb{P}_{\psi(X)}, \mathbb{P}_Z\right) \right].$$

**Proof**: Follows directly from the fact that the MMD is an integral probability metric, together with the bound in Theorem 3.5.

**Lemma B.6** *Let $g \in \mathcal{H}_h$, then*

$$\left| \mathbb{E}_{\mathbb{P}_X}[g(\psi(X))] - \mathbb{E}_{\mathbb{P}_Z}[g(Z)] \right| \leq \|g\|_{\mathcal{H}_h} \mathrm{EMMD}\left(\mathbb{P}_{\psi(X)}, \mathbb{P}_Z\right).$$

**Proof**: Follows directly from the fact that the MMD is an integral probability metric.

**Remark B.7** *Lemma B.5 establishes that for any function $f \in \mathcal{H}_k$, the error in approximating integrals in the ambient space is controlled by our procedure. Lemma B.6 shows the analogous result in the latent space for any $g \in \mathcal{H}_h$. Together, these highlight that our framework ensures reliable integration both before and after decoding.*

# C  ADDITIONAL DETAILS

In this section, we provide additional details that could not be included in the main body of the paper due to space constraints.

## C.1  OPTIMISATION ON THE STIEFEL MANIFOLD

In Section 3.2.1, we consider the optimisation problem

$$V_* = \underset{V \in \mathbb{R}^{d \times p},; V^\top V = I_p}{\arg\min} \mathrm{RMMD}^2\left(\hat{\mathbb{P}}_X, \hat{\mathbb{P}}_{XVV^\top}\right).$$

This is a *Stiefel manifold*–constrained optimisation problem (Absil et al., 2007). Numerous algorithms have been developed for optimisation on manifolds; here, we describe a standard gradient-based approach.

### C.1.1  STIEFEL MANIFOLD OPTIMISATION

Before outlining the gradient descent algorithm, we briefly review the necessary mathematical background.

The Stiefel manifold is defined as

$$\mathrm{St}(d, p) := \{V \in \mathbb{R}^{d \times p} : V^\top V = I_p\},$$

i.e., the set of $\mathbb{R}^{d \times p}$ matrices with orthonormal columns. The *tangent space* $\mathcal{T}_V \mathrm{St}(d, p)$ at a point $V \in \mathrm{St}(d, p)$ is the set of all directions in which one can make an infinitesimal move from $V$ while remaining on the manifold. It is given by

$$\mathcal{T}_V \mathrm{St}(d, p) := \{W \in \mathbb{R}^{d \times p} : W^\top V + V^\top W = 0\},$$

where the condition $W^\top V + V^\top W = 0$ enforces that motion along $W$ preserves the orthonormality of the columns of $V$ to first order.

Given an arbitrary matrix $W \in \mathbb{R}^{d \times p}$, its *projection* onto the tangent space at $V$ removes any component that points off the manifold, ensuring the result is a valid tangent vector. This projection mapping $\Pi_V : \mathbb{R}^{d \times p} \to \mathcal{T}_V \mathrm{St}(d, p)$ is defined as

$$\Pi_V(W) := W - V \operatorname{sym}(V^\top W)$$
$$= W - \frac{1}{2} V(V^\top W + W^\top V),$$

where $\operatorname{sym}(A) := \frac{1}{2}(A + A^\top)$ extracts the symmetric part of $A$. Intuitively, $\Pi_V(W)$ subtracts the component of $W$ that would change the orthonormality of $V$'s columns, leaving only the component corresponding to a feasible on-manifold direction. Hence, if we move $V$ an infinitesimal amount in the direction $\Pi_V(W)$, the orthonormality of $V$ is preserved to first order, and the resulting displacement lies along the Stiefel manifold.

Because we do not perform any line search in our setting, we require a mechanism to map arbitrary elements of $\mathcal{T}_V \text{St}(d, p)$ back onto the Stiefel manifold after each update. This is achieved via a *retraction* (Absil et al., 2007). Among the many possible choices, we adopt the *QR retraction*:

$$\Phi_V(X) := \mathcal{Q}(V + X)$$
$$= Q \text{ factor from the QR decomposition of } V + X,$$

for $X \in \mathcal{T}_V \text{St}(d, p)$. Geometrically, the QR retraction first takes a step in the tangent direction $X$, then "snaps" the result back onto the manifold by orthonormalising its columns via the QR decomposition, thereby preserving the manifold constraint while remaining close to the intended update direction. The QR retraction is computationally efficient, numerically stable, and widely used in manifold optimisation (Absil et al., 2007).

### C.1.2 Gradient Descent on the Stiefel Manifold

We are now in a position to present a standard gradient descent procedure on the Stiefel manifold (Absil et al., 2007).

Given a loss function $L : \mathbb{R}^{d \times p} \to \mathbb{R}$ defined on arbitrary matrices $V \in \mathbb{R}^{d \times p}$, Stiefel-manifold–constrained gradient descent can be performed as follows:

**Step 1**: At the $t^{\text{th}}$ iteration, take the usual gradient of the objective function $\nabla_V L(V_t) \in \mathbb{R}^{d \times p}$ in Euclidean space, denote it by $G_t$.

**Step 2**: Project the Euclidean gradient $G_t$ onto the tangent space at the current iterate $V_t$:

$$\Pi_{V_t}(G_t) = G_t - V_t \text{sym}(V_t^\top G_t).$$

**Step 3**: We now have a valid descent direction, so given some step size $\gamma_t > 0$, we take a step as:

$$V' := V_t - \gamma_t \Pi_{V_t}(G_t).$$

**Step 4**: However, because we are not doing any kind of Stiefel-aware line search, this update is not guaranteed to lie on the Stiefel manifold, so we must retract it, i.e.

$$V_{t+1} = \mathcal{Q}(V_t - \gamma_t \Pi_{V_t}(G_t))$$
$$= \mathcal{Q}(V_t - \gamma_r \left[\nabla_V L(V_t) - V_t \text{sym}(V_t^\top \nabla_V L(V_t))\right]).$$

The updated parameter $V_{t+1}$ is guaranteed to remain on the Stiefel manifold, while also moving in a descent direction dictated by the loss function.

We have presented a standard gradient descent approach, but this can be extended to adaptive optimisers such as ADAM (Kingma & Ba, 2017) (see, e.g., Bécigneul & Ganea (2019); Li et al. (2020)). In practice, we use vanilla ADAM on gradients projected onto the Stiefel tangent space at each step, followed by a QR retraction to restore the orthonormality constraint. This provides a practical approximation to a full Stiefel-constrained ADAM, which also applies vector transport to momentum and second-moment estimates. In our setting, the simpler projected-gradient variant performs well while being much easier to implement.

### C.2 Pull-Back Kernel

In Theorem 3.5, we required the latent space kernel $h$ to be defined as a *pull-back kernel* (Paulsen & Raghupathi, 2016),

$$h(\mathbf{z}, \mathbf{z}') := k(\phi(\mathbf{z}), \phi(\mathbf{z}')), \quad \mathbf{z}, \mathbf{z}' \in \mathbb{R}^p,$$

where $\phi : \mathbb{R}^p \to \mathbb{R}^d$ is the decoder. Here, we briefly discuss the implications of this choice. If strict control of the DMMD is required with respect to an ambient-space kernel $k(\cdot, \cdot)$, for example, when the latent compressed set may later be decoded back into ambient space—then the latent-space kernel should be taken as the pull-back of $k$. However, if the objective is downstream performance in the latent space, this is not necessary, and it may be preferable to define the kernel directly in latent space. This has two advantages: (i) reduced computational cost, since the pull-back kernel requires evaluating $\phi$ for every pair $(\mathbf{z}, \mathbf{z}')$ in addition to the higher cost of evaluating distances in

ambient space; and (ii) the compressed set may more faithfully reflect the geometry of the latent space learned by the encoder.

Finally, we note that kernel lengthscales in this work are set using the median heuristic. For linear autoencoders, Euclidean distances are preserved exactly, so the median pairwise distance, and hence the kernel lengthscale under the heuristic (Garreau et al., 2018), is identical in both spaces. For nonlinear autoencoders, the encoder introduces distortions, so the median heuristic yields different scales depending on the chosen kernel.

### C.3 EXTENSION TO LABELLED DATA

In Section 3.3, we described how bilateral distribution compression can be extended to the labelled case, in this section we provide the full technical details. We begin by introducing the machinery required to embed *joint* distributions into function space, namely, the *tensor product reproducing kernel Hilbert space*.

#### C.3.1 TENSOR PRODUCT REPRODUCING KERNEL HILBERT SPACES

Let $l : \mathcal{Y} \times \mathcal{Y} \to \mathbb{R}$ be the reproducing kernel inducing the RKHS $\mathcal{H}_l$, and denote $\mathcal{H}_k \otimes \mathcal{H}_l$ to be the tensor product of the RKHSs $\mathcal{H}_k$ and $\mathcal{H}_l$, consisting of functions $g : \mathbb{R}^d \times \mathcal{Y} \to \mathbb{R}$. Then, for $h, h' \in \mathcal{H}_k$ and $f, f' \in \mathcal{H}_l$, the inner product in $\mathcal{H}_k \otimes \mathcal{H}_l$ is given by $\langle f \otimes g, f' \otimes g' \rangle_{\mathcal{H}_k \otimes \mathcal{H}_l} := \langle g, g' \rangle_{\mathcal{H}_k} \langle f, f' \rangle_{\mathcal{H}_l}$. Under the integrability condition $\mathbb{E}_{\mathbb{P}_X}[k(X, X)] < \infty$, $\mathbb{E}_{\mathbb{P}_Y}[l(Y, Y)] < \infty$, one can define the *joint* kernel mean embedding $\mu_{X,Y} := \mathbb{E}_{\mathbb{P}_{X,Y}}[k(X, \cdot)l(Y, \cdot)] \in \mathcal{H}_k \otimes \mathcal{H}_l$ such that $\mathbb{E}_{\mathbb{P}_{X,Y}}[g(X, Y)] = \langle \mu_{X,Y}, g \rangle_{\mathcal{H}_k \otimes \mathcal{H}_l}$ for all $g \in \mathcal{H}_k \otimes \mathcal{H}_l$ (Park & Muandet, 2020). The joint kernel mean embedding can be estimated straightforwardly as $\hat{\mu}_{X,Y} := \sum_{i=1}^n k(\boldsymbol{x}_i, \cdot)l(\boldsymbol{y}_i, \cdot)$ with i.i.d. samples from the joint distribution. The tensor product structure is advantageous as it permits the natural construction of a tensor product kernel from kernels defined on $\mathbb{R}^d$ and $\mathcal{Y}$. This insight is particularly applicable when $\mathbb{R}^d$ and $\mathcal{Y}$ have distinct characteristics that make a direct definition of a p.d. kernel difficult, for example, if $\mathcal{Y} = \{0, 1, \dots, C\}$, as is the case in multi-class classification tasks.

Given additional random variables $X' : \Omega \to \mathbb{R}^d$, $Y' : \Omega \to \mathcal{Y}$, and the embedding $\mu_{X',Y'}$ of $\mathbb{P}_{X',Y'}$, one can define the *Joint Maximum Mean Discrepancy* (JMMD) (Long et al., 2017) as

$$\text{JMMD}(\mathbb{P}_{X,Y}, \mathbb{P}_{X',Y'}) := \|\mu_{X,Y} - \mu_{X',Y'}\|_{\mathcal{H}_k \otimes \mathcal{H}_l}.$$

For a particular class of *characteristic* tensor product kernels (Sriperumbudur et al., 2011), the mapping $\mathbb{P}_{X,Y} \mapsto \mu_{X,Y}$ is *injective* (Szabó & Sriperumbudur, 2018). Hence, it is the case that $\text{JMMD}(\mathbb{P}_{X,Y}, \mathbb{P}_{X,Y}) = \|\mu_{X,Y} - \mu_{X',Y'}\|_{\mathcal{H}_k \otimes \mathcal{H}_l} = 0$ if and only if $\mathbb{P}_{X,Y} = \mathbb{P}_{X',Y'}$. By exploiting the reproducing property, we can express the square of the JMMD purely in terms of kernel evaluations as follows:

$$\begin{aligned}
\text{JMMD}^2(\mathbb{P}_{X,Y}, \mathbb{P}_{X',Y'}) = \; & \mathbb{E}_{(\boldsymbol{x},\boldsymbol{y}),(\boldsymbol{x}',\boldsymbol{y}')\sim\mathbb{P}_{X,Y}}\left[k(\boldsymbol{x}, \boldsymbol{x}')l(\boldsymbol{y}, \boldsymbol{y}')\right] \\
& - 2\mathbb{E}_{(\boldsymbol{x},\boldsymbol{y})\sim\mathbb{P}_{X,Y},(\boldsymbol{x}',\boldsymbol{y}')\sim\mathbb{P}_{X',Y'}}\left[k(\boldsymbol{x}, \boldsymbol{x}')l(\boldsymbol{y}, \boldsymbol{y}')\right] \\
& + \mathbb{E}_{(\boldsymbol{x},\boldsymbol{y}),(\boldsymbol{x}',\boldsymbol{y}')\sim\mathbb{P}_{X',Y'}}\left[k(\boldsymbol{x}, \boldsymbol{x}')l(\boldsymbol{y}, \boldsymbol{y}')\right].
\end{aligned}$$

This expression can be estimated straightforwardly using U-statistics or V-statistics with $\mathcal{O}_p(n^{-1/2})$ convergence (Muandet et al., 2017).

#### C.3.2 JOINT BILATERAL DISTRIBUTION COMPRESSION

Let $W : \Omega \to \mathcal{Y}$ denote the random variable representing the compressed responses, then with the machinery given in the previous section, we can obtain joint analogues of the DMMD, RMMD, and EMMD:

$$\text{DJMMD}(\mathbb{P}_{X,Y}, \mathbb{P}_{\phi(Z),W}), \quad \text{RJMMD}(\mathbb{P}_{X,Y}, \mathbb{P}_{\phi(\psi(X)),Y}), \quad \text{EJMMD}(\mathbb{P}_{\psi(X),Y}, \mathbb{P}_{Z,W})$$

corresponding to the *Decoded Joint Maximum Mean Discrepancy* (DJMMD), *Reconstruction Joint Maximum Mean Discrepancy* (RJMMD), and *Encoded Joint Maximum Mean Discrepancy* (EJMMD), respectively. Then, we replace our two stage optimisation procedure with the following:

**Step 1**: Optimise the autoencoder $\phi \circ \psi$ to minimise the RJMMD. Letting the encoder and decoder be parameterised by $\boldsymbol{\alpha}$ and $\boldsymbol{\beta}$ respectively, we solve

$$\boldsymbol{\alpha}_*, \boldsymbol{\beta}_* = \underset{\boldsymbol{\alpha}, \boldsymbol{\beta} \in \Theta}{\arg\min} \quad \text{RJMMD}^2(\hat{\mathbb{P}}_{X,Y}, \hat{\mathbb{P}}_{\phi_{\boldsymbol{\beta}}(\psi_{\boldsymbol{\alpha}}(X)),Y})$$

via minibatch gradient descent on $\boldsymbol{\alpha}$ and $\boldsymbol{\beta}$.

**Step 2**: Given the optimal encoder $\psi_{\boldsymbol{\alpha}_*}$, optimise the compressed set $\mathcal{C} \subset \mathbb{R}^p$ to minimise the EJMMD. That is, we solve

$$\mathcal{C}^* = \underset{\mathcal{C} \subset \mathbb{R}^p \times \mathcal{Y}}{\arg\min} \quad \text{EJMMD}^2(\hat{\mathbb{P}}_{\psi_{\boldsymbol{\alpha}_*}(X),Y}, \hat{\mathbb{P}}_{Z,W})$$

via gradient descent on $\mathcal{C}$. If gradient descent in the response space is not directly applicable—such as in multi-class classification, one can apply a one-hot encoding to the class labels, perform optimisation in this continuous representation, and then cluster the results back onto the space of valid one-hot encodings.

Note that joint analogues of Theorems B.1 and B.2 follow directly:

**Corollary C.1** *Given $k : \mathcal{X} \times \mathcal{X} \to \mathbb{R}$, $h : \mathbb{R}^p \times \mathbb{R}^p \to \mathbb{R}$, and $l : \mathcal{Y} \times \mathcal{Y} \to \mathbb{R}$, ket $r^d((\boldsymbol{x}, \boldsymbol{y}), (\boldsymbol{x}', \boldsymbol{y}')) = k(\boldsymbol{x}, \boldsymbol{x}') \, l(\boldsymbol{y}, \boldsymbol{y}')$ and $r^p((\boldsymbol{z}, \boldsymbol{y}), (\boldsymbol{z}', \boldsymbol{y}')) = h(\boldsymbol{z}, \boldsymbol{z}') \, l(\boldsymbol{y}, \boldsymbol{y}')$ be characteristic tensor product kernels. Further, assume $\phi \circ \psi : \mathcal{X} \to \mathcal{X}$ is an autoencoder defined by the composition of measurable maps $\psi : \mathcal{X} \to \mathbb{R}^p$ and $\phi : \mathbb{R}^p \to \mathcal{X}$. Suppose $\mathbb{P}_{X,Y}$ and $\mathbb{P}_{Z,W}$ are distributions such that*

$$\text{RJMMD}_{r^d}\left(\mathbb{P}_{X,Y}, \mathbb{P}_{\phi(\psi(X)),Y}\right) = 0 \quad \text{and} \quad \text{EJMMD}_{r^p}\left(\mathbb{P}_{\psi(X),Y}, \mathbb{P}_{Z,W}\right) = 0.$$

*Then it follows that $\text{DJMMD}_{r^d}\left(\mathbb{P}_{X,Y}, \mathbb{P}_{\phi(Z),W}\right) = 0$.*

**Corollary C.2** *Let $k : \mathcal{X} \times \mathcal{X} \to \mathbb{R}$ and $l : \mathcal{Y} \times \mathcal{Y} \to \mathbb{R}$ be positive definite kernels, and let $\psi : \mathcal{X} \to \mathbb{R}^p$, $\phi : \mathbb{R}^p \to \mathcal{X}$ be measurable. Given the tensor product kernels*

$$r^p((\boldsymbol{z}, \boldsymbol{y}), (\boldsymbol{z}', \boldsymbol{y}')) = k(\phi(\boldsymbol{z}), \phi(\boldsymbol{z}')) \, l(\boldsymbol{y}, \boldsymbol{y}'), \quad r^d((\boldsymbol{x}, \boldsymbol{y}), (\boldsymbol{x}', \boldsymbol{y}')) = k(\boldsymbol{x}, \boldsymbol{x}') \, l(\boldsymbol{y}, \boldsymbol{y}')$$

*we have that, for any probability measures $\mathbb{P}_{X,Y}$ and $\mathbb{P}_{Z,W}$,*

$$\text{DJMMD}_{r^d}\left(\mathbb{P}_{X,Y}, \mathbb{P}_{\phi(Z),W}\right) \leq \text{RJMMD}_{r^d}\left(\mathbb{P}_{X,Y}, \mathbb{P}_{\phi(\psi(X)),Y}\right) + \text{EJMMD}_{r^p}\left(\mathbb{P}_{\psi(X),Y}, \mathbb{P}_{Z,W}\right).$$

### C.4 COMPLEXITY ANALYSIS

In this section we will provide a thorough analysis of the computational complexity of the proposed approach, both in terms of time and storage.

We first observe that for commonly used kernels such as the Gaussian, IMQ, and Laplace kernels, evaluation cost is linear in the size of the input dimension. That is, for $\boldsymbol{x}, \boldsymbol{x}' \in \mathbb{R}^d$, computing $k(\boldsymbol{x}, \boldsymbol{x}')$ costs $\mathcal{O}(d)$. We assume this holds throughout the remainder of this section. [2]

We also denote the cost of evaluating the autoencoder by $\mathcal{O}(c)$. For a linear autoencoder this cost is exactly $\mathcal{O}(dp)$. For neural autoencoders, the cost depends entirely on the chosen architecture and is less straightforward to express in closed form. For example, for a fully connected autoencoder with encoder layer widths $h_0 = d, h_1, \ldots, h_E = p$ and decoder layer widths $g_0 = p, g_1, \ldots, g_D = d$, the total evaluation cost is

$$\sum_{i=1}^{E} h_{i-1} h_i + \sum_{i=1}^{D} g_{i-1} g_i = \mathcal{O}(dh_1 + h_{E-1}p + pg_1 + g_{D-1}d) = \mathcal{O}(d + p)$$

$$=: \mathcal{O}(c).$$

With this stated, we can now discuss the complexity of the two stage approach.

---

[2]Note that if one wishes to use the pull-back kernel from Theorem 3.5, one must account for the additional cost of querying the decoder, as well as the resulting higher cost of computing in the ambient dimension, rather than the latent dimension.

**Autoencoder Training**: We train the autoencoder using minibatch gradient descent targeting the RMMD loss, which dominates the computational cost. Evaluating the loss for a batch of size $B$ requires

$$\mathcal{O}(B^2(d+c)).$$

Training with epochs requires approximately $\frac{n}{B}$ batches per epoch, giving a per-epoch cost of

$$\mathcal{O}(nB(d+c)).$$

Assuming $E$ epochs, the total training cost is

$$\mathcal{O}(EnB(d+c)),$$

which is linear in both the dataset size $n$ and the input dimension $d$. The batch size is often chosen to be constant, typically the largest that fits into available GPU memory.

**Compressed Set Training**: Once the autoencoder has been trained, we optimise a compressed set in latent space $\mathbb{R}^p$, targeting the EMMD. Since the first term of the EMMD is invariant with respect to the compressed set $\mathcal{C}$, the total computational cost for $T$ gradient steps is

$$\mathcal{O}(Tp(nm+m^2)).$$

**Overall Computational Cost**: Combining the above, for ambient dimension $d$, latent dimension $p$, dataset size $n$, compressed set size $m$, batch size $B$, number of autoencoder epochs $E$, number of compressed set gradient steps $T$, and autoencoder evaluation cost $c$, the overall cost is

$$\mathcal{O}(EnB(d+c) + Tp(nm+m^2)) = \mathcal{O}(n(EBd + EBc + Tmp + Tm^2p)),$$

which is linear in the dataset size $n$ and the input dimension $d$. Note that by ignoring constants $E, B, T$ and noting $p \ll d$ and $m \ll n$, we have that BDC is overall $\mathcal{O}(nd)$.

**Overall Memory Cost**: The dominant memory cost in autoencoder training arises from storing the gradients of the kernel Gram matrices in the RMMD computation, which scales as $\mathcal{O}(B^2d)$. For distribution compression in latent space, the main cost comes from storing the gradients of the kernel Gram matrices in the EMMD computation, scaling as $\mathcal{O}(mnp)$. Thus, the overall storage complexity is

$$\mathcal{O}(B^2d + mnp),$$

which grows linearly with both the dataset size $n$ and the input dimension $d$. One must also consider the cost of storing the autoencoder parameters during optimisation, however this will heavily depend on the specific architecture considered, so we omit it from discussion here.

## C.5 PSEUDOCODE

In this section we give pseudocode for the unlabelled and labelled variants of BDC, we present them with standard stochastic gradient descent, however in practice any gradient descent method implemented in Optax (DeepMind et al., 2020) may be used, we use ADAM for our experiments. Algorithm 1 and 2 give the pseudocode for BDC-NL and BDC-L respectively.

---

**Algorithm 1** Bilateral Distribution Compression with Nonlinear Autoencoder (BDC-NL)

---

**Input**: Dataset $\mathcal{D}$; coreset size $m \in \mathbb{N}$; latent dimension $p \in \mathbb{N}$; reconstruction kernel $k : \mathcal{X} \times \mathcal{X} \to \mathbb{R}$; compression kernel $h : \mathbb{R}^p \times \mathbb{R}^p \to \mathbb{R}$; response kernel $l : \mathcal{Y} \times \mathcal{Y} \to \mathbb{R}$ (if applicable); nonlinear autoencoder $\phi_{\boldsymbol{\beta}} \circ \psi_{\boldsymbol{\alpha}}$; number of initial sets $C \in \mathbb{N}$; batch size $B \in \mathbb{N}$; max epochs $E$; max compression steps $T$; autoencoder step size $\eta$; coreset step size $\nu$

————————————— *Step 1: Optimise the autoencoder* —————————————-
**for** $e = 1$ **to** $E$ **do**
   Shuffle $\mathcal{D}$
   **for** each batch $\mathcal{B}$ of size $B$ from $\mathcal{D}$ **do**
      Compute reconstruction loss $\mathcal{L}_{\text{RMMD+MSRE}}$ using (5)
      Update autoencoder parameters $\boldsymbol{\theta} \leftarrow \boldsymbol{\theta} - \eta \nabla_{\boldsymbol{\theta}} \mathcal{L}_{\text{RMMD+MSRE}}$
   **end for**
**end for**
————————————— *Step 2: Optimise the compressed set* —————————————
Encode $\mathcal{D}$ into latent space using the trained encoder, denote as $\mathcal{Z} \subset \mathbb{R}^p$
**for** $j = 1$ **to** $C$ **do**
   Randomly sample $\mathcal{C}_j \subseteq \mathcal{Z}$ of size $m$
   Compute Encoded MMD $\mathcal{L}_{\text{EMMD},j}$ via (4)
**end for**
Initialise $\mathcal{C} \leftarrow \mathcal{C}_{j_*}$ where $j_* = \arg\min_j \mathcal{L}_{\text{EMMD},j}$
**for** $t = 1$ **to** $T$ **do**
   Compute Encoded MMD $\mathcal{L}_{\text{EMMD}}$ via (4)
   Update $\mathcal{C} \leftarrow \mathcal{C} - \nu \nabla_{\mathcal{C}} \mathcal{L}_{\text{EMMD}}$
**end for**
**Output**: compressed set $\mathcal{C}$

---

---

**Algorithm 2** Bilateral Distribution Compression with Linear Autoencoder (BDC-L)

---

**Input**: Dataset $\mathcal{D}$; coreset size $m \in \mathbb{N}$; latent dimension $p \in \mathbb{N}$; reconstruction kernel $k : \mathcal{X} \times \mathcal{X} \to \mathbb{R}$; compression kernel $h : \mathbb{R}^p \times \mathbb{R}^p \to \mathbb{R}$; response kernel $l : \mathcal{Y} \times \mathcal{Y} \to \mathbb{R}$ (if applicable); number of initial sets $C \in \mathbb{N}$; batch size $B \in \mathbb{N}$; max epochs $E$; max compression steps $T$; projection step size $\eta$; coreset step size $\nu$

————————————— *Step 1: Optimise the linear projection* —————————————-
Initialise encoder matrix $V \in \mathbb{R}^{d \times p}$ with $\mathcal{N}(0, 1/d)$ draws
Define decoder as $V^\top$, so the autoencoder is the idempotent projection $\boldsymbol{x} \mapsto VV^\top \boldsymbol{x}$
**for** $e = 1$ **to** $E$ **do**
   Shuffle $\mathcal{D}$
   **for** each batch $\mathcal{B}$ of size $B$ from $\mathcal{D}$ **do**
      Compute Reconstruction MMD $\mathcal{L}_{\text{RMMD}}$ using (2)
      Compute Euclidean gradient $G = \nabla_V \mathcal{L}_{\text{RMMD}}$
      Project $G$ onto the tangent space of the Stiefel manifold at $W$
      Update $V \leftarrow \text{Retr}(V - \eta G)$, where $\text{Retr}$ is a retraction onto the Stiefel manifold
   **end for**
**end for**
————————————— *Step 2: Optimise the compressed set* —————————————
Encode $\mathcal{D}$ into latent space using the trained encoder, denote as $\mathcal{Z} \subset \mathbb{R}^p$
**for** $j = 1$ **to** $C$ **do**
   Randomly sample $\mathcal{C}_j \subseteq \mathcal{Z}$ of size $m$
   Compute EMMD loss $\mathcal{L}_{\text{EMMD},j}$ via (4)
**end for**
Initialise $\mathcal{C} \leftarrow \mathcal{C}_{j_*}$ where $j_* = \arg\min_j \mathcal{L}_{\text{EMMD},j}$
**for** $t = 1$ **to** $T$ **do**
   Compute EMMD loss $\mathcal{L}_{\text{EMMD}}$ via (4)
   Update $\mathcal{C} \leftarrow \mathcal{C} - \nu \nabla_{\mathcal{C}} \mathcal{L}_{\text{EMMD}}$
**end for**
**Output**: compressed set $\mathcal{C}$

---

## D  EXPERIMENT DETAILS

All the experiments were performed on a single NVIDIA GTX 4070 Ti with 12GB of VRAM, CUDA 12.2 with driver 535.183.01 and JAX version 0.4.35.

As is ubiquitous in kernel methods, we standardise the features and responses such that each dimension has zero mean and unit standard deviation.

Unless otherwise stated, the kernel functions $k : \mathbb{R}^d \times \mathbb{R}^d \to \mathbb{R}, h : \mathbb{R}^p \times \mathbb{R}^p \to \mathbb{R}$, and $l : \mathcal{Y} \times \mathcal{Y} \to \mathbb{R}$ are chosen to be the Gaussian kernel, defined as

$$k(\boldsymbol{x}, \boldsymbol{x}') := \exp\left(-\frac{1}{2\alpha_k^2}\|\boldsymbol{x} - \boldsymbol{x}'\|_2^2\right), \quad h(\boldsymbol{z}, \boldsymbol{z}') := \exp\left(-\frac{1}{2\alpha_h^2}\|\boldsymbol{z} - \boldsymbol{z}'\|_2^2\right),$$

$$l(\boldsymbol{y}, \boldsymbol{y}') := \exp\left(-\frac{1}{2\alpha_l^2}\|\boldsymbol{y} - \boldsymbol{y}'\|_2^2\right),$$

where the lengthscales $\alpha_k, \alpha_h, \alpha_l > 0$ are set via the median heuristic on the relevant space. That is, given a dataset $\{\boldsymbol{z}_i\}_{i=1}^m$, the median heuristic is defined to be

$$H_m := \mathrm{Med}\left\{\|\boldsymbol{z}_i - \boldsymbol{z}_j\|_2^2 \ : \ 1 \le i \le j \le p\right\}$$

with lengthscale $\alpha := \sqrt{H_m/2}$ such that $h(\boldsymbol{z}, \boldsymbol{z}') := \exp\left(-\frac{1}{H_m}\|\boldsymbol{z} - \boldsymbol{z}'\|^2\right)$. This heuristic is a very widely used default choice that has shown strong empirical performance (Garreau et al., 2018).

For discrete response distributions $\mathbb{P}_Y$, gradient descent in the response space is not directly applicable, for example, in multi-class classification. In such cases, one may apply a one-hot encoding to the class labels, perform optimisation in this continuous representation, and then project the results back onto the space of valid one-hot encodings via clustering.

For all experiments, we use the Adam optimiser (Kingma & Ba, 2017) as a sensible default choice. However, the implementations allow for an arbitrary choice of optimiser via the Optax package (DeepMind et al., 2020).

To obtain reasonable seeds for latent-space minimisation of the compressed set, we follow a similar approach to Kernel Herding (Chen et al., 2010): we draw several auxiliary sets of size $m$ as uniform random subsamples of the embedded original data, and select as the initial seed the set that yields the smallest EMMD.

We now provide the full details for each of the experiments presented in the main body of this work, for additional experiments see Section D.1:

**Gaussian Mixture**: We generate $n = 20{,}000$ points from the Gaussian-mixture distribution given by $\mathbb{P}_X = \frac{1}{9}\sum_{i=1}^9 \mathcal{N}(\boldsymbol{\mu}_i, \sigma_i)$ with

$$\boldsymbol{\mu}_1 = [0, 0]^\top, \quad \boldsymbol{\mu}_2 = [1, 1]^\top \quad \boldsymbol{\mu}_3 = [-1, -1]^\top \quad \boldsymbol{\mu}_4 = [-1, 1]^\top \quad \boldsymbol{\mu}_5 = [1, -1]^\top$$

$$\boldsymbol{\mu}_6 = [2, 0]^\top, \quad \boldsymbol{\mu}_7 = [-2, 0]^\top \quad \boldsymbol{\mu}_8 = [0, 2]^\top \quad \boldsymbol{\mu}_9 = [0, -2]^\top$$

$$\Sigma_1 = \Sigma_2 = \Sigma_3 = \Sigma_4 = \Sigma_5 = \Sigma_6 = \Sigma_7 = \Sigma_8 = \Sigma_9 = I_2.$$

Compressed sets are constructed at sizes $m \in \{2, 5, 9, 25, 50, 75, 100, 150, 200\}$. We reduce from $d = 250$ ambient features to a latent dimension of $p = 2$. The linear autoencoder is an idempotent rank-2 projection matrix, initialised with random Gaussian draws and trained via Stiefel manifold optimisation. Training is performed for $E = 3$ epochs with a batch size of $B = 64$. For each compressed set, we run at most 3,000 iterations of gradient descent, terminating early when the gradient norm falls below $10^{-7}$. Optimisation uses ADAM with a learning rate of $10^{-1}$. To compute the RMMD we employ a Gaussian kernel, with lengthscales set by the median heuristic based on 1,000 samples from the relevant distribution, while the EMMD is estimated using the pull-back kernel. Exact computation of the DMMD is possible via the analytical formulae derived in Section 4.1.

**Swiss-Roll Manifold**: We generate $n = 20{,}000$ training points and an additional $n_t = 1{,}000$ test points from the Swiss-roll manifold distribution. Compressed sets are constructed at size $m = 200$. We compress to latent dimension of $p = 3$ from $d = 200$ ambient features representing

a total 99.99% compression. The linear autoencoder is an idempotent rank-3 projection matrix trained via Stiefel manifold optimisation, and initialised with random Gaussian draws, while the nonlinear autoencoder consists of a two-hidden-layer feed-forward encoder with ReLU activations, paired with a two-hidden-layer feed-forward decoder (non–weight-tied) also using ReLU. The linear autoencoder is trained for $E = 5$ epochs, while the nonlinear autoencoder is trained for $E = 10$ epochs, both use batch size $B = 64$. We do a maximum of 5,000 iterations of gradient descent on the compressed set, with convergence assessed by the gradient norm, stopping when it falls below $10^{-8}$. All gradient descent is performed using ADAM with a learning rate of $10^{-2}$. Gaussian kernels are used throughout, with lengthscales set by the median heuristic, estimated from 1,000 samples of the relevant distribution.

**CT Slice Dataset**: We use $n = 50,000$ training points with $n_t = 3,500$ held-out test points. Compressed sets are constructed at size $m = 200$. We compress to latent dimension of $p = 27$ from $d = 384$ ambient features representing a total 99.97% compression. The linear autoencoder is an idempotent rank-27 projection matrix, trained via Stiefel manifold optimisation and initialised using PCA. The nonlinear autoencoder consists of a one-hidden-layer feed-forward encoder with ReLU activations, paired with a one-hidden-layer non–weight-tied decoder, also with ReLU activations. The linear autoencoder is trained for $E = 10$ epochs, while the nonlinear autoencoder is trained for $E = 50$ epochs, both with a batch size of $B = 1024$. For compressed set optimisation, we perform at most 5,000 iterations of gradient descent, with early stopping when the gradient norm falls below $10^{-8}$. Gradient descent on the compressed set is carried out using ADAM with learning rate $10^{-2}$, while autoencoder training uses learning rate $10^{-3}$. Gaussian kernels are employed throughout, with lengthscales set via the median heuristic estimated from 1,000 samples of the relevant distribution.

**MNIST Dataset**: We use $n = 60,000$ training points with $n_t = 10,000$ held-out test points. Compressed sets are constructed at size $m = 200$. We compress to latent dimension of $p = 28$ from $d = 784$ ambient features representing a total 99.99% compression. The linear autoencoder is an idempotent rank-28 projection matrix, trained via Stiefel manifold optimisation and initialised using PCA. The nonlinear autoencoder consists of a one-hidden-layer feed-forward encoder with ReLU activations, paired with a one-hidden-layer non–weight-tied decoder, also with ReLU activations. The linear autoencoder is trained for $E = 10$ epochs, while the nonlinear autoencoder is trained for $E = 50$ epochs, both with a batch size of $B = 1024$. For compressed set optimisation, we perform at most 2,500 iterations of gradient descent, with early stopping when the gradient norm falls below $10^{-8}$. Gradient descent on the compressed set is carried out using ADAM with learning rate $10^{-2}$, while autoencoder training uses learning rate $10^{-3}$. We implement M3D following Algorithm 1 of Zhang et al. (2024). M3D Training uses a batch size of 128 from the full dataset and the entire corresponding class of the compressed set. We perform five iterations for each of 500 randomly sampled encoders, all chosen to match the BDC architecture, namely a one-hidden-layer neural network with ReLU activations. We initialise the M3D compressed set with a random subsample of the dataset, as in the other methods. Gaussian kernels are employed throughout, including in M3D, with lengthscales set via the median heuristic estimated from 1,000 samples of the relevant distribution.

**Clusters**: We use $n = 102,000$ training points, $n_e = 2,000$ of which is uniform noise, with $n_t = 20,000$ held-out test points. Compressed sets are constructed at size $m = 300$, with latent dimensions $p = 2$ (BDC-NL) and $p = 5$ (BDC-L), corresponding to 99.999% and 99.997% compression from $d = 500$ ambient features. The linear autoencoder is an idempotent rank-5 projection matrix, trained via Stiefel manifold optimisation and initialised with random Gaussian draws. The nonlinear autoencoder is a one-hidden-layer feed-forward encoder with ReLU activations, paired with a non–weight-tied one-hidden-layer decoder, also with ReLU activations. The linear autoencoder is trained for $E = 10$ epochs, while the nonlinear autoencoder is trained for $E = 50$ epochs, both with a batch size of $B = 1024$. For compressed set optimisation, we run up to 2,500 iterations of gradient descent with ADAM (learning rate $10^{-3}$), applying early stopping when the gradient norm falls below $10^{-8}$. We reduce this to 500 iterations for ADC due to slowness. Gaussian kernels are used throughout: RMMD lengthscales are set via the median heuristic on 1,000 samples, while EMMD lengthscales are taken as one-tenth of the median heuristic to emphasise local differences.

## D.1 Additional Experiments

In this section we include the results from any additional experiments.

| Dataset | Method | | | | |
|---|---|---|---|---|---|
| | FULL | M3D | ADC | BDC-NL | BDC-L |
| *Swiss-Roll-NL* | 944.1 | NA | $197.65 \pm 1.88$ | $9.54 \pm 0.86$ | $8.59 \pm 0.23$ |
| *Swiss-Roll-L* | 734.32 | NA | $197.30 \pm 1.20$ | $8.22 \pm 0.57$ | $7.33 \pm 0.17$ |
| *Swiss-Roll-L-IMQ* | 751.70 | NA | $184.10 \pm 12.58$ | $9.24 \pm 0.51$ | $8.48 \pm 0.18$ |
| *Wave* | NA | NA | $199.74 \pm 0.31$ | $41.27 \pm 0.97$ | $25.82 \pm 0.20$ |
| *Buzz* | NA | NA | $238.68 \pm 3.82$ | $54.62 \pm 2.14$ | $35.32 \pm 0.79$ |
| *CT-Slice* | NA | NA | $776.86 \pm 30.47$ | $117.81 \pm 4.31$ | $57.10 \pm 1.90$ |
| *MNIST* | NA | $827.34 \pm 27.99$ | $1003.62 \pm 1.80$ | $206.65 \pm 3.01$ | $72.26 \pm 2.29$ |
| *Clusters* | 1800.62 | NA | $691.00 \pm 26.50$ | $84.91 \pm 2.84$ | $46.74 \pm 1.82$ |

Table 2: Overall time comparison (construction of the compressed set plus downstream task fitting) in seconds, across datasets, reported as mean and standard deviation.

### D.1.1    ADDITIONAL REGRESSION EXPERIMENT: SWISS-ROLL WITH LINEAR PROJECTION

In Section 4.2.1, we applied BDC to data on a $p = 3$ dimensional Swiss-roll manifold projected into $d = 200$ dimensions using a random nonlinear map. Here, we retain the same setup but replace the nonlinear projection to ambient dimension with a linear one $V \in \mathbb{R}^{200 \times 3}$ with entries drawn i.i.d. from $\mathcal{N}(0, 1)$. In addition, we reduce the nonlinear autoencoder to a single hidden layer in both encoder and decoder, and train both the linear and nonlinear autoencoders for only $E = 3$ epochs. As shown in Figure 10, BDC with a linear autoencoder (BDC-L) successfully recovers the manifold, as expected. Moreover, BDC with a nonlinear autoencoder (BDC-NL) is likewise able to recover this linear embedding. We note that performance is comparable to ADC, but achieved at substantially lower cost, as shown in Table 2.

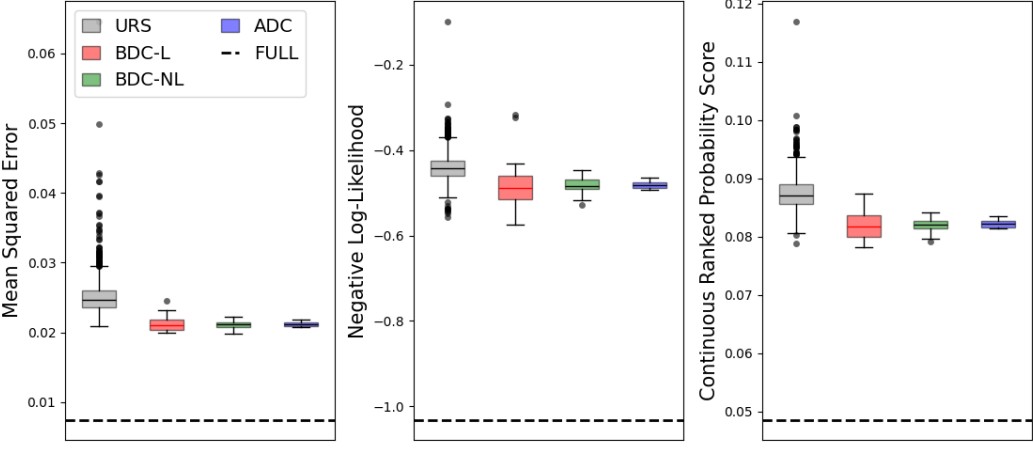

Figure 10: Test performance on the *Swiss-Roll* dataset linearly projected to $d = 200$ dimensions. BDC with a nonlinear autoencoder (BDC-NL, green) and a linear autoencoder (BDC-L, red) are compared against ADC (blue), each over 25 runs. URS (grey) over 1,000 runs, and FULL is shown as a black dashed line.

### D.1.2    ADDITIONAL REGRESSION EXPERIMENT: SWISS-ROLL WITH LINEAR PROJECTION AND IMQ KERNEL

In our experiments, we have mostly used the Gaussian kernel, as it is a universally standard choice in the kernel embedding literature, and in kernel-based methods more broadly. However, we emphasise that our algorithms are kernel-agnostic: any characteristic feature and response kernels may be used, provided their gradients are computable. As with any kernel method, the optimal kernel choice is unknown and problem-specific, hence it should be selected with care by the practitioner to align

with their inductive biases. The Gaussian kernel is a very common default choice due to its ability to capture smooth similarities.

In this section, we revisit the linear *Swiss-Roll* dataset from Section D.1.1, this time using the inverse multi-quadratic (IMQ) kernel instead of the Gaussian kernel for distribution compression. IMQ kernels are characteristic (Weinstein & Miller, 2021) and take the form:

$$k(\boldsymbol{x}, \boldsymbol{x}') := \left(1 + \frac{||\boldsymbol{x} - \boldsymbol{x}'||^2}{2\lambda^2}\right)^{-1/2}.$$

Figure 11 shows that performance of our BDC framework is still strong, despite the change in kernel.

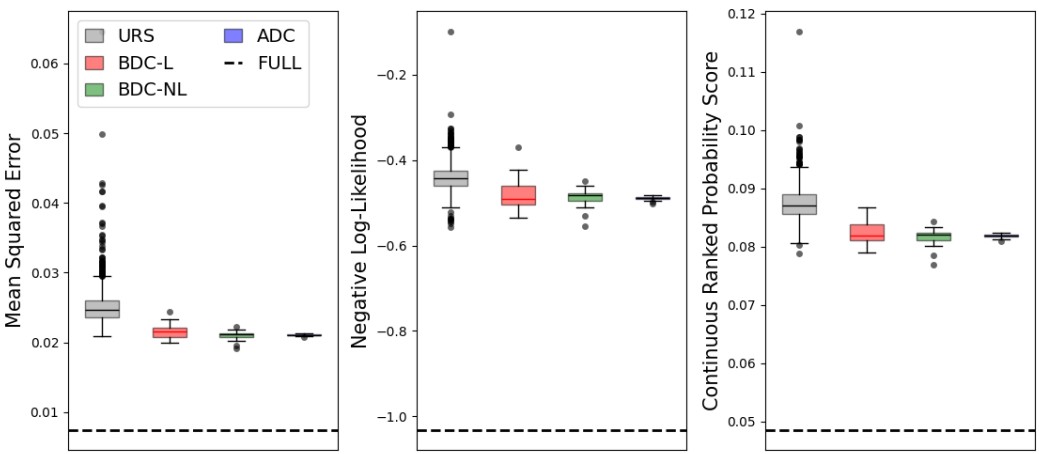

Figure 11: Test performance on the *Swiss-Roll* dataset linearly projected to $d = 200$ dimensions. IMQ kernels are used for distribution compression. BDC with a nonlinear autoencoder (BDC-NL, green) and a linear autoencoder (BDC-L, red) are compared against ADC (blue), each over 25 runs. URS (grey) over 1,000 runs, and FULL is shown as a black dashed line.

### D.1.3 ADDITIONAL REGRESSION EXPERIMENT: WAVE ENERGY PREDICTION

The *Wave* dataset (Neshat et al., 2020), consists of $n = 36,043$ instances where each configuration is described by $d = 98$ spatial features (the $x$ and $y$ coordinates of 49 wave energy converters). The regression target is the total power output, making the task one of predicting energy yield directly from device locations.

Compressed sets are constructed at size $m = 200$. We project to a $p = 16$ dimensional latent space, representing a total compression of 99.83%. The linear autoencoder is an idempotent rank-16 projection matrix, trained via Stiefel manifold optimisation and initialised using PCA. The nonlinear autoencoder consists of a one-hidden-layer feed-forward encoder with ReLU activations, paired with a one-hidden-layer non–weight-tied decoder, also with ReLU activations. The linear autoencoder is trained for $E = 10$ epochs, while the nonlinear autoencoder is trained for $E = 50$ epochs, both with a batch size of $B = 1024$. For compressed set optimisation, we perform at most 5,000 iterations of gradient descent, with early stopping when the gradient norm falls below $10^{-8}$. Gradient descent on the compressed set is carried out using ADAM with learning rate $10^{-2}$, while autoencoder training uses learning rate $10^{-3}$. Gaussian kernels are employed throughout, with lengthscales set via the median heuristic estimated from 1,000 samples of the relevant distribution.

Results are shown in Figure 12. While ADC outperforms BDC on predictive metrics, BDC-L achieves competitive accuracy at substantially lower runtime. URS performs poorly—particularly on NLL—showing both low accuracy and high variability. Interestingly, BDC-L outperforms BDC-NL, suggesting either that the neural architecture is suboptimal for this task or that the data are closer to a linear structure than the nonlinear model can exploit. Overall, strong performance appears to require access to the full ambient dimension, which may explain the advantage of ADC and the limited evidence for manifold structure. Due to computational constraints, we were unable to train a GP on the full dataset.

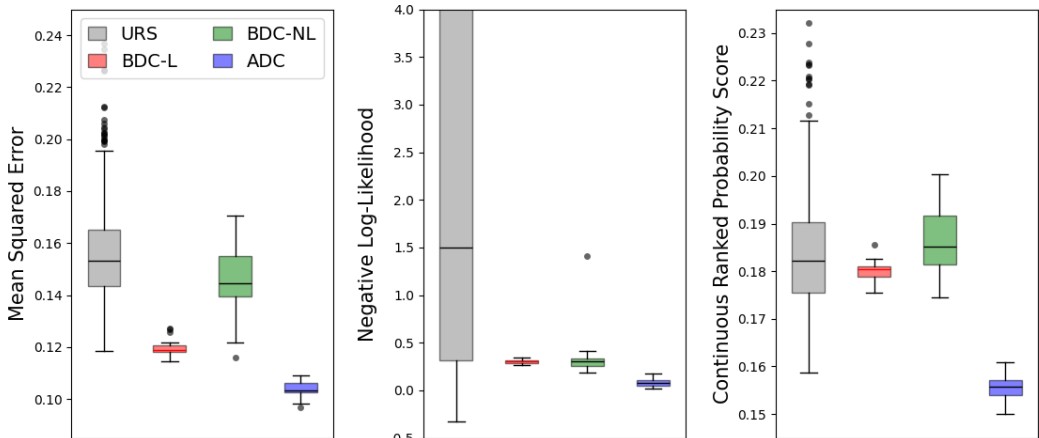

Figure 12: Test performance on the *Wave* dataset. BDC with a nonlinear autoencoder (BDC-NL, green) and a linear autoencoder (BDC-L, red) are compared against ADC (blue), each over 25 runs. URS (grey) over 1,000 runs, and FULL is shown as a black dashed line.

### D.1.4 ADDITIONAL REGRESSION EXPERIMENT: SOCIAL MEDIA BUZZ PREDICTION

The *Buzz* dataset (Kawala et al., 2013), consists of $n = 583250$ instances where each configuration is described by $d = 77$ temporal features describing the evolution of social media activity on specific topics. The prediction task is regression, with the target being the mean number of active discussions (NAD) over the two weeks following the observation window. We subsample down to $n = 50,000$ data points due to memory limitations, as experiments were performed on a single GPU with access to just 12GB of VRAM, however we note that memory scales linearly with $n$ and $d$ (see Section C.4)

Compressed sets are constructed at size $m = 200$. We project to a $p = 17$ dimensional latent space, representing a total compression of $99.91\%$. The linear autoencoder is an idempotent rank-17 projection matrix, trained via Stiefel manifold optimisation and initialised using PCA. The nonlinear autoencoder consists of a one-hidden-layer feed-forward encoder with ReLU activations, paired with a one-hidden-layer non–weight-tied decoder, also with ReLU activations. The linear autoencoder is trained for $E = 10$ epochs, while the nonlinear autoencoder is trained for $E = 50$ epochs, both with a batch size of $B = 1024$. For compressed set optimisation, we perform at most 5,000 iterations of gradient descent, with early stopping when the gradient norm falls below $10^{-8}$. Gradient descent on the compressed set is carried out using ADAM with learning rate $10^{-2}$, while autoencoder training uses learning rate $10^{-3}$. Gaussian kernels are employed throughout, with lengthscales set via the median heuristic estimated from 1,000 samples of the relevant distribution.

Results are shown in Figure 13. Both BDC variants achieve accuracy competitive with ADC at substantially lower cost, with BDC-NL performing most consistently well. This suggests the presence of manifold structure in the dataset. Due to computational constraints, we were unable to train a GP on the full dataset.

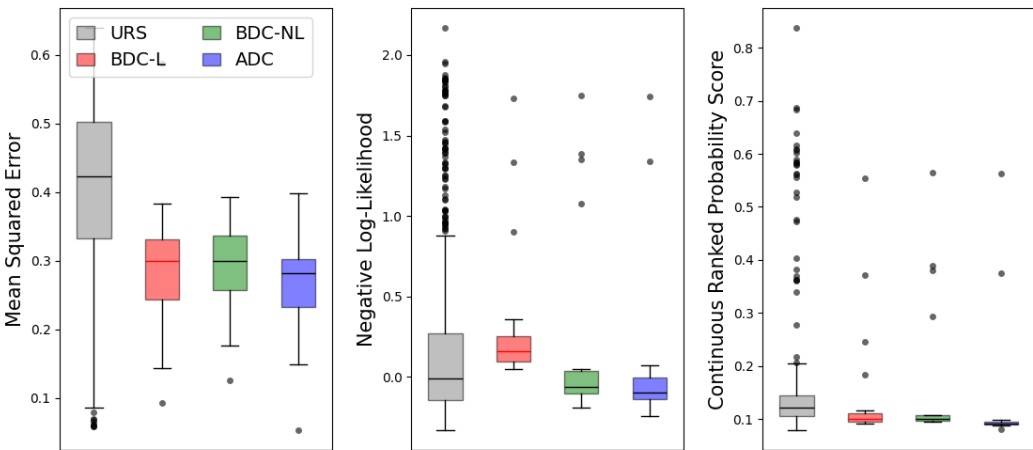

Figure 13: Test performance on the *Buzz* dataset. BDC with a nonlinear autoencoder (BDC-NL, green) and a linear autoencoder (BDC-L, red) are compared against ADC (blue), each over 25 runs. URS (grey) over 1,000 runs, and FULL is shown as a black dashed line.

### D.1.5   EFFECT OF INCREASING SIZE OF LATENT DIMENSION ON RMMD

In this section, we examine how the choice of latent dimension influences the RMMD achieved by different autoencoders. We report both the marginal RMMD between $\mathbb{P}_X$ and its reconstruction $\mathbb{P}_{\phi(\psi(X))}$, and the joint RMMD between $\mathbb{P}_{X,Y}$ and $\mathbb{P}_{\phi(\psi(X)),Y}$. Three models are compared: BDC-L (red), a linear autoencoder randomly initialised with Gaussian draws; BDC-PCA-L (orange), a linear autoencoder initialised with PCA; and BDC-NL (blue), a nonlinear autoencoder with fully connected encoder and decoder using ReLU activations. For BDC-NL, we evaluate three depths—BDC-NL-1, BDC-NL-2, and BDC-NL-3—with one, two, and three hidden layers, respectively. Nonlinear models are trained for 100 epochs and linear models for 25, both with batch size 1024. On the *Buzz* and *Wave* datasets, hidden layer sizes are 64–48–32 (three layers), 64–32 (two layers), and 64 (one layer), for *CT-Slice* we increase these to 128–96–64, and for *MNIST* increase them again to 256–128–64. As a reference, we also report the cumulative explained variance of PCA components.

Figure 14 presents results on the *CT-Slice* dataset. At small latent dimensions, the nonlinear autoencoder achieves lower RMMD than the linear autoencoder, but this advantage diminishes as the latent size increases. The single-layer neural network performs slightly worse at small dimensions but better at larger ones, suggesting that additional layers are only beneficial here under extreme compression and otherwise complicate optimisation. More advanced neural architecture or alternate optimisation strategies may be required here to see improved results. Similar behaviour is observed on the *Wave* (Figure 15), *Buzz* (Figure 16) and *MNIST* (Figure 17) datasets. For *Buzz* in particular, performance degrades with larger latent dimensions, especially in deeper networks, with the linear autoencoder outperforming at higher latent sizes. This suggests that the added complexity of the nonlinear autoencoder provides no benefit for this dataset, and that the optimal latent representation may in fact lie in a linear subspace. Across all datasets, PCA initialisation improves the optimisation of the linear autoencoder compared with random initialisation, though the effect varies: it is minimal for *CT-Slice* and *Wave* but much more pronounced for *Buzz*. Finally, marginal and joint RMMD recovery are closely related but not identical; for example, in Figure 16, BDC-L outperforms two of the deeper BDC-NL variants in joint RMMD at the largest latent size, while the opposite holds for marginal RMMD.

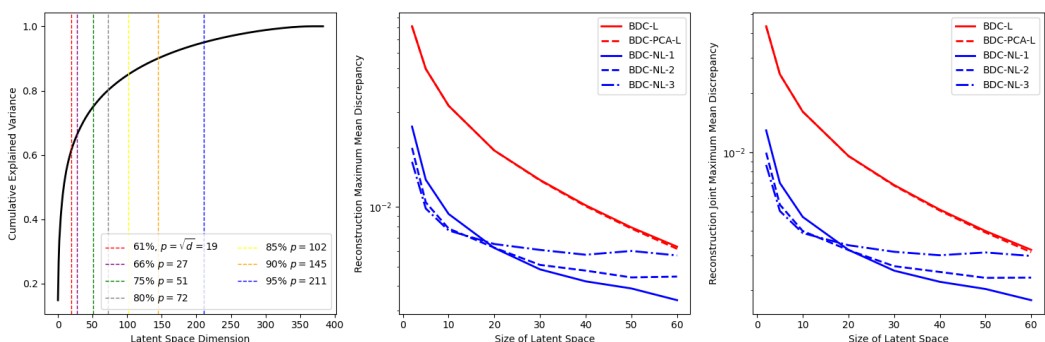

Figure 14: Compression performance versus size of latent dimension on the *CT-Slice* dataset. Left: Cumulative explained variance of PCA with annotated thresholds at 61% ($\sqrt{d}$), 66%, 75%, 80%, 85%, 90%, and 95%. Right: RMMD and RJMMD as a function of latent space dimension for BDC with a linear autoencoder (BDC-L, red), a PCA-initialised linear autoencoder (BDC-PCA-L, orange), and a nonlinear autoencoder (BDC-NL, blue).

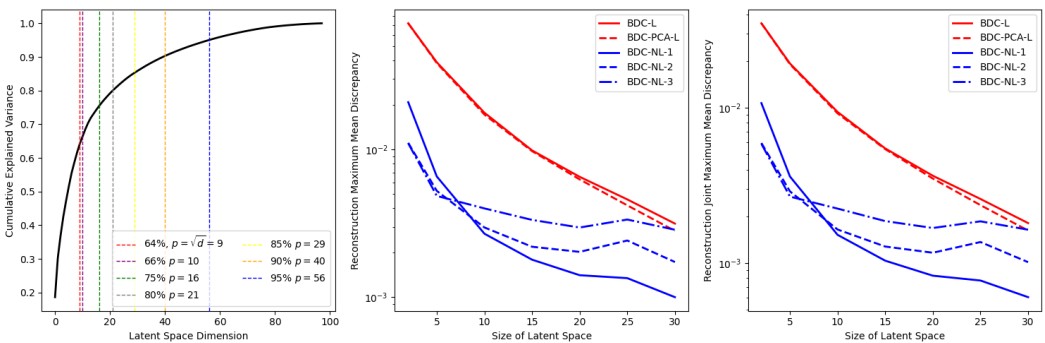

Figure 15: Compression performance versus size of latent dimension on the *Wave* dataset. Left: Cumulative explained variance of PCA with annotated thresholds at 64% ($\sqrt{d}$), 66%, 75%, 80%, 85%, 90%, and 95%. Right: RMMD and RJMMD as a function of latent space dimension for BDC with a linear autoencoder (BDC-L, red), a PCA-initialised linear autoencoder (BDC-PCA-L, orange), and a nonlinear autoencoder (BDC-NL, blue).

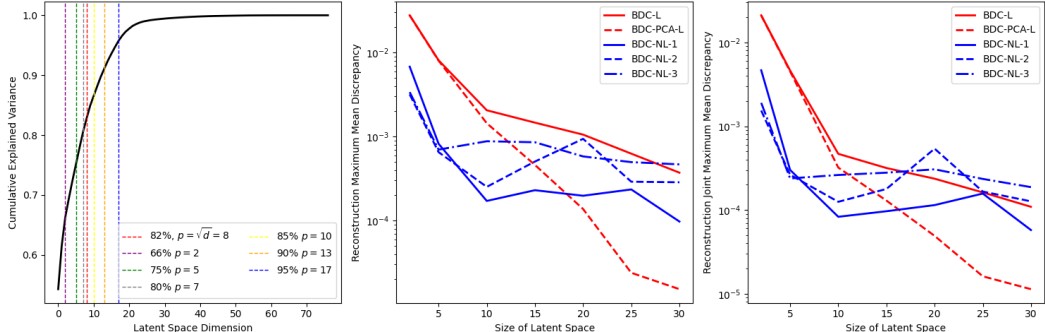

Figure 16: Compression performance versus size of latent dimension on the *Buzz* dataset. Left: Cumulative explained variance of PCA with annotated thresholds at 66%, 75%, 80%, 82% ($\sqrt{d}$), 85%, 90%, and 95%. Right: RMMD and RJMMD as a function of latent space dimension for BDC with a linear autoencoder (BDC-L, red), a PCA-initialised linear autoencoder (BDC-PCA-L, orange), and a nonlinear autoencoder (BDC-NL, blue).

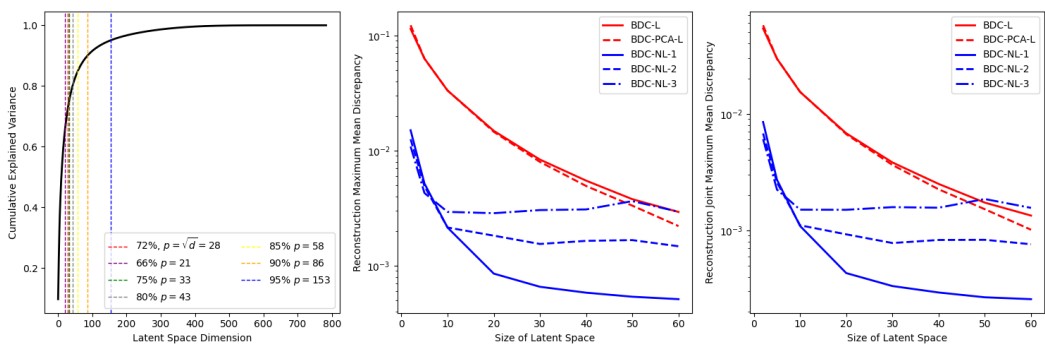

Figure 17: Compression performance versus size of latent dimension on the *MNIST* dataset. Left: Cumulative explained variance of PCA with annotated thresholds at 66%, 72% ($\sqrt{d}$),75%, 80%, 85%, 90%, and 95%. Right: RMMD and RJMMD as a function of latent space dimension for BDC with a linear autoencoder (BDC-L, red), a PCA-initialised linear autoencoder (BDC-PCA-L, orange), and a nonlinear autoencoder (BDC-NL, blue).

### D.1.6 EFFECT OF INCREASING SIZE OF LATENT DIMENSION ON CT-SLICE

In this section, we investigate how increasing the latent dimension affects GP regression performance on the *CT-Slice* dataset from Section 4.2.1. We use the same experimental setup as in Section D.1, constructing compressed sets of size $m = 200$, but now vary the latent dimension over $p \in \{5, 10, 15, 20, 25, 30, 35, 40, 45, 50\}$ from $d = 384$ ambient features.

Results are shown in Figure 18. The nonlinear autoencoder yields a more informative latent space for $p \leq 20$ compared with the linear model. However, its performance begins to degrade for $p > 10$, while for $p > 20$ the linear autoencoder appears to have stabilised. This may reflect the increased difficulty of training larger networks, or indicate that the true intrinsic dimension of the dataset lies closer to a $p \approx 10$ nonlinear embedding in $d = 384$ ambient dimensions, so that additional dimensions contribute little feature–response information while complicating GP optimisation. Another possibility is that training the compressed set itself becomes harder at higher dimensions, since we rely on gradient descent and the optimisation surface of nonlinear embeddings may become particularly challenging as the dimension increases. Computation time also increases with latent dimension, largely due to the added cost of gradIt may also be the case that more ient evaluations in higher dimensions. As reported in Table 1, ADC requires $776.86s \pm 40.47s$, which is substantially more expensive than even the largest latent dimension tested here, despite having worse performance.

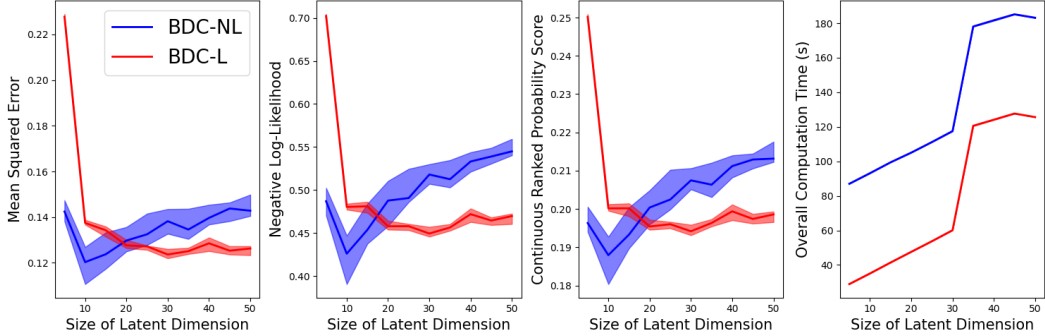

Figure 18: GP performance versus size of latent dimension on the *CT-Slice* dataset. We report BDC with a nonlinear autoencoder (BDC-NL, blue) and a linear autoencoder (BDC-L, red), each over 25 runs.

An interesting extension would be to explore more sophisticated neural autoencoders here as it may be the case that the one-hidden-layer autoencoder we have used here is not capable of making effective use of the additional latent dimensions.

### D.1.7 EFFECT OF INCREASING SIZE OF LATENT DIMENSION ON MNIST

In this section, we investigate how increasing the latent dimension affects GP classification performance on the *MNIST* dataset from Section 4.2.2. We use the same experimental setup as in Section D.1, constructing compressed sets of size $m = 200$, but now vary the latent dimension over $p \in \{2, 5, 10, 15, 20, 25, 30, 40, 50\}$ from $d = 784$ ambient features.

Results are shown in Figure 19. The nonlinear autoencoder yields a more informative latent space for $p \leq 20$ compared with the linear model. Both autoencoders appear to have their optimal performance around $p = 10$. This may indicate that the true intrinsic dimension of the dataset lies closer to a $p \approx 10$ embedding in $d = 784$ ambient space, and that additional dimensions are no longer contributing to feature–response information while complicating optimisation. We see that computation time increases with latent dimension, due to the cost of gradient evaluations in higher dimensions. As reported in Table 1, ADC and M3D require $1003.62 \pm 1.80s$ and $827.34 \pm 27.99$ respectively, which is substantially more expensive than even the largest latent dimension tested here, despite having worse performance.

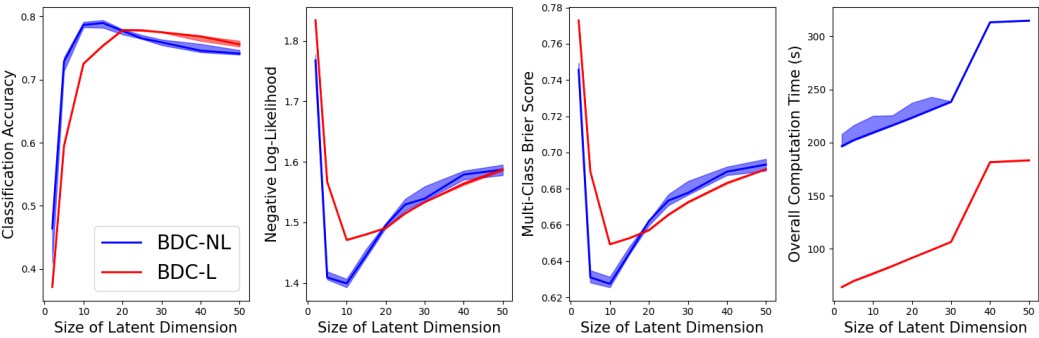

Figure 19: GP performance versus size of latent dimension on the *MNIST* dataset. We report BDC with a nonlinear autoencoder (BDC-NL, blue) and a linear autoencoder (BDC-L, red), each over 10 runs.

An interesting extension would be to explore more sophisticated neural autoencoders, such as a convolutional encoder paired with a deconvolutional decoder, which are known to yield improved performance on image tasks. It may be the case that the simple one-hidden-layer autoencoder we have used here is not capable of making use of the additional latent dimensions.

### D.2 EXACT BILATERAL DISTRIBUTION COMPRESSION

As discussed in Section 4.1, for certain distribution–kernel pairs the DMMD, RMMD, and EMMD admit closed-form expressions. In particular, Briol et al. (2025) show that for the Gaussian kernel

$$k(\boldsymbol{x}, \boldsymbol{x}') = \exp\left(-\frac{1}{2\lambda^2}\|\boldsymbol{x} - \boldsymbol{x}'\|_2^2\right),$$

and a Gaussian distribution $\mathbb{P}_X = \mathcal{N}(\boldsymbol{\mu}, \Sigma)$ with arbitrary covariance $\Sigma \in \mathbb{R}^{d \times d}$, we have

$$\mu_{\mathbb{P}_X}(\boldsymbol{x}) = \left|I + \frac{1}{\lambda^2}\Sigma\right|^{-1/2} \exp\left(-\frac{1}{2}(\boldsymbol{x} - \boldsymbol{\mu})^\top(\lambda^2 I + \Sigma)^{-1}(\boldsymbol{x} - \boldsymbol{\mu})\right),$$

$$\mathbb{E}_{\mathbb{P}_X}\left[\mu_{\mathbb{P}_X}(X)\right] = \left|I + \frac{2}{\lambda^2}\Sigma\right|^{-1/2}$$

Now, let $V \in \mathbb{R}^{d \times p}$ be a projection matrix. A standard property of Gaussian distributions is that they remain Gaussian under affine transformations. Hence, if $\mathbb{P}_X = \mathcal{N}(\boldsymbol{0}, \Sigma)$, then

$$\mathbb{P}_{XV} = \mathcal{N}(\boldsymbol{0}, V^\top \Sigma V), \quad \mathbb{P}_{XVV^\top} = \mathcal{N}(\boldsymbol{0}, VV^\top \Sigma VV^\top).$$

The above formulas therefore still apply, allowing exact computation of the DMMD, EMMD, and RMMD for any covariance $\Sigma$, projection matrix $V$, and compressed set $\mathcal{C}$.

Assume we have a Gaussian mixture distribution, i.e. $\mathbb{P}_X = \sum_i \omega_i \mathcal{N}(\boldsymbol{\mu_i}, \Sigma_i)$, $\sum_i \omega_i = 1$, $\omega_i \geq 0$, then one can show that

$$\mu_{\mathbb{P}_X}(\boldsymbol{x}) = \sum_i \omega_i \left| I + \tfrac{1}{\lambda^2}\Sigma_i \right|^{-1/2} \exp\left(-\tfrac{1}{2}(\boldsymbol{x} - \boldsymbol{\mu}_i)^\top \left(\lambda^2 I + \Sigma_i\right)^{-1}(\boldsymbol{x} - \boldsymbol{\mu}_i)\right),$$

$$\mathbb{E}_{\mathbb{P}_X}\left[\mu_{\mathbb{P}_X}(X)\right] = \sum_{i,j} \omega_i\omega_j \left| I + \frac{\Sigma_i + \Sigma_j}{\lambda^2} \right|^{-\frac{1}{2}} \exp\left(-\tfrac{1}{2}(\boldsymbol{\mu}_i - \boldsymbol{\mu}_j)^\top \left(\lambda^2 I + \Sigma_i + \Sigma_j\right)^{-1}(\boldsymbol{\mu}_i - \boldsymbol{\mu}_j)\right).$$

The affine property holds for Gaussian mixtures also, that is,

$$\mathbb{P}_{XV} = \sum_i \omega_i \mathcal{N}(\boldsymbol{0}, V^\top \Sigma_i V), \quad \mathbb{P}_{XVV^\top} = \sum_i \omega_i \mathcal{N}(\boldsymbol{0}, VV^\top \Sigma_i VV^\top),$$

and so the above formulae can be applied directly.

### D.3 DECODED MNIST IMAGES

In this section, we present a figure showing the optimised images produced by each method—ADC, BDC-L, BDC-NL, and M3D—from the image experiment described in Section 4. Both M3D and ADC optimise compressed sets directly in the ambient space of dimension $d = 784$. In contrast, BDC-L and BDC-NL operate in the latent space of an autoencoder with dimension $p = 28$. For the BDC methods, the displayed images are decoded reconstructions of the compressed sets, mapped back into the ambient space for visual inspection.

In Figure 20, the reconstructed images produced by BDC-NL are sharper than those from BDC-L, which is expected since the nonlinear autoencoder can recover more information than the linear variant. Comparing M3D and ADC, the M3D images also appear sharper and the sharpness of the M3D reconstructions is broadly comparable to that of BDC-NL. While these images are useful for visual comparison and help us reason about the different methods, it is important to note that the downstream task in BDC is performed directly in the latent space rather than in the ambient image space. Indeed, in Figure 7 we observed that BDC-L achieved the best Gaussian Process predictive metrics in the shortest time, despite producing lower-quality reconstructions here. This suggests that the latent space learned by BDC-L may be more informative about the feature–response relationship than that of BDC-NL. Moreover, both BDC methods substantially outperformed ADC and M3D.

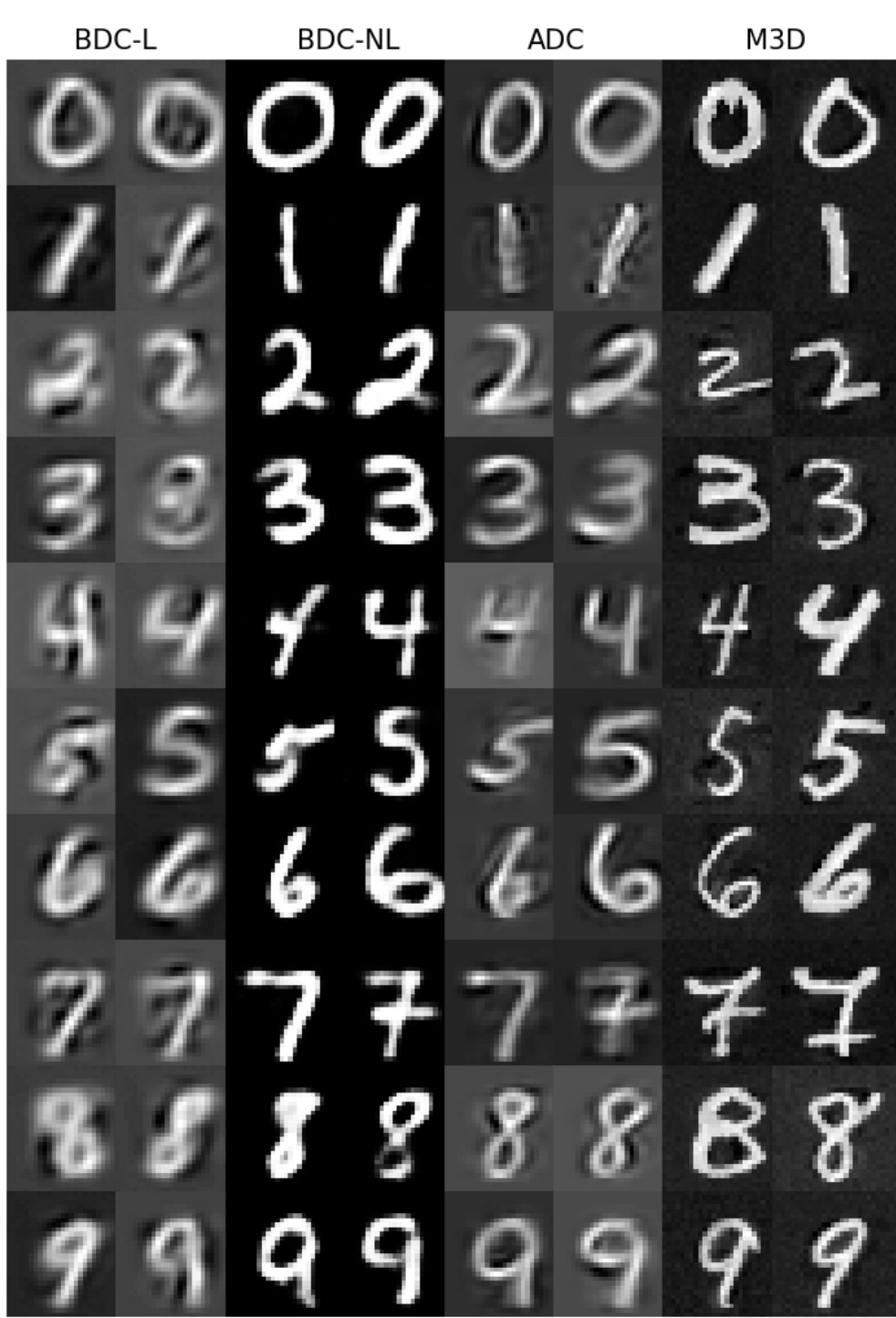

Figure 20: Example images optimised by each method from the *MNIST* experiment from Section 4.2.2. M3D and ADC optimise compressed sets directly in the ambient pixel space ($d = 784$), whereas BDC-L and BDC-NL optimise compressed sets in the latent space of dimension $p = 28$, with images shown after decoding back to the ambient space. Rows correspond to digit classes 0–9, with two representative images per class.

