# OpenReview forum: "Bilateral Distribution Compression: Reducing Both Data Size and Dimensionality"
_ICLR.cc/2026/Conference — Submitted to ICLR 2026_

### Official Review · Reviewer_KPqY · 2025-10-26

**Soundness:** 2
**Presentation:** 2
**Contribution:** 2
**Rating:** 6
**Confidence:** 2

**Summary:**

This paper introduces Bilateral Distribution Compression (BDC), a two‑stage framework to compress datasets simultaneously along the sample and feature axes. Stage 1 trains an autoencoder by minimising a Reconstruction MMD (RMMD) between the data and its reconstruction. Stage 2 constructs a size‑$m$ compressed set in the latent space by minimising an Encoded MMD (EMMD) between encoded data and the compressed set. The paper defines the target discrepancy Decoded MMD (DMMD) between the original data and the decoded compressed set. It proves that vanishing RMMD and EMMD imply vanishing DMMD and also gives a bound using a pull‑back kernel . Empirically, BDC (linear and nonlinear variants) is evaluated on synthetic and real datasets for regression, classification, and clustering, reporting competitive accuracy with substantially lower runtime than ambient‑space compression baselines.

**Strengths:**

- DMMD formalises distributional fidelity after decoding; RMMD/EMMD decompose the otherwise entangled optimisation. This clarifies targets vs surrogates.
- Thm. 3.3 links zero RMMD/EMMD to zero DMMD (Sec. 3.1.4), aligning the two‑stage training with the main goal.
- Across Swiss‑Roll/CT‑Slice/MNIST/Clusters, BDC reaches comparable or superior predictive metrics with markedly lower runtime, reinforcing practical value (impact). Exact BDC verifies the RMMD+EMMD bound on DMMD with Gaussian mixtures, directly validating the theory.

**Weaknesses:**

- The DMMD ≤ RMMD+EMMD bound (Thm. 3.5) requires the pull‑back kernel, yet the main experiments typically define kernels directly in latent space (Sec. 4: “Unless otherwise stated…”), so the bound may not apply in most results.
- The hybrid loss (Eq. (5)) fixes the relative scale (RMMD + MSRE/nd) without a tunable weight or ablation over trade‑offs, limiting guidance.
- On MNIST, M3D is run without its up‑sampling trick explicitly “to enable a fair comparison,” potentially under‑representing that method’s best‑known configuration.
- The joint DJMMD/RJMMD/EJMMD framework is well‑defined, but it is not always clear which experiments actually used the joint objectives vs marginal ones.

**Questions:**

- For each supervised task, explicitly state whether RJMMD/EJMMD (joint) or RMMD/EMMD (marginal) were used, and report the corresponding values. Specify the response kernel choices ($l$) per dataset (delta vs Gaussian on one‑hots; real‑valued $y$) and include an ablation on $l$.
- Is it possible to introduce a tunable weight, in Eq. (5), sweep this weight, and report sensitivity of test metrics and RMMD?

---

> ### Author Response · Authors · 2025-11-14
>
> We thank the reviewer for their time and constructive feedback. We hope that the following resolves some of your concerns.
>
> # **Weaknesses**
>
> ## **Weakness 1** – The DMMD ≤ RMMD+EMMD bound (Thm. 3.5) requires the pull‑back kernel...
>
> Thank you for raising this. Theorem 3.5 uses the pull-back kernel to provides fine-grained control of DMMD, which may be important in settings involving sensitive data (e.g., medical applications). However, as discussed in the paper, Theorem 3.3 is the main theoretical motivation for our approach: it shows that vanishing RMMD and EMMD implies vanishing DMMD for *any* characteristic kernel. Since our experiments primarily use standard characteristic kernels defined directly in latent space, Theorem 3.3 is the relevant guarantee for these results.
>
> In practice, using the pull-back kernel incurs increased computational cost as it requires repeated evaluation of the decoder. For this reason we use latent-space kernels in most experiments.
>
> *We will clarify this further in the final version to aid practitioners*.
>
> ## **Weakness 2** – The hybrid loss (Eq. (5)) fixes the relative scale (RMMD + MSRE/nd) without a tunable weight...
>
> We normalise data prior to training, and the MSRE term is further normalised by $1/(nd)$. This ensures that the MSRE does not grow arbitrarily with dataset size or dimensionality and places RMMD and MSRE on comparable scales. We chose an equal weighting between the two terms as it performed well empirically and avoided introducing an additional hyperparameter that could complicate optimisation.
>
> As requested by the reviewer, we conducted a new ablation experiment on the CT Slice dataset  where we introduce an explicit weight on the MSRE term. The resulting plot (https://ibb.co/27wyx8r7) shows that, once the RMMD is sufficiently weighted to prevent mixing collapse (see Figure 2), downstream performance remains relatively stable across a broad range of MSRE weights. This indicates that the method is not highly sensitive to the relative weighting and that the equal-weight choice is "good enough".
>
> We thank the reviewer for highlighting this direction. Incorporating an adaptive weighting scheme, where the RMMD–MSRE balance is updated during optimisation, for example following approaches such as [R1], is a promising avenue for future work and may improve performance.
>
> *We will include a thorough discussion of this in the final version*.
>
> ## **Weakness 3** – On MNIST, M3D is run without its up‑sampling trick...
>
> Thank you for highlighting this. We did not use the up-sampling trick in M3D to ensure a fair comparison with our competitor ADC, which, like M3D, produces a compressed set in the ambient space. In contrast, BDC performs bilateral compression by reducing both the sample size and the data dimension, so M3D is not solving the same problem setting. The up-sampling step in M3D is primarily designed to reduce memory usage, not computation, and increases the size of the compressed set by a factor equal to the amount of upsampling. This would substantially increase the computational cost of the downstream task and make fair comparison across methods more difficult. Importantly, even without up-sampling, M3D is significantly slower than BDC, so enabling it would only widen the runtime gap. For these reasons, we believe the configuration used in our experiments provides the most equitable basis for comparison.
>
> We also note that M3D is specifically designed for image-classification settings, whereas our framework applies additionally to more generla domains such regression and clustering. As such, MNIST represents only a small portion of the empirical contributions of this work, and we include M3D primarily for completeness rather than as a central comparison
>
> ## **Weakness 4** – The joint DJMMD/RJMMD/EJMMD framework is well‑defined but...
>
> Thank you for pointing this out. We agree that this could be made more explicit. In all experiments involving labels we use the joint DJMMD/RJMMD/EJMMD objectives, except where explicitly stated otherwise. *We will revise the final version to make this usage completely clear and unambiguous throughout the experimental section.*
>
> # **Questions**
>
> ## **Question 1** – For each supervised task...
>
> Thank you for this suggestion. As noted in our response to Weakness 4,  *we will make these choices more explicit for every  experiment in the final version*. We do note however that Section D.1.2 includes an ablation over the kernel choices $k$ and $l$ where we observe that the effect on performance is minimal, indicating robustness to the kernel selection. **We will emphasise this more clearly in the revised manuscript*.
>
> ## **Question 2** – Is it possible to introduce a tunable weight...
>
> As noted in our response to Weakness 2,  we have included new results on  test metrics (https://ibb.co/27wyx8r7).
>
> # **References**
> [R1] - Chen et al. GradNorm: Gradient Normalization for Adaptive Loss Balancing in Deep Multitask Networks. 2017.

---

> > ### Comment · Reviewer_KPqY · 2025-11-28
> >
> > Thanks for the response. Most of my concerns are addressed. I will keep my original score.

---

> > > ### Author Response · Authors · 2025-11-28
> > >
> > > Thank you for taking the time to read our comments. If you have any additional concerns or questions please do not hesitate to raise them.

---

### Official Review · Reviewer_5BrC · 2025-10-29

**Soundness:** 3
**Presentation:** 3
**Contribution:** 2
**Rating:** 4
**Confidence:** 4

**Summary:**

The paper proposes Bilateral Distribution Compression (BDC), a framework for compressing datasets along both the sample and feature dimensions while preserving their underlying distribution. The method introduces three kernel-based discrepancy measures—Reconstruction MMD (RMMD), Encoded MMD (EMMD), and Decoded MMD (DMMD)—and shows theoretically that minimizing RMMD and EMMD ensures a small DMMD.

BDC operates in two stages: (1) learning a low-dimensional representation via RMMD-based autoencoder training, and (2) optimizing a compact latent set via EMMD. The authors also extend BDC to supervised data using joint feature–label kernels.

Experiments on several synthetic and real datasets show that BDC achieves similar or better performance than existing distribution compression methods, with much lower computational cost.

**Strengths:**

S1 (Problem Formulation):
Clearly defines a new problem—bilateral distribution compression—that reduces both the number of samples and feature dimensions. This formulation fills a gap between dataset condensation and dimensionality reduction.

S2 (Theoretical Guarantees):
Provides rigorous proofs showing that minimizing RMMD and EMMD ensures small or vanishing DMMD, establishing a solid theoretical foundation. The link to PCA under the quadratic kernel further supports correctness and interpretability.

S3 (Clarity and Presentation):
The paper is clearly structured and easy to follow. Key concepts and notation are introduced systematically, with intuitive explanations and helpful figures to illustrate the two-stage optimization idea.

**Weaknesses:**

W1 (Comparison to Sequential Baselines):
The paper does not clearly demonstrate how BDC improves over a simple two-step approach that first performs Kernel Herding (or other MMD-based compression) followed by dimensionality reduction (e.g., PCA or autoencoder). The advantage of the proposed integrated formulation remains conceptually underexplained.

W2 (Ablation and Sensitivity):
The analysis of hyperparameters—such as latent dimension p, compressed size m, and kernel type—is limited. More systematic ablation studies would help clarify robustness and the contribution of each component.

**Questions:**

Q1 (Regarding W1 – Comparison to Sequential Baselines):
How does BDC fundamentally differ from a sequential approach that first applies Kernel Herding (or similar MMD-based subset selection) and then performs dimensionality reduction with PCA or an autoencoder?
Specifically, does the proposed two-stage optimization provide any theoretical or empirical advantage beyond this straightforward pipeline?

Q2 (Regarding W2 – Ablation and Sensitivity):
Could the authors provide more analysis on how the choice of latent dimension p, compressed size m, and kernel type affects both reconstruction fidelity and downstream task performance?
For instance, is there a systematic way to choose these parameters, or are the results highly sensitive to them?

**Details Of Ethics Concerns:**

No ethical concerns are apparent.

---

> ### Author Response · Authors · 2025-11-14
>
> We thank the reviewer for their time and constructive feedback. We hope that the following resolves some of your concerns.
>
> # **Weaknesses**
>
> ## **Weakness 1** – Comparison to Sequential Baselines
>
> Thank you for raising this point. The advantages of an integrated formulation are substantial. First, performing kernel herding or ADC in the ambient space is far more expensive, as shown in Table 1: gradient descent in high dimensions incurs a significant computational cost. By contrast, BDC learns a low dimensional latent space *before* performing compression, resulting in substantial speed-ups.
>
> Second, a sequential approach necessarily loses information during the initial compression step, which in turn limits the quality of the learned latent space. Since the latent geometry is critical for downstream tasks, learning it directly from the full dataset is important. To illustrate this, we conducted an additional experiment following the reviewer's suggestion: we first run ambient-space ADC and then apply PCA to the resulting compressed set. The results on CT-Slice (provided at the link below) show that BDC (red/green) significantly outperforms this sequential baseline (purple), while also being considerably faster.
>
> https://ibb.co/7tKJSdzx
>
> Third, applying PCA in general does not provide distributional guarantees. As noted in Theorem 3.1, PCA preserves only second-order structure, whereas our approach, using a characteristic kernel, preserves *all* moments of the distribution under Theorems 3.3 and 3.5.
>
> *We will include an extended discussion of this in the final version of the paper.*
>
> ## **Weakness 2** – Ablation and Sensitivity
>
> Thank you for highlighting this important point. We fully agree that ablation studies play a crucial role in evaluating robustness, however, we respectfully disagree that these aspects are particularly underexplored. Section D.1 contains several ablations analysing the effect of key hyperparameters. In particular, we study kernel choice in Section D.1.2, neural architecture in Section D.1.5, and latent dimension in Sections D.1.6 and D.1.7. These results indicate that the method is not overly sensitive to reasonable choices of these parameters. During the rebuttal process we realised that we had not included an ablation over the compressed set size m; we now provide this for the CT Slice dataset at the following link (https://ibb.co/QFKmnWW5). We hope this additional experiment addresses the reviewer’s concern.
>
> *With the extra space available, we will include a dedicated ablation subsection in the main paper to make these findings more accessible.*
>
> # **Questions**
>
> ## **Question 1** – Regarding Weakness 1
>
> We hope that our response to Weakness 1 clarifies this point. The differences between BDC and a sequential “compression-then-dimension-reduction’’ pipeline are substantial. BDC is considerably faster, more performant (https://ibb.co/7tKJSdzx) and comes with theoretical guarantees. That is, applying PCA or an autoencoder after compression does not preserve the distribution:  Theorem 3.1 tells us that PCA captures only second-order structure, while BDC with a characteristic kernel preserves all moments of the distribution under Theorems 3.3 and 3.5.
>
> ## **Question 2** – Regarding Weakness 2
>
> As noted in our response to Weakness 2, Section D.1 provides empirical analysis of the effect of kernel choice (D.1.2), neural architecture (D.1.5), latent dimension p (D.1.6–D.1.7), and, the compressed size m (https://ibb.co/QFKmnWW5). These results indicate that BDC is not overly sensitive to reasonable parameter choices.
>
> Regarding the reviewer’s second point, the choice of kernel reflects the inductive bias of the practitioner, and there is substantial work on kernel selection e.g. [R1]. For the compressed size m, the trade-off is mainly computational: as shown in the new CT Slice ablation (https://ibb.co/QFKmnWW5), increasing m improves fidelity but increases cost, so practitioners can select m according to resource constraints.
>
> The latent dimension p plays a different role, as it determines the geometry in which downstream tasks operate. If p is too small, important structure may be lost; if too large, optimisation may become unnecessarily difficult. Fortunately, there is extensive literature on intrinsic dimension estimation [R2], and existing estimators provide practical guidance. In Section D.1.6 (Figure 18), the optimal p for both BDC-L and BDC-NL is around 10. Running intrinsic-dimension estimators from the skdim package [R2] yields an average estimate of around 11.5, suggesting close alignment between the empirically optimal choice and standard estimators.
>
> *We will include a discussion of the above in the final version to make the guidance clearer for practitioners.*
>
> # **References**
> [R1] - Thomas Hofmann et al. Kernel methods in machine learning. 2007
>
> [R2] - Jonathan Bac et al. Scikit-dimension: a Python package for intrinsic dimension estimation. 2021

---

> > ### Comment · Reviewer_5BrC · 2025-11-26
> >
> > Thank you for your response. I have increased the score for Contribution.

---

> > > ### Author Response · Authors · 2025-11-27
> > >
> > > Thank you for taking the time to read our comments and for raising your contribution score. If you have any further questions or concerns, please do not hesitate to let us know.

---

### Official Review · Reviewer_NwDn · 2025-10-31

**Soundness:** 2
**Presentation:** 3
**Contribution:** 2
**Rating:** 6
**Confidence:** 3

**Summary:**

This paper introduces Bilateral Distribution Compression (BDC), a framework that simultaneously reduces both the number of samples and the dimensionality of a dataset while preserving its underlying distribution. The method combines ideas from kernel-based distribution matching (MMD) and autoencoding. It defines a new measure, the Decoded MMD (DMMD), which quantifies how well a compressed latent set (after decoding) approximates the original data distribution. The paper proposes a two-stage optimization: first, train an autoencoder using Reconstruction MMD (RMMD); second, optimize a compressed latent set using Encoded MMD (EMMD). The authors provide theoretical guarantees linking RMMD, EMMD, and DMMD, and show empirical results on synthetic and real datasets (e.g., Swiss-Roll, CT-Slice, MNIST) demonstrating improved or comparable performance to existing methods (e.g., ADC, M3D) at lower cost.

**Strengths:**

S1. The paper is well-motivated and addresses a relevant problem: compressing datasets in both sample and feature dimensions simultaneously.

S2. The method is conceptually elegant, bridging kernel-based distribution matching with representation learning.

S3. The theoretical analysis (linking RMMD, EMMD, and DMMD) provides clear intuition and guarantees for the proposed two-stage approach.

S4. The empirical results are diverse, spanning regression, classification, and clustering, with consistent improvements in both runtime and accuracy over strong baselines.

S5. The approach is domain-agnostic and task-agnostic, potentially useful beyond the demonstrated applications.

**Weaknesses:**

W1. The experimental section is somewhat limited: while results are shown on a few datasets, they are relatively small and mostly standard (swiss-roll, mnist, ce-slice). Evaluation on larger-scale or higher-dimensional modern datasets would strengthen the empirical case.

W2. There is no ablation study on key hyperparameters such as latent dimension, kernel choice, or autoencoder architecture. These factors likely affect the trade-off between reconstruction fidelity and compression rate.

W3. Can you compare withmore  recent strong baselines in dataset condensation and generative compression (e.g., diffusion-based or adversarial distillation methods)?

**Questions:**

See above

---

> ### Author Response · Authors · 2025-11-14
>
> We thank the reviewer for their time and generous feedback. We hope that the following resolves your remaining concerns.
>
> # **Weaknesses**
>
> ## **Weakness 1** – The experimental section is somewhat limited...
>
> While we agree that evaluation on even larger datasets would be interesting, our experiments already include settings that are considered large within the distribution compression literature, such as CT Slice (50k samples in 384 dimensions) and a 100k-sample clustering dataset in 500 dimensions. These go beyond what existing methods such as kernel herding [R1], kernel thinning [R2] ,and gradient flow [R3], typically evaluate on, particularly in terms of dimension.
>
> In Section C.4 we provide a complexity analysis showing that BDC scales linearly in time and memory with both the number of samples and the ambient dimension, so in principle the method should extend naturally to larger problems.
>
> Our decision not to include experiments at larger scale is due solely to limited computational resources, rather than any limitation of the method itself. We view exploring such large scale applications as an exciting direction for future work.
>
> ## **Weakness 2** – There is no ablation study on key hyperparameters such as latent dimension, kernel choice, or autoencoder architecture...
>
> Thank you for raising this important point. We agree that ablation studies are essential for assessing robustness in practical settings. *We refer the reviewer to Section D.1, which includes several ablation experiments.* For example, we study the effect of kernel choice in Section D.1.2, neural architecture in Section D.1.5, and latent dimensionality in Sections D.1.6 and D.1.7. During the rebuttal process we also identified that we had not included an ablation over the compressed set size; we now provide this for the CT Slice dataset at the following link (https://ibb.co/QFKmnWW5). We also include additional results on the effect of kernel choice on RMMD and EMMD for the Swiss-Roll data at the following link (https://ibb.co/psVp4Zr).
>
> *With the additional space available, we will include a dedicated ablation section in the main paper to make these results more visible.* We hope this address your concern.
>
> ## **Weakness 3** –  Can you compare withmore recent strong baselines in dataset condensation and generative compression (e.g., diffusion-based or adversarial distillation methods)?
>
> Our framework targets distribution compression in *both sample size and data dimension*, which, to the best of our knowledge, is not provided by existing neural approaches. We already compare against ADC, the strongest non dimensional distribution compression baseline, and observe consistently improved performance.
>
> We include results for M3D, a recent neural image condensation method, and show favourable performance on MNIST. More recent diffusion and adversarial distillation methods (discussed in Section A.1.2) are also designed specifically for *image classification*. These approaches optimise latent codes inherited from a pre trained generator, but the codes are never used as downstream representations; they serve only as intermediates for synthesising images for classifier training. Consequently, *they do not perform dimension reduction*, their latent dimensionality is fixed by the generator architecture, and they are applicable only in domains where suitable pre trained generative models exist, preventing comparison to our experimental use-cases. They also require substantially more computational resources.
>
> In contrast, our method applies to a much broader range of tasks, including regression and clustering, learns a latent space directly from the data with tunable dimensionality, and provides theoretical guarantees of distribution level preservation (Theorems 3.3 and 3.5), which generative approaches do not.
>
> *We will expand the related work section in the final version to make these distinctions clearer.*
>
> # **References**
> [R1] - Yutian Chen et al. “Super-Samples from Kernel Herding”. 2012
>
> [R2] - Raaz Dwivedi and Lester Mackey. "Kernel Thinning". 2024
>
> [R3] - Michael Arbel, Anna Korba, Adil Salim, Arthur Gretton. Maximum Mean Discrepancy Gradient Flow. 2019

---

### Official Review · Reviewer_3hib · 2025-10-31

**Soundness:** 3
**Presentation:** 3
**Contribution:** 3
**Rating:** 4
**Confidence:** 3

**Summary:**

This paper proposes a framework called Bilateral Distribution Compression (BDC) to address the problem of modern datasets being simultaneously large in both sample size ($n$) and dimensionality ($d$)1111. The authors introduce the Decoded Maximum Mean Discrepancy (DMMD) as a core metric to quantify the difference between the original data distribution and the compressed set after being decoded from a low-dimensional latent space.

**Strengths:**

1.Originality & Significance: The paper formulates a novel and highly significant problem. Most methods address either sample reduction or dimensionality reduction; tackling both simultaneously is a key challenge for modern, large-scale datasets. The proposed BDC framework is a practical, scalable solution with linear time/memory complexity ($O(nd)$), which is a major advantage.

2.Quality (Theory & Empirical): The method is well-grounded. The introduction of the three MMD variants clearly defines the objectives. The two-stage optimization is a logical decoupling of the problem, justified by sound theoretical bounds.

**Weaknesses:**

1.The method's efficacy relies heavily on the manifold hypothesis. If the data lacks a low-dimensional structure, BDC's forced dimensionality reduction can underperform ambient-space-only methods (ADC)

2.The choice of the latent dimension $p$ is purely heuristic and lacks principled guidance.

**Questions:**

1.Regarding the choice of $p$: Given that the choice of $p$ is critical to performance, can the authors provide a more principled strategy for selecting $p$? For example, could the RMMD-vs-$p$ curves be used to automatically find an 'Elbow point' for $p$?

2.Regarding the $RMMD+MSRE$ hybrid objective: In BDC-NL, how was the relative weighting between RMMD and MSRE set? How sensitive is the downstream task performance to this weight?

---

> ### Author Response · Authors · 2025-11-14
>
> We thank the reviewer for their time and constructive feedback. We hope that the following resolves your concerns.
>
> # **Weaknesses**
>
> ## **Weakness 1** – The method's efficacy relies heavily on the manifold hypothesis...
>
> Thank you for raising this point. We agree that, as discussed in Section A.2.2, dimensionality reduction implicitly assumes that the data admit a low-dimensional latent structure that captures the essential information in the ambient space. If such a structure truly does not exist, then one should not expect BDC to outperform purely ambient-space methods such as ADC. However, we have shown in a number of experimental scenarios that BDC does in fact outperform ADC, including on real data.
>
> More generally, the manifold hypothesis is a widely accepted and empirically supported principle [R1], observed across many modern datasets in fields such as vision, language and biological data analysis. In many practical scenarios, latent structure does exist and can be exploited by BDC to improve both efficiency and compression quality.
>
> *We will make this implicit assumption more explicit in the final version.*
>
> ## **Weakness 2** – The choice of the latent dimension $p$ is purely heuristic and lacks principled guidance.
>
> Thank you for raising this, we agree that the paper should include discussion on how to choose $p$ in practice. The dimension $p$ determines the geometry in which downstream tasks operate: if $p$ is too small, essential structure may be lost, while if $p$ is too large, optimisation may become unnecessarily difficult. Fortunately, there is substantial prior work on estimating intrinsic dimensionality [R2], providing principled guidance for choosing $p$. In Section D.1.6 (Figure 18), we observe that the empirically optimal $p$ for both BDC-L and BDC-NL is around 10. Applying several intrinsic-dimension estimators from the \texttt{skdim} package [R2] to the same dataset yields an average estimate of roughly 11.5, showing close agreement between standard estimators and the value that performs best in practice.
>
> *We will make sure to include a discussion on choosing $p$ in the final version to provide more explicit guidance for practitioners.*
>
> # **Questions**
>
> ## **Question 1** – Regarding the choice of $p$: Given that the choice of $p$ is critical to performance...
>
> As noted in our response to Weakness 2, we agree that the paper should provide clearer guidance on how to select $p$ in practice. *We will include discussion on existing prior work on intrinsic-dimension estimation [R2], which offers a principled way to guide the choice of $p$*.
>
> ## **Question 2** – 2. Regarding the $RMMD+MSRE$ hybrid objective...
>
> Thank you for raising this interesting question.
>
> We normalise all datasets prior to training, and the MSRE term is further normalised by $1/(nd)$. This ensures that the MSRE component of the loss does not grow arbitrarily with dataset size or dimensionality and places RMMD and MSRE on comparable scales. For this reason, we chose an equal weighting between the two terms: it performed well empirically and avoided introducing an additional hyperparameter that could complicate optimisation.
>
> Following the reviewer’s suggestion, we conducted a new experiment on the CT Slice dataset in which we introduced an explicit weight on the MSRE term. The resulting plot (https://ibb.co/27wyx8r7) shows that, once the RMMD is sufficiently weighted to prevent mixing collapse (see Figure 2), downstream performance remains relatively stable across a broad range of MSRE weights. This indicates that the method is not highly sensitive to the relative weighting and that the equal-weight choice is "good enough".
>
> We thank the reviewer for highlighting this direction. Incorporating an adaptive weighting scheme, where the RMMD–MSRE balance is updated during optimisation, for example following approaches such as [R3], is a promising avenue for future work and may improve performance.
>
> *We will include a thorough discussion of this in the final version*.
>
> # **References**
> [R1] - Nick Whiteley, Annie Gray, Patrick Rubin-Delanchy. Statistical exploration of the Manifold Hypothesis. 2022.
>
> [R2] - Jonathan Bac et al. Scikit-dimension: a Python package for intrinsic dimension estimation. 2021.
>
> [R3] - Chen et al. GradNorm: Gradient Normalization for Adaptive Loss Balancing in Deep Multitask Networks. 2017.

---

### Official Review · Reviewer_QZTY · 2025-11-01

**Soundness:** 2
**Presentation:** 2
**Contribution:** 2
**Rating:** 6
**Confidence:** 2

**Summary:**

This paper proposes Bilateral Distribution Compression (BDC), a framework that jointly reduces both the sample size and the feature dimensionality of large datasets while preserving their underlying distribution. The method builds upon kernel-based distribution matching using Maximum Mean Discrepancy (MMD) and introduces a two-stage optimization:
(1) learning a low-dimensional representation via minimizing Reconstruction MMD (RMMD), and
(2) optimizing a latent compressed set via Encoded MMD (EMMD).

**Strengths:**

The idea of bilateral compression:  reducing both data size and dimensionality simultaneously.   is conceptually elegant and practically useful, extending traditional dataset condensation methods.

The paper provides theoretical guarantees linking RMMD, EMMD, and DMMD, giving the approach a clear probabilistic interpretation.

BDC is task-agnostic and can handle both supervised and unsupervised data. It’s applicable beyond specific domains like images.

The experiments, though modest in scale, demonstrate the method’s efficiency and ability to preserve downstream performance across regression, classification, and clustering tasks.

**Weaknesses:**

The framework mainly reorganizes existing MMD techniques (RMMD/EMMD/DMMD) in a two-stage pipeline; the novelty is more in formulation than in fundamental methodology.

Most experiments use relatively simple datasets (Swiss Roll, MNIST) and small-scale setups. It is unclear whether the method scales to large, high-dimensional real-world datasets (e.g., ImageNet or tabular domains).

Comparisons are limited to kernel MMD and ADC methods; more modern or neural-based data condensation approaches (e.g., Diffusion-based dataset distillation) are missing.

**Questions:**

How sensitive is the final performance to the choice of kernel functions (e.g., Gaussian vs. polynomial) in RMMD/EMMD?

How does BDC scale when applied to datasets with millions of samples or high-dimensional image data?

Could the method be integrated with diffusion-based or transformer-based representation learners instead of simple autoencoders?

In supervised settings, how are label distributions preserved during compression, especially for imbalanced datasets?

---

> ### Author Response · Authors · 2025-11-14
>
> We thank the reviewer for their time and constructive feedback. We hope that the following resolves some of your concerns.
>
> # **Weaknesses**
>
> ## **Weakness 1** – The framework reorganises existing MMD techniques...
>
> We agree that the MMD itself is a well established discrepancy measure. However, to the best of our knowledge *this paper is the first to formalise RMMD and DMMD*. Related work has used MMD in latent spaces (EMMD), such as M3D, but our contribution defines RMMD and DMMD, establishes their properties, and provides theoretical analysis for EMMD, RMMD and DMMD in Theorems 3.1, 3.4 and 3.5.
> ## **Weakness 2** – Experiments use simple datasets...
> We appreciate the reviewer’s concern. While we do not include ImageNet, we evaluate on high dimensional tabular datasets such as CT Slice (50k samples, 384 dimensions) and a 100k sample clustering dataset in 500 dimensions. *Related approaches, as discussed in Section A.1.2 are not applicable to regression or clustering at any scale, so our experiments already extend beyond prior work.*
>
> Section C.4 gives a time and memory complexity analysis showing *BDC scales linearly in sample size and dimensionality, suggesting feasibility on larger datasets*. Our omission of very large scale experiments reflects resource limits, not methodological constraints.
>
> *We will move a complexity discussion into the main paper to make the scalability clearer.*
>
> ## **Weakness 3** – Comparisons are limited...
> We thank the reviewer for raising this important concern. Our contribution provides a distribution compression framework that extends compression along both sample size and data dimension. To the best of our knowledge, *existing approaches do not offer this combination*. We already compare against the current state of the art for standard (non dimensional) distribution compression, ADC, and observe that *our method consistently outperforms it across a range of settings*.
>
> Neural based dataset condensation methods typically focus on image classification, whereas *our framework applies further to regression and clustering domains*. We do include a comparison to M3D, a recent neural based image condensation method, and show favourable performance on MNIST.
>
> Existing diffusion based methods (Section A.1.2) are not directly comparable: they optimise latent codes inherited from a pre trained generator, but these codes are never used as downstream representations and are only intermediates for generating images. They therefore *do not perform dimension reduction*; their latent dimensionality is fixed by the generator; and they require pre trained models and substantial compute. In contrast, our method learns a latent space directly from data with tunable dimensionality, applies to arbitrary domains via kernels, and provides theoretical distribution level guarantees (Theorems 3.3 and 3.5), which diffusion methods do not.
>
> *We will expand the related work section to clarify these differences more clearly.*
>
> # **Questions**
>
> ### **Question 1** – Sensitivity to kernel choice...
>
> In Section D.1.2 we show that *downstream performance is stable when replacing the Gaussian kernel with the IMQ kernel*, indicating that *BDC is not overly sensitive to kernel choice*. We do note that polynomial kernels are not characteristic, so they do not induce a valid MMD and are thus not appropriate here.
>
> The figure (https://ibb.co/psVp4Zr) reports RMMD and EMMD (via BDC-NL) for the swiss-roll dataset using IMQ and Gaussian kernels for both $k$ and $l$. It does not appear that kernel choice has had a large effect, given the scales under consideration.
>
> *We will include a more dedicated section on kernel ablation for RMMD and EMMD in the final work*.
>
> ### **Question 2** – How does BDC scale?
>
> As in Weakness 3, *BDC scales linearly in both the number of target samples and the ambient dimensionality*. We did not run experiments on datasets with millions of samples or high dimensional image benchmarks due to resource constraints, but the linear scaling indicates that the method should extend naturally to larger problems in principle.
> ### **Question 3** – Integration with diffusion or transformer based learners...
>
> Thank you for raising this interesting point. As discussed in Section A.2.1, BDC may in principle operate in the latent space of any dimension reduction method, e.g KPCA, UMAP, or neural methods. However, to retain Theorems 3.3 and 3.5, reconstructions must minimise RMMD; these approaches may not satisfy this without modification. Exploring this is a promising direction for future work but lies beyond the scope of the present contribution.
>
> ### **Question 4** – Preserving label distributions...
>
> Section 3.3 extends BDC to operate on the joint distribution of features and labels. Labels are optimised alongside the features, so the compressed set preserves the joint distribution under Theorems 3.3 and 3.5. Therefore imbalances are retained as we preserve the joint rather than the marginal.

---

### Author Response · Authors · 2025-11-30

# *Final Remarks*

We thank the reviewers for their time and for providing thoughtful, constructive feedback. We regret that we did not have the opportunity to fully engage in the peer review process due to the unfortunate issue with OpenReview.

In this final remark, we address several common concerns raised during the review process.

## *Ablation Studies*
Several reviewers expressed concern about a lack of ablation studies.

We first note that the paper already contains ablation studies. For example, we study the effect of kernel choice in Section D.1.2, neural architecture in Section D.1.5, and latent dimensionality in Sections D.1.6 and D.1.7. **We agree that these existing results should be highlighted more explicitly in the final paper, where we will include a dedicated ablation section**.

**During the rebuttal process we also performed three additional ablation studies**; on the size of the compressed set on CT-Slice  (https://ibb.co/QFKmnWW5); on the effect of kernel choice on RMMD and EMMD for Swiss-Roll  (https://ibb.co/psVp4Zr); on a new explicit weight on the MSRE term in the hybrid loss (see Equation 5). The results (https://ibb.co/27wyx8r7) show that, once the RMMD is sufficiently weighted to prevent mixing collapse (see Figure 2), performance remains stable across a broad range of weights.

**These additional ablation results support those which already exist in the work, showing that the method is not overly sensitive to sensible choices of hyperparemeters.**

## *Choice of Latent Dimension p*
A few reviewers raised concern about how one should choose the size of the latent dimension $p$.

The dimension $p$ determines the geometry in which downstream tasks operate: if  $p$ is too small, essential structure may be lost, while if $p$ is too large, optimisation may become unnecessarily difficult. Fortunately, there is substantial prior work on estimating intrinsic dimensionality [R1], providing principled guidance for choosing $p$. In existing ablation studies, Section D.1.6 (Figure 18) observes that the empirically optimal $p$ for both BDC-L and BDC-NL is around 10. Applying several intrinsic-dimension estimators from the *skdim* package [R1] to the same dataset yields an average estimate of roughly 11.5, showing close agreement between standard estimators and the value that performs best in practice.

**We will include additional guidance reflecting this discussion in the final version.**

## *Comparison to Neural Baselines*
A few reviewers expressed concern that we should compare to recent neural-based dataset condensation methods.

We first stress that our contribution provides a distribution compression framework that extends compression along both sample size and data dimension. To the best of our knowledge, **no existing approaches offers this combination.** We already compare against the current state of the art for standard (non dimensional) distribution compression, ADC, and observe that **our method consistently outperforms it across a range of settings**.

**Additionally, we already include comparison to a recent neural image condensation method (M3D), and show very favourable performance on MNIST.**

Going further, recent diffusion and adversarial distillation methods (Section A.1.2) are designed specifically for image classification. They optimise latent codes inherited from a pre trained generator, but the **codes are never used as downstream representations**; they serve only as intermediates for synthesising images for classifier training. Thus, **they do not perform dimension reduction**, and they are applicable only in domains where suitable pre-trained generative models exist, **preventing comparison to our experimental use-cases**. They also require substantially more computational resources.

In contrast, our method applies to a much broader range of tasks, including regression and clustering, provides additional compression via dimensionality reduction, and provides theoretical guarantees of distributional preservation (Theorems 3.3, 3.5), which generative approaches do not.

**We will ensure that the existing related work section in the main body clarifies these differences more clearly.**
## *Larger Experiments*
Finally, some reviewers expressed a desire to see the method applied to larger datasets such as ImageNet.

We already evaluate on high dimensional datasets such as CT Slice (50k samples, 384 dimensions) and a 100k sample clustering dataset in 500 dimensions. Additionally, Section C.4 provides a time and memory complexity analysis showing **BDC scales linearly in sample size and dimensionality, suggesting feasibility on larger datasets**. Our omission of very large scale experiments reflects resource limits, not methodological constraints.

**We will move a complexity discussion into the main paper to make the scalability clearer.**

## *References*

[R1] - Jonathan Bac et al. Scikit-dimension: a Python package for intrinsic dimension estimation. 2021.

---

### Meta-Review · Area_Chair_caG1 · 2026-01-02

**Summary:**

The paper proposes Bilateral Distribution Compression (BDC), a method for compressing distributions in a way that simultaneously reduces both the number of points and the dimension. The method trains an autoencoder to reduce the dimension and then selects a coreset in the latent space.

Reviewers appreciated the merits of the problem and the solution, the clear presentation, and the clean theoretical analysis, even though it is idealized and asymptotic. Concerns were raised about the level of conceptual novelty over existing techniques used in combination; the limited guidance on hyperparameter settings; and the scope of the experimental evaluation. The authors' rebuttal was comprehensive and helpful, though its content did not materially change the situation regarding these points.

This is a very borderline submission. It is a solid and well-thought out methodology paper that could find an audience. There are no major weaknesses that would bar acceptance. However, the conceptual and technical scope might be deemed somewhat limited for ICLR, and no strong consensus for acceptance has been formed. Considering the original reviews and the authors' rebuttal, this situation would have likely persisted if discussions could proceed as usual, placing the paper close but under the acceptance bar.

**Reviewer Concerns:**

The rebuttal expanded the guidance on hyperparameter the setting. As for the scale of experiments, the rebuttal posits they are not needed since the method scales linearly with $n$ and $d$. I do not find this satisfactory as the method invokes computationally nontrivial components (the autoencoder) and has significant dependence on other parameters ($B^2, m, E$). The authors' remark on computational limitations is understandable. Nonetheless, the asymptotic dependence on $n$ and $d$ by itself does not obviously implies scalability and does not obviate the usefulness of larger-scale experiments. Furthermore, larger-scale experiments are meant not only for substantiating the feasibility of scale but also its interplay with utility.

**Reviewer Scores:**

Considering the original reviews and the content of the rebuttal, I believe the scores would have likely continued to hover around the borderline and would have not received picture-chaging updates.

---

### Decision · Program_Chairs · 2026-01-26

Reject